# PASER: Post-Training Data Selection for Efficient Pruned Large Language Model Recovery

**Bowei He**[1]*, **Lihao Yin**[2], **Hui-Ling Zhen**[2], **Xiaokun Zhang**[1], **Mingxuan Yuan**[2], **Chen Ma**[1]†
[1] City University of Hong Kong, [2] Huawei Noah's Ark Lab, Hong Kong,

## Abstract

Model pruning is an effective approach for compressing large language models (LLMs). However, this process often leads to significant degradation of model capabilities. While post-training techniques such as instruction tuning are commonly employed to recover model performance, existing methods often overlook the uneven deterioration of model capabilities and incur high computational costs. Moreover, some irrelevant instructions may also introduce negative effects to model capacity recovery. To address these challenges, we propose the **P**ost-training d**A**ta **S**election method for **E**fficient pruned large language model **R**ecovery (**PASER**). PASER aims to identify instructions to recover the most compromised model capacities with a certain data budget. Our approach first applies manifold learning and spectral clustering to group recovery instructions in the semantic space, revealing capability-specific instruction sets. Then, the data budget is adaptively allocated across clusters by the degree of corresponding model capability degradation. In each cluster, we prioritize data samples that lead to the most decline of model performance. To mitigate potential negative tuning effects, we also detect and filter out conflicting or irrelevant recovery data. Extensive experiments demonstrate that PASER significantly outperforms conventional baselines, effectively recovering the general capabilities of pruned LLMs while utilizing merely 4%-20% of the original post-training data. We provide the code repository in Link.

## 1 Introduction

Model pruning, which aims at reducing model parameter amounts while maintaining model capabilities, has been a promising approach for large language model (LLM) compression. Mainstream LLM pruning schemes are unstructured (Frantar & Alistarh, 2023), semi-structured (Sun et al., 2024), and structured pruning (Ma et al., 2023). Even though effective, the model capability degradation after pruning is almost unavoidable, especially under high pruning ratios. This degradation phenomenon is often more severe for the structured pruning scheme (Dong et al., 2024), which has been widely adopted in industrial LLM compression thanks to its hardware-friendly property. Therefore, recovery post-training after pruning has been a silver bullet to address this issue (Ma et al., 2023; Zhao et al.). Among various types of data including pre-training corpora and extensive fine-tuning datasets (Xia et al., 2024a; Sun et al., 2024), instruction tuning data has demonstrated unique advantages for efficient capability recovery (Ma et al., 2023; Zhao et al.; Zhang et al., 2024; Chen et al., 2023). Compared to recovery via continued pre-training which requires massive computational resources, instruction tuning enables effective restoration with a much smaller data scale by explicit supervision. Furthermore, through the diverse task coverage, like language modeling, common sense reasoning, mathematical problem solving, and code generation (Taori et al., 2023; Wu et al., 2024), instruction tuning preserves the model's general-purpose capabilities while preventing over-specialized recovery.

Conventional schemes (Ma et al., 2023) usually employ the full instruction tuning datasets like Alpaca (Taori et al., 2023) to conduct the recovery post-training. However, this can bring significant computation overhead and even unsatisfactory recovery performance (See Appendix A). An intuitive solution is to leverage part of the original data for training, thus consuming less data and correspondingly reducing the computation resource demand. Nevertheless, directly utilizing the naively split data

---

*Work done as an intern in Huawei Noah's Ark Lab, Hong Kong, `boweihe2-c@my.cityu.edu.hk`
†Corresponding author, `chenma@cityu.edu.hk`

subset (e.g., first 20% of the data), can lead to sub-optimal performance, or even further performance degradation. Moreover, the recovered performance considerably varies for models trained with different subsets, implying the sensitivity to data constitution. Therefore, selecting the most valuable instruction-tuning data that can contribute to recovery performance positively and reduce training costs becomes crucial. Though previous works have noticed the significance of selecting high-quality data for general instruction tuning (Wang et al., 2024; Cao et al.; Li et al., 2024a), few of them are specifically designed for the recovery post-training. Note such general high-quality standards, such as clear structure and natural, human-like expressions, may not effectively target the specific capabilities (e.g., math, code) severely compromised during pruning, which is essential for recovery post-training.

Considering the above limitations, the ideal recovery post-training data selection approach should exhibit the following properties: (1) **Targeted and Balanced Capability Recovery**: Given the uneven deterioration of different capabilities in the pruning (see Appendix A), the ideal selection method should identify and prioritize the severely-degraded ones, while ensuring balanced recovery of the model's overall functionality. (2) **Recovery Training Efficiency**: Limited computing resources pose serious efficiency challenges to the LLM recovery post-training. An ideal method should be able to select instructions that are both most beneficial for recovery and light in computational cost, thereby accelerating the training process and optimizing resource utilization. (3) **Mitigation of Negative Tuning Effects**: Recognizing that not all instruction data is beneficial for model recovery, an optimal approach should not only identify the most advantageous instructions, but also filter out potentially harmful or irrelevant ones. This significantly reduces the risk of negative tuning effects during the recovery training, ensuring that the selected data contributes positively to model recovery.

To achieve such goals, we propose the **P**ost-training d**A**ta **S**election method for **E**fficient pruned large language model **R**ecovery (**PASER**). First, we perform semantic-structural recovery instruction clustering to obtain data groups corresponding to different LLM capabilities. Second, we select recovery instructions in a capability degradation-aware manner, where the data budget is allocated across different clusters based on their corresponding capability degradation degrees. In particular, when selecting samples within each capability cluster, the post-training computation cost of each sample is also taken into consideration to optimize the efficiency of the recovery process. Finally, we construct the concept consistency graph to maintain the semantic consistency across selected instructions, thus preventing the introduction of conflicting or irrelevant samples. Furthermore, we analyze the time complexity (Section 3.5) and error bound (Appendix C) of PASER theoretically, which guarantee the efficiency and effectiveness, respectively. We take the LLaMA 2/3/3.1, Baichuan2, Qwen2.5/3, and Mixtral 8×7B as the target LLMs and perform the experiments under different LLM pruning schemes and different-sized instruction tuning datasets. The comparison with random and conventional instruction tuning data selection baselines demonstrates that PASER can more effectively enhance the recovered LLM performance on language modeling and various reasoning tasks. Meanwhile, the recovery training overhead can also be reduced significantly.

## 2 RELATED WORKS

**Large Language Model Pruning** can be generally divided into three categories: unstructured pruning, semi-structured pruning, and structured pruning. Unstructured pruning removes individual weights without structural constraints, with representative works including SparseGPT (Frantar & Alistarh, 2023), Wanda (Sun et al., 2024), BESA (Xu et al., 2024b), and OWL (Yin et al., 2024). This technique allows for maximum flexibility in weight selection and can achieve high compression rates while maintaining model performance. However, the resulting irregular sparsity patterns limits the practical acceleration. Semi-structured pruning (Guo et al., 2024; Malla et al., 2024; Frantar & Alistarh, 2023; Sun et al., 2024) targets specific patterns like N:M sparsity, balancing flexibility and hardware efficiency. Structured pruning approaches like LLM-Pruner (Ma et al., 2023) and SliceGPT (Ashkboos et al., 2024) remove entire structural components, offering better hardware compatibility and attracting industry attention (Ko et al., 2023; An et al., 2024; Song et al., 2024; Xia et al., 2024a). However, structured pruning faces more severe performance degradation, highlighting the importance of recovery post-training.

**Instruction Tuning** has emerged as a crucial technique for enhancing LLMs (Wei et al.; Wang et al., 2023), improving their adaptability to novel tasks (Sanh et al.; Liang et al.; Zhou et al., 2024). Recent works have explored instruction tuning as a post-compression recovery mechanism (Zhao et al.; Ma et al., 2023). While promising, this combination faces challenges from reduced model capacity and computational costs. Most current approaches use general instruction datasets without considering

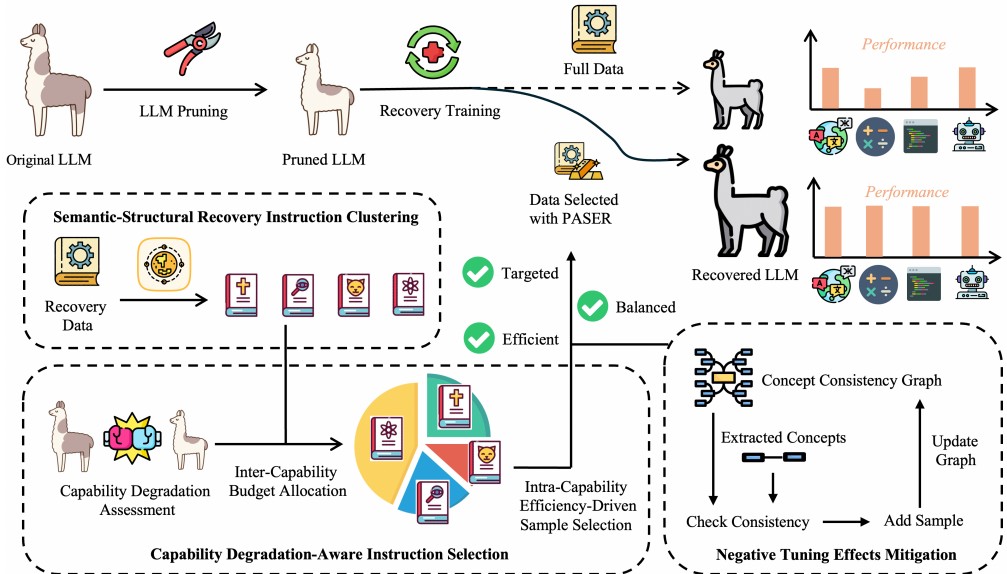

Figure 1: Visualization for our proposed PASER recovery post-training data selection framework.

compressed model's characteristics or disproportionately affected capabilities. Our work addresses these gaps by proposing a novel framework for post-training data selection in pruned LLM recovery.

**Data Selection** has been a widely studied approach to improve instruction tuning efficiency and model performance. Instruction Mining (Cao et al.) defines a set of rule-based evaluation metrics to assess the quality of instructional data. IFD (Li et al., 2024a) and Nuggets (Li et al., 2024b) leverage trainable LLMs to develop scoring functions for instruction data selection. Other works like LESS (Xia et al., 2024b) and DELIFT (Agarwal et al., 2025) rely on information value analysis tools like influence function to select diverse and informative samples. Earlier methods like similarity-based sampling in embedding space (Hardt & Sun, 2024) can also reduce data redundancy and emphasize representative examples. More broadly, Datamodels (Ilyas et al., 2022) analyze how individual examples shape model predictions, thus identifying the most beneficial training data. Unlike above these methods, our this work focuses specifically on post-pruning capability recovery, where the goal is not to select generally helpful instructions but to construct a capability-aware, degradation-sensitive data subset that targetedly restores pruned models.

## 3 METHODOLOGY

In this section, we first formulate the problem and then introduce three main components of the PASER framework (shown in Figure 1): semantic-structural recovery instruction clustering ($S^2$RIC), capability degradation-aware instruction selection (CDAIS), and negative tuning effects mitigation (NTEM). Furthermore, we provide the time complexity analysis for the PASER process.

### 3.1 PROBLEM FORMULATION

Let $M_o$ denote the original large language model and $M_p$ the pruned version of this model. We define the instruction tuning dataset as $D = \{(x_i, y_i)\}_{i=1}^N$, where $x_i$ represents an instruction and $y_i$ is its corresponding output. Our goal is to select a subset $S \subset D$ to efficiently recover the performance of $M_p$. We formulate the problem as an optimization task:

$$S^* = \underset{S \subset D, |S| \leq B}{\arg\min} \ \mathbb{E}_{(x,y) \sim \mathcal{T}}[\mathcal{L}(M_r(S), x, y)],$$
$$\text{s.t.} \quad M_r(S) = \text{RecoveryTrain}(M_p, S), \tag{1}$$

where $M_r(S)$ is the recovered model after training on subset $S$, $\mathcal{T}$ is the distribution of downstream evaluation tasks, $\mathcal{L}$ is a loss function. $B(B < N)$ is the recovery data budget, i.e., the maximum number of samples allowed in the selected subset.

## 3.2 SEMANTIC-STRUCTURAL RECOVERY INSTRUCTION CLUSTERING

After the LLM pruning, different model capabilities can be affected unevenly because such capabilities rely on distinct sets of internal modules (Meng et al., 2022; Dai et al., 2022), which are selectively removed. To ensure targeted and balanced recovery, we need to identify and group data points related to similar capabilities. To achieve this goal, we hypothesize that distinct geometric structures of recovery instruction data in the high-dimensional semantic space may correspond to different aspects of LLM capabilities. This hypothesis is based on the intuition that instructions pertaining similar capabilities are likely to cluster together in the semantic space, forming identifiable topological structures. Thus, we employ a two-step approach on the embedding space of instructions. First, to enhance the natural separation between capability-specific instruction clusters, we apply a diffusion kernel to SentenceBERT (Reimers & Gurevych, 2019) embeddings for manifold learning:

$$e(x_i) = \text{DiffusionKernel}(\text{SentenceBERT}(x_i)). \tag{2}$$

Here, $e(x_i)$ is the obtained low-dimensional manifold representation of instruction $x_i$. This process helps uncover the intrinsic geometric structure in the semantic space while reducing dimensionality and preserving non-linear relationships. Then, non-negative matrix factorization (NMF)-based spectral clustering (Ding et al., 2005) is conducted based on such $e(x_i)$ to identify natural groupings of instructions that potentially correspond to different LLM capabilities, leading to $K$ clusters:

$$\begin{aligned} C &= \{c_1, \dots, c_K\} \\ &= \text{NMFSpectralClustering}(\{e(x_i)|(x_i, y_i) \in D\}). \end{aligned} \tag{3}$$

The details are provided as below. In the first step of manifold learning, we first obtain the Sentence-BERT embedding of each instruction $x_i$. Then, an adjacency matrix $A$ is constructed based on the pairwise Euclidean distances of these embeddings: $A_{ij} = \exp(-\frac{\|\text{SentenceBERT}(x_i) - \text{SentenceBERT}(x_j)\|_2^2}{2\sigma^2})$, where $\sigma$ is a scaling parameter, typically set to the median of all pairwise distances. The degree matrix $D$ is then computed as a diagonal matrix where each diagonal element is the sum of the corresponding row in $A$: $D_{ii} = \sum_{j=1}^n A_{ij}$. Using these matrices, we define the normalized graph Laplacian $L = I - D^{-1/2}AD^{-1/2}$, where $I$ is the identity matrix. We then apply the diffusion kernel to this Laplacian $K_t = \exp(-tL)$, where $K_t$ is the diffusion kernel at time $t$. The diffusion time $t$ is selected using the spectral gap method: $t_{opt} = \arg\max_t \left(\frac{d\log(\lambda_2(t))}{d\log(t)}\right)$, where $\lambda_2(t)$ is the second eigenvalue of $K_t$. The low-dimensional manifold representation $e(x_i)$ is then obtained by selecting the top $d$ eigenvectors of $K_{t_{opt}}$: $e(x_i) = [\phi_1(x_i), \phi_2(x_i), ..., \phi_d(x_i)]$, where $\phi_j$ are the eigenvectors of $K_{t_{opt}}$ corresponding to the $d$ largest eigenvalues.

In the second step, we perform NMF-based spectral clustering on these low-dimensional representations. Specifically, we construct a similarity matrix $S$ from the manifold representations $S_{ij} = \exp(-\frac{\|e(x_i) - e(x_j)\|^2}{2\sigma^2})$. We then determine the optimal number of clusters $K$ by performing NMF with different values of $k$ and selecting the one that minimizes the Frobenius norm of the approximation error $K = \arg\min_k \|S - W_k H_k^T\|_F$, where $W_k$ and $H_k$ are non-negative matrices resulting from NMF with $k$ components. With this optimal $K$, we decompose the similarity matrix $S$ using NMF such that $S \approx WH^T$, where $W$ and $H$ are non-negative matrices with $K$ columns. Finally, we assign each data point to a cluster based on the maximum value in each row of $W$, where $c_i = \arg\max_j W_{ij}, i = 1, ..., N$. This results in $K$ clusters $C = \{c_1, \dots, c_K\}$, where the number of clusters $K$ is adaptively determined through the above process.

## 3.3 CAPABILITY DEGRADATION-AWARE INSTRUCTION SELECTION

**Capability Degradation Assessment** To prioritize the severely affected capabilities and finally achieve the balanced recovery of pruned LLMs, we need a measure of capability degradation to guide the data selection. For each cluster $c_k$ obtained in Section 3.2, we define the capability degradation score (CDS) with the Jensen-Shannon divergence (JSD) (Fuglede & Topsoe, 2004), which measures distributional differences between the original and pruned models' outputs as follows:

$$\frac{1}{|c_k|} \sum_{(x,y) \in c_k} \frac{1}{|y|} \sum_{m=1}^{|y|} \text{JSD}(P(\mathbf{t}_m|M_p; x, y_{<m}) || P(\mathbf{t}_m|M_o; x, y_{<m})). \tag{4}$$

Here, $P(\mathbf{t}_m|M_p; x, y_{<m})$ represents the output probability distribution on the $m$-th token variable of the pruned model $M_p$ given input $x$ and previous $m - 1$ tokens of output $y$. Taking a token value $t_m^i (1 \le i \le |\text{Voc}|)$ in this distribution as an example, its corresponding probability is

$$P(t_m^i|M_p; x, y_{<m}) = \frac{\exp(\frac{\text{logit}(t_m^i)}{\tau})}{\sum_{j=1}^{|\text{Voc}|} \exp(\frac{\text{logit}(t_m^j)}{\tau})}, \tag{5}$$

where $\tau$ is the softmax temperature and the $|\text{Voc}|$ indicates the vocabulary size. $\text{logit}(\cdot)$ is the logit value for tokens produced by LLM. Similarly, the $P(\mathbf{t}_m|M_o; x, y_{<m})$ represents the output probability distribution for the original model $M_o$. The JSD is actually the symmetrized and smoothed version of the Kullback–Leibler divergence (KLD) (Kullback & Leibler, 1951): $\text{JSD}(X||Y) = \frac{1}{2}\text{KLD}(X||M) + \frac{1}{2}\text{KLD}(Y||M)$. The distribution $M = \frac{1}{2}(X + Y)$ is the mixed distribution of $X$ and $Y$. Thus, the obtained CDS quantifies the average performance degradation for data points in each capability cluster. The choice of JSD over simple loss variations as the performance degradation signal is motivated by its unique properties. First, its symmetry ensures consistent comparison between the pruned model $M_p$ and the original model $M_o$, while its bounded range (0 to 1) provides a normalized measure of divergence. This facilitates easier interpretation and comparison across different capability clusters. Moreover, JSD's robustness to outliers and its information-theoretic foundation allow for a more nuanced assessment of capability degradation, capturing subtle changes in model outputs that might not be apparent from loss or accuracy values alone (Dutta et al., 2024) due to the sampling uncertainty. In fact, the output distribution reveals the model's thinking process more comprehensively than just sampling results, which is critical for capability degradation assessment. Besides, the smoothing effect introduced by the mixed distribution in JSD calculation also contributes to a more stable assessment across diverse instruction types. Thanks to these properties, we obtain a comprehensive and reliable assessment of capability degradation by employing JSD, enabling more accurate identification and prioritization of the capabilities most severely affected by model pruning.

**Inter-Capability Budget Allocation** Sampling a subset of high-quality data $S$ from the original set $D$ to achieve better training performance is the objective of the data selection process. Typically, an instruction data budget $B(B < N)$ should be maintained to ensure the efficiency on data utilization and training process. Under this budget, we design an adaptive selection approach based on the above CDS for balanced recovery while focusing on severely affected capabilities. In detail, we allocate the sampling budget to each cluster proportionally to its corresponding CDS:

$$n_k = \left\lfloor B \cdot \frac{\text{CDS}(c_k)}{\sum_{j=1}^K \text{CDS}(c_j)} \right\rfloor. \tag{6}$$

$n_k$ is the sample number budget allocated to cluster $c_k$.

**Intra-Capability Efficiency-Driven Sample Selection** To maximize computational efficiency during the recovery post-training phase, we need to select samples that offer the highest recovery benefit relative to their computational cost. Within each cluster $c_k$, we select the top $n_k$ samples based on their Individual Efficiency Scores (IES):

$$\text{IES}(x, y) = \frac{\frac{1}{|y|}\sum_{m=1}^{|y|} \text{JSD}(P(\mathbf{t}_m|M_p; x, y_{<m})||P(\mathbf{t}_m|M_o; x, y_{<m}))}{\log \text{ComputationalCost}(x, y)}. \tag{7}$$

Here, ComputationalCost is instantiated with the quadratic term of sequence length $(|x| + |y|)^2$ as the approximated complexity for LLM training. The use of JSD captures the degree of divergence between the pruned and original models' outputs, indicating areas where capabilities have been most affected and thus offering the highest potential for targeted recovery. The logarithmic term balances the consideration of computational cost, allowing for a more careful selection that favors efficient samples without overly penalizing high-potential, moderately costly instances.

### 3.4 NEGATIVE TUNING EFFECTS MITIGATION

To prevent performance degradation due to conflicting or irrelevant information, we need to detect and mitigate the potential negative tuning effects. We introduce a Concept Consistency Graph (CCG) to model relationships between concepts in the selected data. Here, a concept refers to a key knowledge unit or semantic element extracted from an instruction tuning sample. Concepts play a crucial role in

capturing the essential information within instructions and help to identify potential conflicts that could lead to a negative tuning effects. By managing relationships between concepts, we aim to maintain semantic consistency across the selected instruction tuning dataset, thereby reducing the risk of learning conflicting or irrelevant information during the recovery process. The formal definition for CCG is provided as follows:

**Definition 1** (Concept Consistency Graph). *A CCG is a graph $G = (V, E)$ where the vertices $V$ represent concepts, and an edge $(v_i, v_j) \in E$ exists if concepts $v_i$ and $v_j$ co-occur in at least one instruction tuning sample without conflict.*

For each new sample $(x, y)$, we first extract its concept $C(x, y)$ and then check for consistency: IsConsistent$(x, y) = \forall v_i, v_j \in C(x, y) : (v_i, v_j) \in E$ or $\{v_i, v_j\} \not\subset V$. We only add samples that are consistent with the existing CCG, updating the graph with each addition. This approach ensures that we maintain a coherent set of instructions, minimizing the risk of negative tuning effects by avoiding the introduction of conflicting concepts during the recovery training process. Furthermore, to mitigate the impact of rare nuanced conflicts that raw co-occurrence-based CCG may miss, we employ several lightweight enhancement schemes: (i) consolidating concept variants with simple semantic normalization (e.g., merging paraphrases and surface-form variants); (ii) treating low-confidence cases conservatively by softly down-weighting rather than hard-removing them; and (iii) providing an optional, low-budget Natural Language Inference reranker for only top-risk pairs flagged by CCG. The full version of our algorithm is provided in Appendix B.

### 3.5 TIME COMPLEXITY ANALYSIS

The theoretical analysis for PASER are as two folds: 1) time complexity analysis regarding efficiency; 2) error bound analysis regarding effectiveness. As the main bottleneck for real-world application, we first explore PASER's time complexity here.

**Theorem 1.** *The overall time complexity of PASER is $O(N \log N + NC^2)$, where $N$ is the number of instructions in $D$, and $C$ is the maximum number of concepts in any instruction tuning sample.*

*Proof.* We analyze each step of the algorithm in detail. The Semantic-Structural Recovery Instruction Clustering, including SentenceBERT embedding, Diffusion Kernel computation, and NMF Spectral Clustering, has a dominant complexity of $O(N \log N)$. For the Capability Degradation Assessment step, computing JSD for each sample and calculating CDS for each cluster take $O(N)$ time in total. The Inter-capability Budget Allocation, which involves allocating the budget to clusters, has a time complexity of $O(K)$, where $K$ is the number of clusters. However, since $K \leq N$, this step does not dominate the overall complexity. During Intra-capability Efficiency-Driven Sample Selection, for each cluster $c_k$, we perform sorting by JSD ($O(|c_k| \log |c_k|)$), iterate through sorted samples ($O(|c_k|)$), perform consistency checks (IsConsistent, $O(C^2)$ per sample), and update the CCG ($O(C^2)$ per sample). Considering all clusters, this step's total complexity is $O(N \log N + NC^2)$. Thus, the overall time complexity is dominated by the clustering step and the intra-capability sample selection step. Therefore, the total time complexity is $O(N \log N + NC^2)$.

In practice, $C$ is often much smaller than $N$ ($C \ll N$) and can be considered as a constant factor for large $N$. Thus, we can simplify the complexity to $O(N \log N)$. This analysis demonstrates that PASER is computationally efficient and scalable for large instruction tuning datasets. In addition, to guarantee the performance effectiveness, we also conduct a comprehensive **theoretical error bound analysis** for PASER in Appendix C.

## 4 EXPERIMENTS

### 4.1 EXPERIMENT SETUP

**Target LLMs** The experiments are mainly performed on several open-source popular English LLMs: LLaMA2-7B/13B/70B (Touvron et al., 2023) (hf version), LLaMA3-8B (Dubey et al., 2024)(instruct version), and bilingual LLMs: Baichuan2-7B/13B (Yang et al., 2023)(base version), which support both English and Chinese. Besides, several more recent LLMs are also adopted in Appendix J.

**Instruction Tuning Datasets** As for the original recovery post-training data, we choose two different-size instruction tuning datasets: Alpaca (Taori et al., 2023) and LaMini (Wu et al., 2024). Alpaca contains 52K instruction-following samples generated using OpenAI's *text-davinci-003* model based

Table 1: Recovery performance of different instruction tuning data selection methods under various pruning schemes on LLaMA2-7B model. The 'bold' represents the best performance under the same pruning scheme. Here, the Alpaca is taken as the original dataset.

| Pruning | Recovery Post-training | WikiText2↓ | PTB↓ | BoolQ | PIQA | HellaSwag | WinoGrande | ARC-e | ARC-c | OBQA | Average |
|---|---|---|---|---|---|---|---|---|---|---|---|
| w/o pruning | w/o Training | 12.62 | 22.14 | 71.13 | 78.40 | 72.79 | 67.17 | 69.36 | 40.70 | 40.80 | 62.91 |
| LLM-Pruner ratio=25% | w/o Training | 20.34 | 38.81 | 61.87 | 76.61 | 65.86 | 60.22 | 63.13 | 37.37 | 39.40 | 57.78 |
| | Full Data | 736.42 | 1273.10 | 37.83 | 53.21 | 26.42 | 49.57 | 25.29 | 28.16 | 29.00 | 35.64 |
| | Random | 93.77 | 180.62 | 57.61 | 64.37 | 45.39 | 55.87 | 43.78 | 31.94 | 34.90 | 47.69 |
| | Instruction Mining | 23.31 | 40.63 | 61.59 | 75.68 | 66.08 | 60.71 | 62.34 | 37.96 | 39.20 | 57.65 |
| | IFD | 19.76 | 33.30 | 63.55 | 77.23 | 67.21 | 60.90 | 63.46 | 37.81 | 40.00 | 58.59 |
| | Nuggets | 20.02 | 35.19 | 63.62 | **77.43** | 67.36 | 61.08 | 63.77 | 37.64 | **39.90** | 58.69 |
| | PASER | **16.40** | **26.35** | **67.25** | 77.29 | **68.98** | **66.97** | **67.84** | **39.54** | 39.80 | **61.10** |
| SliceGPT ratio=25% | w/o Training | 44.53 | 80.07 | 65.54 | 66.87 | 54.16 | 63.38 | 58.46 | 34.56 | 36.90 | 54.27 |
| | Full Data | 38.24 | 68.53 | 68.75 | 69.84 | 57.92 | 66.18 | 62.37 | 36.82 | 38.30 | 57.17 |
| | Random | 41.86 | 74.92 | 66.89 | 68.21 | 55.79 | 64.56 | 60.23 | 35.47 | 37.60 | 55.54 |
| | Instruction Mining | 39.75 | 71.28 | 67.87 | 68.93 | 56.42 | 65.76 | 61.89 | 36.23 | 37.60 | 56.39 |
| | IFD | 37.75 | 67.48 | 69.23 | 70.54 | 58.38 | 67.12 | 63.75 | 37.18 | 38.40 | 57.80 |
| | Nuggets | 23.86 | 35.42 | 69.89 | 71.21 | 58.79 | 67.56 | **72.23** | 37.47 | 38.60 | 59.39 |
| | PASER | **12.24** | **21.53** | **72.75** | **79.84** | **73.92** | **69.18** | 71.37 | **41.82** | **41.30** | **64.31** |
| Wanda sparsity=2:4 | w/o Training | 42.10 | 76.85 | 69.30 | 71.99 | 53.06 | 62.75 | 60.94 | 28.07 | 34.60 | 54.39 |
| | Full Data | 27.63 | 50.22 | 70.77 | 74.87 | 63.78 | 65.26 | 65.30 | 34.04 | 37.10 | 58.73 |
| | Random | 35.98 | 65.24 | 69.68 | 73.14 | 58.65 | 63.69 | 63.16 | 31.91 | 36.20 | 56.63 |
| | Instruction Mining | 31.47 | 57.17 | 70.61 | 73.85 | 61.27 | 64.13 | 64.72 | 33.79 | 36.80 | 57.88 |
| | IFD | 25.82 | 46.78 | 71.06 | 75.57 | 64.15 | 65.38 | 66.55 | 35.63 | 37.60 | 59.42 |
| | Nuggets | 23.98 | 43.24 | **71.68** | 76.14 | 64.65 | 65.69 | 66.16 | 36.91 | 38.20 | 59.92 |
| | PASER | **14.13** | **27.22** | 70.77 | **77.87** | **71.78** | **66.26** | **68.30** | **39.04** | **40.10** | **62.02** |
| SparseGPT sparsity=50% | w/o Training | 19.26 | 36.41 | 71.22 | 75.60 | 62.85 | 66.06 | 69.11 | 36.86 | 37.80 | 59.93 |
| | Full Data | 25.83 | 47.26 | 69.10 | 74.15 | 59.68 | 67.76 | 63.74 | **39.59** | 37.80 | 58.83 |
| | Random | 28.74 | 50.85 | 67.84 | 75.39 | 57.14 | 68.92 | 59.76 | 37.34 | 36.60 | 57.57 |
| | Instruction Mining | 24.08 | 45.51 | 70.50 | 74.47 | 61.91 | 65.40 | 67.73 | 36.49 | 37.40 | 59.13 |
| | IFD | 21.19 | 40.05 | 71.06 | 75.13 | 62.79 | 65.72 | 68.80 | 36.23 | 37.20 | 59.56 |
| | Nuggets | 16.21 | 28.95 | 71.64 | 75.67 | 63.33 | 66.05 | 69.49 | 36.60 | 37.40 | 60.03 |
| | PASER | **13.33** | **23.77** | **74.79** | **78.38** | **66.62** | **69.03** | **72.57** | 38.70 | **39.40** | **62.78** |

on 175 human-written seed tasks. LaMini contains a total of 2.58M pairs of instructions and responses synthesized with *gpt-3.5-turbo* based on several existing resources of prompts, including self-instruct (Wang et al., 2023), P3 (Sanh et al., 2022), FLAN (Longpre et al., 2023) and Alpaca (Taori et al., 2023). This dataset can help evaluate the effectiveness and efficiency in large-scale settings.

**Base Pruning Schemes** Different pruning schemes are incorporated into our experiments to explore the applicability of PASER, ranging from structured pruning methods: LLM-Pruner (Ma et al., 2023), SliceGPT (Ashkboos et al., 2024), semi-structured pruning method: Wanda (Sun et al., 2024), and unstructured pruning method: SparseGPT (Frantar & Alistarh, 2023).

**Instruction Tuning Data Selection Baselines** In addition to the random data selection, we also compare PASER with several recent general instruction tuning data selection baselines: Instruction Mining (Cao et al.), IFD (Li et al., 2024a), Nuggets (Li et al., 2024b). Note none of these baselines are customized for post-pruning recovery training scenario. Besides, the evaluation performance of recovery training with the full original dataset is also compared.

**Evaluation Datasets and Tasks** To thoroughly evaluate the performance of recovered LLMs, we employ seven common sense reasoning datasets:BoolQ (Clark et al., 2019), PIQA (Bisk et al., 2020), HellaSwag (Zellers et al., 2019), WinoGrande (Sakaguchi et al., 2021), ARC-Easy (Clark et al., 2018), ARC-Challenge (Clark et al., 2018), and OpenbookQA (Mihaylov et al., 2018). In the practice, we rely on the open-source library[1] to implement the evaluation, where the model needs to rank the choices in the multiple choice tasks or generate the answer in the open-ended generation tasks. The whole process is conducted in the *zero-shot* manner. Besides, we follow (Ma et al., 2023) to evaluate the language modeling capability with the zero-shot perplexity (PPL) analysis on WikiText2 (Merity et al., 2022) and PTB (Marcus et al., 1993). For enhancing evaluation comprehensiveness, we also conduct experiments on mathematical problem solving and code generation tasks, as shown in Appendix D and E, respectively.

## 4.2 EXPERIMENT RESULTS AND ANALYSIS

**Recovery Performance for Different Pruning Schemes** We evaluate the recovery performance of LLaMA2-7B using different instruction tuning data selection methods under structured pruning, semi-structured pruning, and unstructured pruning, respectively. According to the results in Table 1,

---

[1]https://github.com/EleutherAI/lm-evaluation-harness

Table 2: Recovery performance of different instruction tuning data selection methods on different target LLMs under LLM-Pruner scheme. The 'bold' represents the best performance on the same target LLM. Here, the Alpaca is taken as the original dataset. The "Reason" indicates the averaged performance on 7 common sense reasoning datasets.

| Model | Benchmark | w/o pruning | w/o Training | Full Data | Random | Instruction Mining | IFD | Nuggets | PASER |
|---|---|---|---|---|---|---|---|---|---|
| LLaMA2-13B ratio=50% | WikiText2↓ | 11.58 | 73.52 | 27.74 | 39.85 | 44.37 | 38.61 | 33.50 | **21.67** |
| | PTB↓ | 20.24 | 151.19 | 45.08 | 76.20 | 80.82 | 73.25 | 61.26 | **35.09** |
| | Reason↑ | 64.78 | 48.86 | 56.40 | 54.62 | 54.09 | 54.77 | 55.25 | **57.62** |
| LLaMA2-70B ratio=50% | WikiText2↓ | 8.92 | 46.81 | 31.76 | 25.34 | 23.16 | 22.87 | 19.63 | **12.35** |
| | PTB↓ | 15.59 | 92.36 | 56.83 | 48.72 | 43.45 | 43.68 | 36.24 | **21.82** |
| | Reason↑ | 71.72 | 61.14 | 65.56 | 64.03 | 66.74 | 67.27 | 67.73 | **69.62** |
| LLaMA3-8B ratio=25% | WikiText2↓ | 7.36 | 15.47 | 9.58 | 12.52 | 13.25 | 11.04 | 10.31 | **8.09** |
| | PTB↓ | 12.87 | 28.31 | 16.73 | 22.17 | 23.47 | 19.31 | 18.02 | **14.16** |
| | Reason↑ | 70.14 | 63.45 | 67.84 | 65.60 | 65.47 | 66.64 | 67.30 | **69.83** |
| Baichuan2-7B ratio=25% | WikiText2↓ | 14.42 | 28.30 | 25.29 | 27.04 | 34.24 | 24.83 | 21.48 | **16.92** |
| | PTB↓ | 26.78 | 53.34 | 35.81 | 46.83 | 60.93 | 37.81 | 37.65 | **30.76** |
| | Reason↑ | 64.19 | 56.33 | 57.39 | 57.09 | 54.78 | 57.36 | 57.84 | **59.70** |
| Baichuan2-13B ratio=50% | WikiText2↓ | 11.23 | 58.41 | 24.35 | 40.44 | 36.82 | 33.45 | 28.96 | **14.62** |
| | PTB↓ | 18.04 | 116.26 | 42.68 | 76.57 | 70.45 | 63.23 | 53.31 | **29.82** |
| | Reason↑ | 67.25 | 57.59 | 61.64 | 59.12 | 59.38 | 60.34 | 61.09 | **63.75** |

directly employing full data can indeed bring the sub-optimal recovery performance, especially under the LLM-Pruner. This is because the full version of data contains some irrelevant or conflicting information for capability recovery, resulting in the negative tuning effects during the training phase. Besides, considering pruning process has reduced the model's structural capacity, it can hardly satisfy all objectives present in the full dataset simultaneously. Training on full data forces the model to fit tasks beyond its representational bandwidth, which causes: overfitting to easy or dominant domains, forgetting of minority but critical capabilities, and instability in multi-objective optimization. Thus, performance becomes worse than simply leaving the pruned model untouched. Meanwhile, even the general instruction tuning data selection methods like IFD and Nuggets can bring better reasoning and language model performance than full data and random in most cases, validating the necessity of conducting recovery data selection. Furthermore, we can find that previous selection methods can hardly help model recover to the level of unpruned status, under the limited data budget. However, our PASER can not only outperform baselines, but also reduce the averaged reasoning performance degradation to less than 3% under LLM-Pruner, Wanda, and SparseGPT. Especially, when pruning LLaMA2-7B with SliceGPT, our PASER can improve the average reasoning performance to 64.31, higher than the unpruned model. Besides, its zero-shot perplexity on WikiText2 and PTB is also lower than unpruned model slightly. This suggests that allocating recovery budget according to capability degradation levels and prioritizing most-affected samples exhibit the potential of help pruned LLM recover to the capability level of unpruned status. Besides, PASER can also be extended to other LLM post-compression scenarios, like the post-quantization recovery and post-distillation recovery. The corresponding results and analysis are provided in Appendix G and H, respectively. Also, we have explored the recovery effects of different instruction tuning data selection methods on more knowledge-extensive common sense reasoning tasks in Appendix F. In Appendix K, we provided more detailed comparison between our PASER and general data selection methods, including both conceptual differentiation and empirical evidence.

**Robustness over Various Target Large Language Models** To validate whether PASER can maintain the robust effectiveness among various target LLMs, we conduct the experiments on LLaMA2-7B/13B/70B, LLaMA3-8B, Baichuan2-7B/13B, under LLM-Pruner. According to results in Table 2, we can first observe the model capability under high pruning ratio (50%) is hard to recover to unpruned level, especially for relatively smaller model like LLaMA2-13B and Baichuan2-13B. Though, PASER can still outperform random and best-performing data selection baseline, Nuggets, by 4.41 and 2.31 points, respectively on average. Especially, for LLaMA2-70B, our PASER can control the averaged reasoning performance degradation to less than 3%. This can be explained that the structure redundancy in 70B model is relatively higher, paving the way for effective recovery through data selection under high pruning ratios. As for the second smallest model, LLaMA3-8B, PASER can recover the reasoning performance to the 99.56% of the unpruned status, which further demonstrates the robustness of PASER over different target LLMs. Finally, the performance of various recovery methods including PASER on Baichuan2-7B is not satisfying enough given only 25% pruning ratio, which can be attributed to that the pruning process has severely damaged the model internal structure (e.g., attention heads specialized for corresponding capabilities). In such cases, recovery via post-training, even with carefully selected instructions, can be less effective

because the underlying model expressivity has already been impaired. Supplementary experiments on more recent LLMs like Qwen2.5 and Qwen3 can be seen in Appendix J.

**Recovery Performance with Different Instruction Tuning Datasets** In addition to the recovery performance on Alpaca shown in Table 1, we also explore the corresponding results on another larger dataset, LaMini. Especially, considering the space limitation and more severe performance degradation of structured pruning schemes, we provide the experiments results on LLM-Pruner and SliceGPT on Table 3. From this table, we can observe that PASER can still consistently outperform other data selection and random methods. Besides, comparing the results in Table 1 and 3, it can be found that improving the data scale (from 10K to 10K samples) indeed facilitates the recovery performance though the significantly increased computational overhead, which is consistent with the Scaling Law (Kaplan et al., 2020). We can also notice that the performance of full data on LaMini is relatively competitive, which is because the proportion of conflicting or negative data for recovery is much lower than that in Alpaca.

Table 3: Recovery performance of different instruction tuning data selection methods under two structured pruning schemes on LLaMA2-7B model. The 'bold' represents the best performance under the same pruning scheme. Here, the LaMini is taken as the original dataset. The "Reason" indicates the averaged performance on 7 reasoning datasets.

| Pruning | Recovery Post-training | WikiText2↓ | PTB↓ | Reason↑ |
|---|---|---|---|---|
| w/o pruning | w/o Training | 12.62 | 22.14 | 62.91 |
| LLM-Pruner ratio=25% | w/o Training | 20.34 | 38.81 | 57.78 |
| | Full Data | 16.28 | 27.12 | 62.68 |
| | Random | 18.40 | 32.15 | 60.93 |
| | Instruction Mining | 17.83 | 28.87 | 60.76 |
| | IFD | 18.54 | 31.23 | 60.65 |
| | Nuggets | 18.27 | 30.90 | 60.99 |
| | PASER | **13.45** | **22.63** | **63.79** |
| SliceGPT ratio=25% | w/o Training | 44.53 | 80.07 | 54.27 |
| | Full Data | 24.36 | 35.64 | 58.31 |
| | Random | 39.86 | 70.92 | 56.68 |
| | Instruction Mining | 37.75 | 67.28 | 57.53 |
| | IFD | 25.75 | 53.48 | 58.94 |
| | Nuggets | 21.86 | 31.42 | 60.96 |
| | PASER | **14.27** | **23.53** | **65.74** |

**Ablation Study** To validate the contribution of each component in PASER, we conduct comprehensive ablation studies. Specifically, we evaluate three variants: (1) PASER w/o S$^2$RIC: replacing semantic-structural clustering with random clustering while keeping other modules unchanged; (2) PASER w/o CDAIS: randomly sampling equal number of instructions from each cluster instead of using capability degradation-aware selection; (3) PASER w/o NTEM: removing the negative tuning effects mitigation module. The results in Table 4 demonstrate that all three components contribute positively to model recovery across different pruning schemes. The semantic-structural clustering effectively identifies capability-specific instruction groups, leading to 0.18-1.43 points improvement in reasoning performance. Its removal causes degradation in both language modeling (increased perplexity) and reasoning tasks, particularly evident under structured pruning schemes like LLM-Pruner and SliceGPT. The capability degradation-aware selection mechanism enhances recovery efficiency through adaptive budget allocation, contributing 0.29-1.63 points improvement in reasoning tasks while maintaining stable language modeling performance. Negative tuning effects mitigation shows significant impact (0.68-2.39 points improvement), especially under high pruning ratios, highlighting its importance in preventing conflicting information during recovery training. These improvements are consistently observed across different pruning schemes, with particularly pronounced effects in structured pruning where capability degradation tends to be more severe and uneven. More detailed ablation study results and analysis are provided in Appendix L. Besides, we also empirically explore the technique selection of PASER's each component in Appendix N.

**Recovery Post-training Efficiency Analysis** To highlight PASER's advantages on recovery post-training efficiency, we conduct the experiments under different data budgets $B$ and different datasets and record the corresponding averaged reasoning performance and training time in Figure 2. From the first and third subfigures, we can observe that PASER can obtain best recovery performance under different $B/N$ on Alpaca and LaMini. Interestingly, in the first subfigure, when rising $B/N$ from 0.3 to 0.4, the reasoning performance of Random even decreases. It is because expanding data scale also introduces the conflicting or negative data existing in the original dataset. From the second and fourth subfigures, PASER consistently consumes the least training time, which can be attributed to the efficiency-driven sample selection process in PASER. This advantage can be more obvious under low $B/N$ like 0.02 on LaMini. This is because increasing data budget will force PASER to select some relatively more time-consuming samples given the fixed original dataset, weakening its efficiency superiority. Besides, we also study the time consumption during data selection in Appendix O.

Table 4: Ablation study results on LLaMA2-7B for each component under different pruning schemes. The "Reason" indicates the averaged performance on 7 common sense reasoning datasets.

| Ablation Variant | LLM-Pruner (25%) | | | SliceGPT (25%) | | | Wanda (2:4) | | | SparseGPT (50%) | | |
|---|---|---|---|---|---|---|---|---|---|---|---|---|
| | WikiText2↓ | PTB↓ | Reason↑ | WikiText2↓ | PTB↓ | Reason↑ | WikiText2↓ | PTB↓ | Reason↑ | WikiText2↓ | PTB↓ | Reason↑ |
| w/o S²RIC | 18.73 | 32.84 | 59.67 | 14.83 | 25.42 | 63.03 | 15.84 | 30.25 | 61.19 | 14.89 | 26.31 | 62.60 |
| w/o CDAIS | 17.56 | 30.15 | 60.26 | 14.16 | 24.92 | 62.68 | 15.46 | 29.48 | 61.23 | 14.62 | 25.84 | 62.49 |
| w/o NTEM | 19.82 | 35.60 | 59.25 | 15.37 | 27.81 | 61.92 | 16.79 | 31.52 | 61.34 | 15.91 | 28.19 | 61.76 |
| PASER | 16.40 | 26.35 | 61.10 | 12.24 | 21.53 | 64.31 | 14.13 | 27.22 | 62.02 | 13.33 | 23.77 | 62.78 |

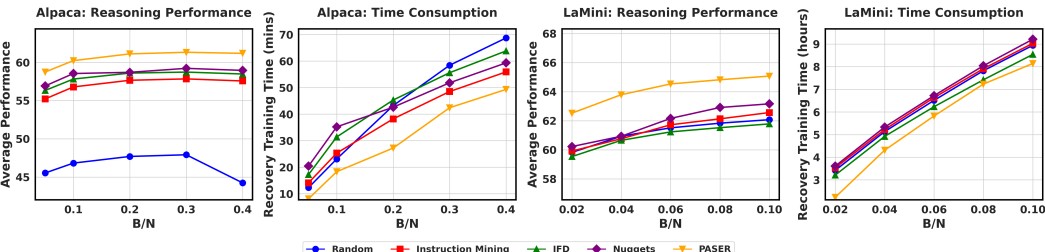

Figure 2: Average reasoning performance and recovery post-training time consumption curves corresponding to different instruction tuning data selection methods. The left two subfigures are for Alpaca while right two subfigures are for LaMini.

## 5 CONCLUSION AND FUTURE WORKS

Recovery post-training has been an important procedure after large language model pruning to restore the critical capabilities. Previous works directly utilize the full instruction tuning dataset, facing high computation cost, risks of untargeted recovery, and negative tuning effects. In this work, we propose the post-training data selection method for efficient pruned model recovery. According to capability degradation degrees, we allocate selection budget across different capability data obtained through semantic-structural clustering. We then select samples where model output behavior has been severely affected while considering computation cost, and introduce a concept consistency graph to mitigate negative tuning effects. Extensive experiments on different LLMs and theoretical analysis demonstrate the effectiveness of our framework. Future work will explore other optimization approaches like data augmentation and revision to further improve recovery efficiency.

## ETHICS STATEMENT

The development and deployment of technologies like PASER for efficient recovery of pruned large language models necessitates careful consideration of ethical implications. While PASER contributes to reducing environmental impact and potentially democratizing AI access by lowering computational requirements, it also raises concerns about potential misuse, bias amplification, and privacy. It's crucial to remain vigilant about these risks, implement robust safeguards, and maintain transparency in the recovery process. Continuous monitoring for fairness and bias in model outputs is essential, as is responsible deployment with appropriate human oversight, especially in high-stakes applications. As the field evolves, ongoing ethical assessment and dialogue with stakeholders are vital to ensure that advancements in large language model efficiency contribute positively to society while minimizing potential harm. Ultimately, the goal should be to harness the benefits of improved model recovery techniques like PASER while proactively addressing the complex ethical challenges they present.

## REPRODUCIBILITY STATEMENT

We provide a code repository `https://github.com/BokwaiHo/PASER` containing the implementation of PASER, along with detailed instructions for reproducing our experiments. All datasets used in this work, including Alpaca and LaMini for instruction tuning and standard reasoning/QA benchmarks (e.g., BoolQ, PIQA, ARC, HellaSwag, WinoGrande, OpenbookQA), are publicly available. Hyperparameters, pruning schemes, and evaluation protocols are described in the main text and appendices to ensure clarity. Our theoretical analysis, ablation studies, and efficiency measurements are also documented to facilitate independent verification

## ACKNOWLEDGMENTS

This work was supported by the Early Career Scheme (No. CityU 21219323) and the General Research Fund (No. CityU 11220324) of the University Grants Committee (UGC), the NSFC Young Scientists Fund (No. 9240127), and the Donation for Research Projects (No. 9229164 and No. 9220187).

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

CONTENTS

## A  SUPPORTING MATERIALS FOR INTRODUCTION PART

We provide the performance comparison in Figure 3 which supports the claim in the second paragraph of Section 1 about the recovery performance deterioration. From this figure, we can find that employing the full version of recovery data or uniformly split subset to conduct recovery training can hardly achieve satisfying performance.

Besides, we also provide the evidence for the uneven deterioration of different LLM capabilities during the pruning process (corresponding to the third paragraph in Section 1). From the Figure 4, we can observe that the four critical capabilities: language modeling, common sense reasoning, mathematical problem solving, and code generation exhibit significant difference on the performance degradation degrees. This phenomenon exits widely in the provided four pruning settings, which implies the necessity of performing targeted and balanced capability recovery. In fact, even among the various common sense reasoning tasks, this kind of uneven capability deterioration effect is still evident.

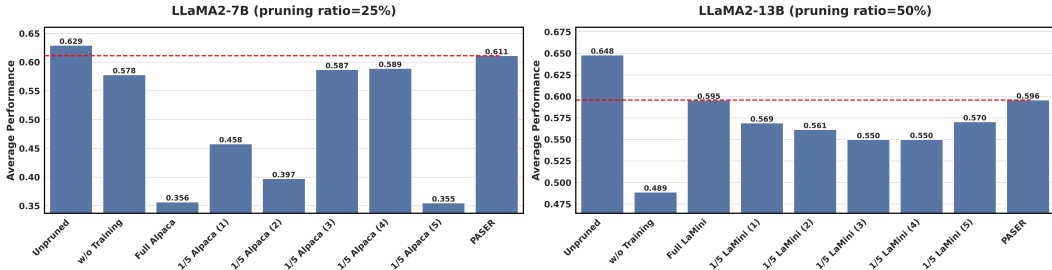

Figure 3: Average performance on seven common LLM reasoning evaluation tasks after recovery post-training with different data. The numbers in brackets represent the group index of the data subset in the full dataset. *Unpruned* indicates the original model and *w/o Training* indicates the pruned model (using LLM-Pruner (Ma et al., 2023)) without the recovery post-training.

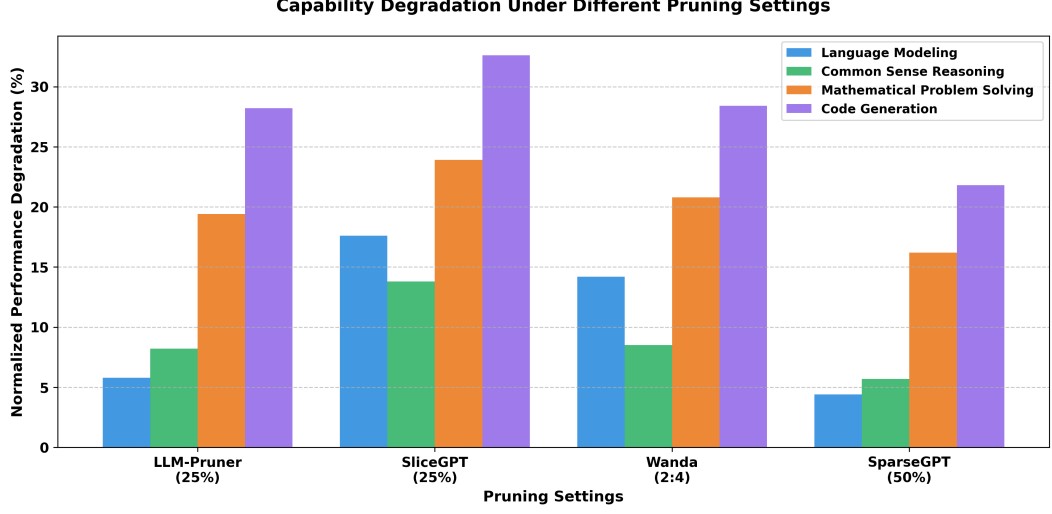

Figure 4: Normalized performance degradation degree(%) on four various capabilities under four LLM pruning settings.

## B  PSEUDOCODE

We provide the full version of our pseudocode in Algorithm 1.

---

**Algorithm 1** Post-training data Selection for efficient pruned large language model recovery (PASER)

---

**Input:** $M_o$: original model, $M_p$: pruned model, $D$: instruction tuning dataset, $B$: data budget, $U$: computational cost budget

  **procedure** PASER($M_o, M_p, D, B, U$)
    $C \leftarrow$ NMFSpectralClustering($\{e(x_i)|(x_i, y_i) \in D\}$)
    **for** $c_k \in C$ **do**
      CDS($c_k$) $\leftarrow$ ComputeCapabilityDegradationScore($c_k, M_o, M_p$)
    **end for**
    $\{n_k\} \leftarrow$ AllocateSamples($\{\text{CDS}(c_k)\}, B$)
    $S \leftarrow \emptyset, G \leftarrow$ InitializeCCG()
    **for** $c_k \in C$ **do**
      $L_k \leftarrow$ SortByIESDescending($c_k$)
      $i \leftarrow 0, \text{count} \leftarrow 0$
      **while** count $< n_k$ and $i < |L_k|$ **do**
        $(x, y) \leftarrow L_k[i]$
        **if** IsConsistent($x, y, G$) and $\sum_{(x',y') \in S \cup \{(x,y)\}}$ ComputationalCost($x', y'$) $\leq U$ **then**
          $S \leftarrow S \cup \{(x, y)\}$
          $G \leftarrow$ UpdateCCG($G, x, y$)
          count $\leftarrow$ count $+ 1$
        **end if**
        $i \leftarrow i + 1$
      **end while**
    **end for**
    **return** $S$
  **end procedure**

---

# C   Theoretical Error Bound Analysis

In this section, we establish theoretical guarantees for our PASER framework by deriving error bounds that characterize how close PASER's solution is to the optimal recovery solution.

## C.1   Problem Formulation Revisited

Recall from Section 3.1 that our objective is to find a subset $S^* \subset D$ of instruction tuning data that minimizes the expected loss on downstream tasks:

$$S^* = \arg \min_{S \subset D, |S| \leq B} \mathbb{E}_{(x,y) \sim \mathcal{T}}[\mathcal{L}(M_r(S), x, y)], \tag{8}$$

where $M_r(S)$ is the recovered model after training on subset $S$, $\mathcal{T}$ is the distribution of downstream evaluation tasks, and $\mathcal{L}$ is a loss function.

## C.2   Theoretical Framework

To establish theoretical guarantees, we introduce the following assumptions:

**Assumption 1** (Performance Decomposition). *The performance on downstream tasks can be decomposed as a weighted sum of performances across different capability clusters:*

$$\mathbb{E}_{(x,y) \sim \mathcal{T}}[L(M_r(S), x, y)] = \sum_{k=1}^{K} w_k \cdot \mathbb{E}_{(x,y) \sim c_k}[L(M_r(S), x, y)], \tag{9}$$

*where $c_k$ represents the $k$-th capability cluster, $w_k \geq 0$ is its weight in the evaluation distribution, and $\sum_{k=1}^{K} w_k = 1$.*

**Assumption 2** (Capability Independence). *The recovery effect on one capability depends primarily on samples from the corresponding cluster, with limited cross-cluster effects. For distinct clusters $c_i$ and $c_j$, and subsets $S_i \subset c_i$ and $S_j \subset c_j$:*

$$|\mathbb{E}_{(x,y) \sim c_i}[L(M_r(S_i \cup S_j), x, y)] - \mathbb{E}_{(x,y) \sim c_i}[L(M_r(S_i), x, y)]| \leq \delta_{ij}, \tag{10}$$

*where $\delta_{ij} \geq 0$ is a small constant representing the cross-cluster influence.*

**Assumption 3** (Conceptual Consistency Monotonicity). *Within a capability cluster, for conceptually consistent subsets, larger training subsets generally produce better recovery effects. Given a Concept Consistency Graph $G = (V, E)$, for any $S_A \subset S_B \subset c_k$ where all pairs of samples in $S_B$ are conceptually consistent according to $G$:*

$$\mathbb{E}_{(x,y)\sim c_k}[L(M_r(S_B), x, y)] \leq \mathbb{E}_{(x,y)\sim c_k}[L(M_r(S_A), x, y)]. \tag{11}$$

**Assumption 4** (Sample Selection Submodularity). *The recovery benefit exhibits approximate submodularity within each capability cluster. For conceptually consistent subsets $S \subset T \subset c_k$ and an element $e \in c_k \setminus T$ that is conceptually consistent with all elements in $T$:*

$$\Delta(e|S) \geq \Delta(e|T) - \alpha_k \cdot |T \setminus S|, \tag{12}$$

*where $\Delta(e|S) = \mathbb{E}_{(x,y)\sim c_k}[L(M_r(S), x, y)] - \mathbb{E}_{(x,y)\sim c_k}[L(M_r(S \cup \{e\}), x, y)]$ represents the marginal benefit of adding element $e$ to subset $S$, and $\alpha_k > 0$ is a cluster-specific constant.*

**Assumption 5** (Intra-Capability Selection Effectiveness). *For each capability cluster $c_k$, let $S_k$ be the subset of size $n_k$ selected by PASER and $S_k^*$ be the optimal subset of the same size. The intra-capability selection effectiveness of PASER is characterized by:*

$$\mathbb{E}_{(x,y)\sim c_k}[L(M_r(S_k), x, y)] \leq \mathbb{E}_{(x,y)\sim c_k}[L(M_r(S_k^*), x, y)] + (1 - \gamma_k) \cdot \Delta_k, \tag{13}$$

*where $\gamma_k \in (0, 1]$ and $\Delta_k = \mathbb{E}_{(x,y)\sim c_k}[L(M_r(\emptyset), x, y)] - \mathbb{E}_{(x,y)\sim c_k}[L(M_r(S_k^*), x, y)]$ represents the maximum possible improvement, which is guaranteed by above Assumption 4.*

**Assumption 6** (Inter-Capability Budget Allocation Effectiveness). *Given the limited cross-cluster influence established in Assumption 2, suboptimal budget allocation across capability clusters cannot catastrophically impact overall recovery performance. Therefore, the effectiveness of PASER's budget allocation across clusters can be bounded by:*

$$\sum_{k=1}^{K} w_k \cdot \mathbb{E}_{(x,y)\sim c_k}[L(M_r(S_k^*), x, y)] \leq \sum_{k=1}^{K} w_k \cdot \mathbb{E}_{(x,y)\sim c_k}[L(M_r(S_k^*(n_k^*)), x, y)] + (1-\beta) \cdot C_1, \tag{14}$$

*where $\beta \in (0, 1]$, $S_k^*$ is the optimal subset of size $n_k$ from cluster $c_k$, $S_k^*(n_k^*)$ is the optimal subset of size $n_k^*$ from cluster $c_k$, and $\{n_k^*\}_{k=1}^{K}$ is the optimal budget allocation across clusters under the same total budget $B$ ($\sum_{k=1}^{K} n_k = \sum_{k=1}^{K} n_k^* = B$). Thus, $S_k^*(n_k^*) = S^* \cap c_k$.*

Based on these assumptions, we establish the following theorem:

**Theorem 2** (PASER Error Bound). *Let $S_{PASER}$ be the subset selected by PASER with budget $B$, $S^*$ be the optimal subset of the same size, and $\gamma = \min_k \gamma_k$. Then there exists a constant $C > 0$ such that:*

$$\mathbb{E}_{(x,y)\sim \mathcal{T}}[L(M_r(S_{PASER}), x, y)] \leq \mathbb{E}_{(x,y)\sim \mathcal{T}}[L(M_r(S^*), x, y)] + (1 - \beta\gamma) \cdot C. \tag{15}$$

### C.3 DETAILED DERIVATION

We now provide a step-by-step derivation of the error bound, explicitly showing how each assumption is utilized.

**Step 1:** Decompose the performance into capability clusters using Assumption 1:

$$\mathbb{E}_{(x,y)\sim \mathcal{T}}[L(M_r(S_{PASER}), x, y)] = \sum_{k=1}^{K} w_k \cdot \mathbb{E}_{(x,y)\sim c_k}[L(M_r(S_{PASER}), x, y)] \tag{16}$$

$$\mathbb{E}_{(x,y)\sim \mathcal{T}}[L(M_r(S^*), x, y)] = \sum_{k=1}^{K} w_k \cdot \mathbb{E}_{(x,y)\sim c_k}[L(M_r(S^*), x, y)]. \tag{17}$$

**Step 2:** Apply Assumption 2 (Capability Independence) to isolate the effect of samples from each cluster:

$$\mathbb{E}_{(x,y)\sim c_k}[L(M_r(S_{PASER}), x, y)] \leq \mathbb{E}_{(x,y)\sim c_k}[L(M_r(S_{PASER} \cap c_k), x, y)] + \sum_{j\neq k} \delta_{kj} \tag{18}$$

$$= \mathbb{E}_{(x,y)\sim c_k}[L(M_r(S_k), x, y)] + \sum_{j\neq k} \delta_{kj}, \tag{19}$$

where $S_k = S_{PASER} \cap c_k$ is the subset selected by PASER from cluster $c_k$.

Similarly for the optimal subset:

$$\mathbb{E}_{(x,y)\sim c_k}[L(M_r(S^*), x, y)] \geq \mathbb{E}_{(x,y)\sim c_k}[L(M_r(S^* \cap c_k), x, y)] - \sum_{j \neq k} \delta_{kj}. \tag{20}$$

**Step 3:** Now we analyze the relationship between PASER's selection $S_k$ and the optimal selection $S_k^*$ within cluster $c_k$.

First, we utilize Assumption 3 (Conceptual Consistency Monotonicity). PASER's Concept Consistency Graph ensures that all selected samples in $S_k$ are conceptually consistent. For any subset $S_k^{CCG} \subset S_k$ that is conceptually consistent:

$$\mathbb{E}_{(x,y)\sim c_k}[L(M_r(S_k), x, y)] \leq \mathbb{E}_{(x,y)\sim c_k}[L(M_r(S_k^{CCG}), x, y)]. \tag{21}$$

Now, we apply Assumption 4 (Sample Selection Submodularity) to analyze the greedy selection process employed by PASER within each cluster. For a conceptually consistent set, the greedy algorithm achieves an approximation ratio that depends on the degree of submodularity.

Let $S_k^G$ be the set obtained by a greedy selection process that maximizes the marginal gain at each step while ensuring conceptual consistency. From the properties of approximated submodular functions (Assumption 4) and the guarantees of the greedy algorithm for such functions:

$$\mathbb{E}_{(x,y)\sim c_k}[L(M_r(S_k^G), x, y)] \leq \mathbb{E}_{(x,y)\sim c_k}[L(M_r(S_k^*), x, y)] + \eta_k \cdot \Delta_k, \tag{22}$$

where $\eta_k \in [0, 1)$ is a constant that depends on the submodularity gap parameter $\alpha_k$ and the size of the selection $n_k$, and $\Delta_k$ is as defined in Assumption 5 (Intra-Capability Selection Effectiveness).

PASER's efficiency-driven selection further refines this greedy selection by prioritizing samples with higher Individual Efficiency Scores. This introduces an additional approximation factor $\phi_k \in [0, 1)$:

$$\mathbb{E}_{(x,y)\sim c_k}[L(M_r(S_k), x, y)] \leq \mathbb{E}_{(x,y)\sim c_k}[L(M_r(S_k^G), x, y)] + \phi_k \cdot \Delta_k. \tag{23}$$

Combining these equations, we get:

$$\mathbb{E}_{(x,y)\sim c_k}[L(M_r(S_k), x, y)] \leq \mathbb{E}_{(x,y)\sim c_k}[L(M_r(S_k^*), x, y)] + (\eta_k + \phi_k) \cdot \Delta_k. \tag{24}$$

Setting $(1 - \gamma_k) = (\eta_k + \phi_k)$, we arrive at the form presented in Assumption 5:

$$\mathbb{E}_{(x,y)\sim c_k}[L(M_r(S_k), x, y)] \leq \mathbb{E}_{(x,y)\sim c_k}[L(M_r(S_k^*), x, y)] + (1 - \gamma_k) \cdot \Delta_k. \tag{25}$$

This explicitly shows how $\gamma_k$ incorporates both the submodularity properties (Assumption 4) and the conceptual consistency properties (Assumption 3) into our error bound.

**Step 4:** Combine Steps 2 and 3:

$$\mathbb{E}_{(x,y)\sim c_k}[L(M_r(S_{PASER}), x, y)] \leq \mathbb{E}_{(x,y)\sim c_k}[L(M_r(S_k), x, y)] + \sum_{j \neq k} \delta_{kj} \tag{26}$$

$$\leq \mathbb{E}_{(x,y)\sim c_k}[L(M_r(S_k^*), x, y)] + (1 - \gamma_k) \cdot \Delta_k + \sum_{j \neq k} \delta_{kj}. \tag{27}$$

**Step 5:** Apply Assumption 5 (Inter-Capability Budget Allocation Optimality) to compare PASER's allocation with the optimal allocation:

$$\sum_{k=1}^{K} w_k \cdot \mathbb{E}_{(x,y)\sim c_k}[L(M_r(S_k^*), x, y)] \leq \sum_{k=1}^{K} w_k \cdot \mathbb{E}_{(x,y)\sim c_k}[L(M_r(S_k^*(n_k^*)), x, y)] + (1 - \beta) \cdot C_1, \tag{28}$$

where $C_1$ is a constant related to the maximum performance difference between different budget allocations. The factor $\beta$ quantifies how well PASER's capability degradation-aware budget allocation approximates the theoretical optimal allocation.

**Step 6:** Combine Steps 1-5 and let $\gamma = \min_k \gamma_k$:

$$\mathbb{E}_{(x,y)\sim\mathcal{T}}[L(M_r(S_{PASER}), x, y)] = \sum_{k=1}^{K} w_k \cdot \mathbb{E}_{(x,y)\sim c_k}[L(M_r(S_{PASER}), x, y)] \tag{29}$$

$$\leq \sum_{k=1}^{K} w_k \cdot \left( \mathbb{E}_{(x,y)\sim c_k}[L(M_r(S_k^*), x, y)] + (1-\gamma_k)\cdot\Delta_k + \sum_{j\neq k}\delta_{kj} \right) \tag{30}$$

$$\leq \sum_{k=1}^{K} w_k \cdot \mathbb{E}_{(x,y)\sim c_k}[L(M_r(S_k^*), x, y)] + (1-\gamma)\cdot\sum_{k=1}^{K} w_k \cdot \Delta_k + \sum_{k=1}^{K} w_k \sum_{j\neq k}\delta_{kj} \tag{31}$$

$$\leq \sum_{k=1}^{K} w_k \cdot \mathbb{E}_{(x,y)\sim c_k}[L(M_r(S_k^*(n_k^*)), x, y)] + (1-\beta)\cdot C_1 + (1-\gamma)\cdot C_2 + C_3 \tag{32}$$

$$= \sum_{k=1}^{K} w_k \cdot \mathbb{E}_{(x,y)\sim c_k}[L(M_r(S^* \cap c_k), x, y)] + (1-\beta)\cdot C_1 + (1-\gamma)\cdot C_2 + C_3 \tag{33}$$

$$\leq \mathbb{E}_{(x,y)\sim\mathcal{T}}[L(M_r(S^*), x, y)] + 2C_3 + (1-\beta)\cdot C_1 + (1-\gamma)\cdot C_2 \tag{34}$$

$$\leq \mathbb{E}_{(x,y)\sim\mathcal{T}}[L(M_r(S^*), x, y)] + 2C_3 + (1-\beta\gamma)\cdot(C_1 + C_2), \tag{35}$$

where:

- $C_2 = \sum_{k=1}^{K} w_k \cdot \Delta_k$ represents the maximum possible improvement across all clusters,
- $C_3 = \sum_{k=1}^{K} w_k \sum_{j\neq k} \delta_{kj}$ represents the total cross-cluster effects.

**Step 7:** Let $C = C_1 + C_2$, considering $\delta_{kj} \to 0$ as claimed in the Assumption 2,

$$\mathbb{E}_{(x,y)\sim\mathcal{T}}[L(M_r(S_{PASER}), x, y)] \leq \mathbb{E}_{(x,y)\sim\mathcal{T}}[L(M_r(S^*), x, y)] + (1-\beta\gamma)\cdot C. \tag{36}$$

### C.4 INTERPRETATION OF THE ERROR BOUND

The error bound provides several key insights, explicitly highlighting the role of each assumption:

**Performance Decomposition (Assumption 1):** This foundational assumption enables modular analysis by decomposing model performance into distinct capabilities, allowing PASER to target specific degraded capabilities with focused recovery resources.

**Capability Independence (Assumption 2):** By establishing bounded cross-capability influence, this assumption validates PASER's independent optimization within each capability cluster, ensuring that improvements in one area don't significantly interfere with others.

**Conceptual Consistency Monotonicity (Assumption 3):** By ensuring monotonicity within conceptually consistent subsets, PASER's CCG mechanism effectively mitigates negative tuning effects, making the selection problem more tractable.

**Sample Selection Submodularity (Assumption 4):** The approximated submodularity enables efficient greedy-like selection methods to achieve near-optimal results within each capability cluster.

**Intra-Capability Selection Effectiveness (Assumption 5):** The parameter $\gamma$ reflects PASER's efficiency in selecting the most valuable samples within each capability cluster. When $\gamma$ approaches 1, PASER's intra-cluster selection approaches optimality.

**Inter-Capability Budget Allocation Effectiveness (Assumption 6):** The parameter $\beta$ reflects PASER's efficiency in allocating the budget across different capability clusters. When $\beta$ approaches 1, PASER's budget allocation approaches the theoretical optimal allocation.

**Combined Effect:** The error term $(1-\beta\gamma)$ indicates that when both selection effectiveness and allocation effectiveness are high, PASER's performance approaches the global optimal solution.

This theoretical framework explains PASER's strong empirical performance: by intelligently allocating budget based on capability degradation degree (improving $\beta$) and selecting the most valuable samples within each cluster while maintaining concept consistency (improving $\gamma$), PASER achieves near-optimal recovery performance.

Table 5: Recovery performance of different instruction tuning data selection methods on mathematical problem solving tasks under various pruning schemes. The 'bold' represents the best performance under the same pruning scheme.

| Recovery Method | LLM-Pruner (25%) | | SliceGPT (25%) | | Wanda (2:4) | | SparseGPT (50%) | |
|---|---|---|---|---|---|---|---|---|
| | GSM8K | Minerva | GSM8K | Minerva | GSM8K | Minerva | GSM8K | Minerva |
| w/o Training | 44.3 | 17.8 | 42.5 | 16.9 | 43.8 | 17.4 | 43.1 | 17.2 |
| Full Data | 46.5 | 19.1 | 44.8 | 18.3 | 45.9 | 18.7 | 45.2 | 18.5 |
| Random | 45.8 | 18.4 | 43.9 | 17.8 | 44.7 | 18.1 | 44.3 | 17.9 |
| Instruction Mining | 46.2 | 18.9 | 44.5 | 18.1 | 45.4 | 18.5 | 44.9 | 18.3 |
| IFD | 46.8 | 19.3 | 45.1 | 18.5 | 45.8 | 18.8 | 45.4 | 18.6 |
| Nuggets | 47.1 | 19.5 | 45.4 | 18.7 | 46.2 | 19.0 | 45.7 | 18.8 |
| PASER | **49.4** | **21.2** | **47.8** | **20.5** | **48.5** | **20.8** | **47.2** | **20.1** |

## D    EVALUATION ON MATHEMATICAL PROBLEM SOLVING TASKS

To validate the effectiveness of PASER beyond common sense reasoning tasks, we conduct additional experiments on mathematical problem solving capabilities. Specifically, we employ two widely-adopted mathematical problem solving benchmarks:

- **GSM8K** (Cobbe et al., 2021): A dataset containing 8.5K high-quality grade school math word problems that test various mathematical problem solving capabilities, including arithmetic, algebra, and word problem solving.

- **Minerva Math** (Lewkowycz et al., 2022): A comprehensive mathematical evaluation dataset covering diverse topics in mathematics ranging from arithmetic to calculus, with problems requiring multi-step reasoning.

The recovery performance under different pruning schemes is presented in Table 5. From these results, we can observe that PASER consistently outperforms baseline methods across all pruning schemes on both mathematical problem solving benchmarks. The improvements are particularly significant under the LLM-Pruner scheme, where PASER achieves 5.1% and 3.4% absolute improvements over w/o Training on GSM8K and Minerva Math, respectively. While different pruning schemes affect the base performance levels, PASER maintains its effectiveness in recovery. For example, under the more aggressive SparseGPT (50%) setting, PASER still achieves 4.1% and 2.9% improvements on GSM8K and Minerva Math over w/o Training. Compared to Full Data training, PASER achieves better performance while using only 20% of the instruction data, demonstrating its efficiency in recovering mathematical problem solving capabilities.

These results, combined with the common sense reasoning results presented in the main paper, demonstrate that PASER is effective across diverse tasks. The strong performance on mathematical tasks is particularly noteworthy given that these problems often require precise, step-by-step reasoning and have less tolerance for errors compared to common sense reasoning tasks. This validates the effectiveness of our capability degradation score in identifying and prioritizing recovery for severely affected capabilities, even in domains requiring high precision.

Table 6: Recovery performance of different instruction tuning data selection methods on code generation tasks under various pruning schemes. The 'bold' represents the best performance under the same pruning scheme. 'P@$k$' indicates 'Pass@$k$'.

| Recovery Method | LLM-Pruner (25%) | | | | | | SliceGPT (25%) | | | | | |
|---|---|---|---|---|---|---|---|---|---|---|---|---|
| | HumanEval | | | MBPP | | | HumanEval | | | MBPP | | |
| | P@1 | P@10 | P@100 | P@1 | P@10 | P@100 | P@1 | P@10 | P@100 | P@1 | P@10 | P@100 |
| w/o Training | 3.4 | 6.2 | 10.4 | 7.1 | 15.5 | 24.6 | 3.0 | 5.9 | 8.7 | 6.2 | 11.8 | 21.5 |
| Full Data | 7.8 | 15.1 | 19.0 | 15.2 | 26.3 | 39.5 | 2.9 | 5.1 | 11.8 | 9.3 | 19.7 | 36.4 |
| Random | 6.2 | 13.7 | 20.5 | 12.8 | 23.8 | 35.0 | 3.1 | 5.8 | 14.2 | 8.5 | 17.2 | 38.6 |
| Instruction Mining | 8.9 | 17.8 | 28.4 | 15.7 | 29.2 | 43.1 | 6.3 | 11.4 | 23.8 | 12.8 | 24.5 | 45.2 |
| IFD | 10.5 | 21.2 | 35.6 | 18.2 | 34.5 | 50.7 | 8.7 | 16.8 | 31.2 | 16.4 | 30.8 | 52.4 |
| Nuggets | 11.8 | 22.9 | 38.3 | 18.9 | 35.8 | 52.4 | 9.5 | 18.5 | 33.9 | 17.6 | 32.6 | 51.9 |
| PASER | **14.4** | **27.6** | **48.2** | **23.1** | **42.6** | **62.0** | **12.9** | **25.2** | **44.5** | **22.3** | **41.0** | **63.7** |

# E    EVALUATION ON CODE GENERATION TASKS

To further explore the PASER's effectiveness on recovering code generation capability, we take two structured pruning schemes (LLM-Pruner, SliceGPT) and perform exhaustive evaluations on two major code generation benchmarks:

- **HumanEval** (Chen et al., 2021): A widely-used code generation benchmark consisting of 164 hand-crafted Python programming problems that test various programming concepts. Each problem contains a function signature with docstring and test cases, requiring models to complete the implementation. The benchmark evaluates functional correctness using the Pass@$k$ metric, which measures the percentage of problems where a correct solution appears in k samples.

- **MBPP** (Austin et al., 2021): A programming benchmark containing 974 Python programming problems focused on basic programming tasks. Each problem includes a natural language description, test cases, and a reference solution, making it particularly suitable for evaluating language-to-code generation capabilities. MBPP uses the same Pass@$k$ evaluation metric as HumanEval but generally features simpler problems with a broader coverage of basic programming concepts.

In our experiments, models are evaluated in zero-shot on HumanEval and 3-shot on MBPP. The results under Pass@$k$ ($k = 1, 10, 100$) metrics are present in Table 6. As shown in the table, code generation capability experiences severe degradation after pruning. The Pass@1 performance on HumanEval drops to merely 3.4% under LLM-Pruner without recovery training. This dramatic decline indicates that code generation, as a complex reasoning task, is particularly vulnerable during model compression. Through capability degradation-aware budget allocation and targeted sample selection, PASER demonstrates remarkable effectiveness in recovering this severely impacted capability. Under LLM-Pruner, it achieves 14.4% Pass@1 on HumanEval, not only substantially outperforming other recovery methods but also surpassing the full data training baseline. The improvement becomes even more pronounced at higher sampling rates, i.e., PASER reaches 48.2% Pass@100 compared to Random's 20.5% and Instruction Mining's 28.4%. This significant performance gap validates our approach of prioritizing recovery resources for severely degraded capabilities and selecting the most relevant instruction samples for recovery training. The superiority of PASER remains consistent across different evaluation settings. On MBPP, which features simpler programming tasks, PASER still maintains a clear advantage over baseline methods, achieving 23.1% Pass@1 and 62.0% Pass@100 under LLM-Pruner. When tested with a different pruning scheme (SliceGPT), which causes even more severe initial degradation (3.0% Pass@1 on HumanEval), PASER successfully recovers the performance to 12.9% Pass@1 and 44.5% Pass@100.

These results comprehensively demonstrate that our capability-aware recovery strategy effectively addresses the disproportionate impact of model compression on complex reasoning abilities, enabling targeted and efficient recovery of critical model capabilities.

Table 7: Recovery performance of different instruction tuning data selection methods on other knowledge extensive tasks under different LLM pruning schemes. The 'bold' represents the best performance under the same pruning scheme. Here, the Alpaca is taken as the original dataset. The LLaMA2-7B is taken as the target here to ensure consistency.

| Benchmark | Pruning | w/o Training | Full Data | Random | Instruction Mining | IFD | Nuggets | PASER |
|---|---|---|---|---|---|---|---|---|
| MMLU | LLM-Pruner (25%) ↓ | 25.35 | 36.27 | 40.91 | 41.42 | 41.86 | 42.30 | **46.37** |
| | SliceGPT (25%) | 28.80 | 40.82 | 41.76 | 42.15 | 42.51 | 42.64 | **46.97** |
| | Wanda (2:4) | 32.58 | 44.45 | 42.36 | 42.53 | 42.73 | 43.55 | **47.04** |
| | SparseGPT (50%) | 34.13 | 45.01 | 42.47 | 44.19 | 43.63 | 43.18 | **47.43** |
| CommonsenseQA | LLM-Pruner (25%) ↓ | 48.47 | 57.50 | 60.21 | 60.65 | 61.22 | 62.59 | **66.44** |
| | SliceGPT (25%) | 49.60 | 61.32 | 61.09 | 60.81 | 62.28 | 62.23 | **66.51** |
| | Wanda (2:4) | 52.65 | 65.04 | 61.71 | 62.13 | 62.47 | 62.91 | **66.79** |
| | SparseGPT (50%) | 53.97 | 64.32 | 62.21 | 63.04 | 62.48 | 63.17 | **67.63** |

# F    EVALUATION ON OTHER KNOWLEDGE-EXTENSIVE TASKS

To supplement more experiments on other knowledge-extensive evaluation tasks, we conduct experiments on MMLU (Hendrycks et al.) and CommonsenseQA (Talmor et al., 2019), with LLaMA2-7B as the target LLM to ensure comparison consistency. As shown in the Table 7, PASER consistently outperforms existing data selection baselines and even surpasses full-data tuning. This demonstrates

that PASER generalizes well beyond the commonsense reasoning benchmarks in the main text and is effective on other knowledge-intensive tasks.

Table 8: Recovery performance of different instruction tuning data selection methods under RTN 4bits and GPTQ 4bits schemes on LLaMA2-7B model. The 'bold' represents the best performance under the same quantization scheme. Here, the Alpaca is taken as the original dataset.

| Quantization | Recovery Post-training | WikiText2↓ | PTB↓ | BoolQ | PIQA | HellaSwag | WinoGrande | ARC-e | ARC-c | OBQA | Average |
|---|---|---|---|---|---|---|---|---|---|---|---|
| w/o Quant | w/o training | 12.62 | 22.14 | 71.13 | 78.40 | 72.79 | 67.17 | 69.36 | 40.70 | 40.80 | 62.91 |
| RTN 4 bits | w/o training | 18.14 | 33.28 | 66.52 | 74.95 | 69.24 | 63.91 | 65.58 | 38.07 | 35.10 | 59.05 |
| | Full Data | 15.83 | 27.41 | 67.35 | 75.70 | 69.94 | 64.57 | 66.22 | 38.48 | 35.90 | 59.74 |
| | Random | 16.72 | 29.56 | 64.53 | 73.08 | 67.48 | 62.28 | 63.93 | 37.13 | 33.90 | 57.48 |
| | Instruction Mining | 16.05 | 27.83 | 66.73 | 75.15 | 69.43 | 64.08 | 65.74 | 38.18 | 35.30 | 59.23 |
| | IFD | 15.21 | 25.74 | 68.16 | 76.40 | 70.60 | 65.18 | 66.83 | 38.83 | 37.40 | 60.49 |
| | Nuggets | 14.68 | 24.53 | 68.99 | 77.13 | 71.28 | 65.82 | 67.46 | 39.21 | 38.70 | 61.23 |
| | PASER | **14.21** | **23.37** | **70.43** | **78.41** | **72.47** | **66.92** | **68.54** | **39.81** | **41.50** | **62.58** |
| GPTQ 4 bits | w/o training | 15.96 | 27.86 | 67.82 | 76.15 | 70.35 | 64.95 | 66.59 | 38.69 | 36.90 | 60.21 |
| | Full Data | 15.62 | 26.95 | 68.00 | 76.31 | 70.50 | 65.09 | 66.73 | 38.78 | 37.40 | 60.40 |
| | Random | 16.31 | 28.74 | 66.81 | 75.24 | 69.49 | 64.14 | 65.79 | 38.22 | 35.70 | 59.34 |
| | Instruction Mining | 15.37 | 26.42 | 68.31 | 76.58 | 70.75 | 65.33 | 66.96 | 38.93 | 37.90 | 60.68 |
| | IFD | 14.83 | 25.16 | 68.96 | 77.15 | 71.29 | 65.83 | 67.47 | **40.21** | 39.00 | 61.42 |
| | Nuggets | 13.52 | 22.93 | 69.74 | 77.83 | 71.93 | 66.43 | 68.06 | 39.56 | 40.20 | 61.96 |
| | PASER | **12.95** | **21.84** | **71.20** | **79.12** | **73.12** | **67.53** | **69.14** | 40.18 | **42.90** | **63.31** |

Table 9: Recovery performance of different recovery data selection methods under RTN 4bits and GPTQ 4bits quantization schemes on LLaMA2-13B model. The 'bold' represents the best performance under the same quantization scheme. Here, the Alpaca is taken as the original dataset.

| Quantization | Recovery Post-training | WikiText2↓ | PTB↓ | BoolQ | PIQA | HellaSwag | WinoGrande | ARC-e | ARC-c | OBQA | Average |
|---|---|---|---|---|---|---|---|---|---|---|---|
| w/o Quant | w/o training | 11.58 | 20.24 | 69.02 | 78.73 | 76.60 | 69.69 | 73.23 | 44.20 | 42.00 | 64.78 |
| RTN 4 bits | w/o training | 17.53 | 32.34 | 63.15 | 74.59 | 72.62 | 65.94 | 69.17 | 41.49 | 37.00 | 60.57 |
| | Full Data | 16.95 | 31.02 | 63.59 | 75.02 | 73.04 | 66.33 | 69.58 | 41.75 | 37.50 | 60.97 |
| | Random | 17.86 | 33.15 | 62.00 | 73.48 | 71.55 | 64.94 | 68.13 | 40.84 | 35.20 | 59.45 |
| | Instruction Mining | 17.24 | 31.68 | 62.83 | 74.27 | 72.32 | 65.65 | 68.87 | 41.29 | 36.10 | 60.19 |
| | IFD | 15.63 | 28.39 | 65.03 | 76.37 | 74.34 | 67.49 | 70.80 | 42.46 | 39.60 | 62.30 |
| | Nuggets | 15.08 | 27.15 | 65.45 | 76.76 | **76.72** | 67.84 | 71.17 | 42.70 | 40.20 | 62.98 |
| | PASER | **12.34** | **23.08** | **67.33** | **78.50** | 76.41 | **69.37** | **72.78** | **43.67** | **41.70** | **64.25** |
| GPTQ 4 bits | w/o training | 14.74 | 26.86 | 64.68 | 76.04 | 74.02 | 67.20 | 70.49 | 42.28 | 39.10 | 61.97 |
| | Full Data | 16.02 | 29.34 | 63.62 | 75.05 | 73.07 | 66.35 | 69.61 | 41.76 | 37.40 | 60.98 |
| | Random | 14.58 | 26.52 | 64.82 | 76.17 | 74.15 | 67.32 | 70.61 | 42.36 | 39.30 | 62.10 |
| | Instruction Mining | 13.67 | 24.59 | 66.37 | 77.58 | 75.51 | 68.56 | 71.91 | **44.15** | 41.30 | 63.63 |
| | IFD | 13.51 | 24.26 | **68.46** | 77.66 | 75.59 | 68.63 | 71.99 | 43.20 | 41.40 | 63.85 |
| | Nuggets | 12.76 | 22.92 | 67.21 | 78.34 | 76.25 | 69.24 | 72.63 | 43.59 | 42.70 | 64.28 |
| | PASER | **11.25** | **20.93** | 68.11 | **79.16** | **77.05** | **69.97** | **73.39** | 44.04 | **44.20** | **65.13** |

Table 10: Recovery performance of different recovery data selection methods on LLaMA2-7B under AWQ 4bits scheme. The 'bold' represents the best performance among different selection methods. Here, the Alpaca is taken as the original dataset. The "Reason" indicates the averaged performance on 7 commonsense reasoning datasets.

| Quantization | Benchmark | w/o Training | Full Data | Random | Instruction Mining | IFD | Nuggets | PASER |
|---|---|---|---|---|---|---|---|---|
| AWQ 4bits | WikiText2↓ | 13.52 | 12.10 | 13.27 | 12.02 | 12.24 | 11.58 | **11.05** |
| | PTB↓ | 23.29 | 22.76 | 22.23 | 21.51 | 20.12 | 19.74 | **18.48** |
| | Reason↑ | 61.47 | 62.32 | 62.41 | 63.19 | 63.46 | 63.13 | **65.54** |

## G  EXTENDED EXPERIMENTS ON POST-QUANTIZATION RECOVERY TRAINING

Though the method descriptions and the experiments in the main body are mainly around the LLM pruning scenario, our PASER framework can actually be extended seamlessly to other LLM compression scenario. To further demonstrate its applicability, we conduct the experiments on post-quantization recovery training and compare our PASER with corresponding instruction tuning data selection baselines. In detail, we choose two most representative methods: Round-To-Nearest (RTN) (Frantar & Alistarh, 2022; Yao et al., 2022), GPTQ (Frantar et al., 2023) to perform the LLM quantization. It should be clarified that RTN method, which rounds all weights to the nearest quantized value on exactly the same asymmetric per-row grid, is actually the fundamental technique in most works about LLM quantization (Frantar & Alistarh, 2022; Yao et al., 2022; Park et al.).

Its runtimes scales well to the models with many billion parameters due to the direct rounding. According to the results provided in Table 8 and 9, we can observe that the PASER can still effectively enhance the recovery performance and outperform the data selection baselines on averaged reasoning performance and zero-shot perplexity for both LLaMA2-7B and LLaMA2-13B models. Meanwhile, recovery data selection baselines can indeed achieve the stronger performance than full data and random baselines, which validates the necessity of conducting recovery data selection even in the LLM quantization scenario. Furthermore, comparing these results with Table 1 and 2, it can be noticed that the improvement space of PASER in Table 8 and 9 has been reduced to some extent. This is because the post-compression performance of such quantization schemes has been competitive enough, which can reflected from the w/o training row. In addition, we also conduct experiments on one of the most robust LLM quantization scheme, AWQ (Lin et al., 2024), and the results are provided in the Table 10. Considering the superior performance of AWQ itself on reducing the performance drop brought by the precision loss, our improvement over baselines is still obvious, further demonstrating the generalizability of PASER on quantization schemes.

Table 11: Recovery performance of different instruction tuning data selection methods on LLaMA2-7B under different LLM distillation schemes. The 'bold' represents the best performance on the same distillation scheme. Here, the Alpaca is taken as the original dataset. The "Reason" indicates the averaged performance on 7 commonsense reasoning datasets.

| Distillation | Benchmark | w/o Training | Full Data | Random | Instruction Mining | IFD | Nuggets | PASER |
|---|---|---|---|---|---|---|---|---|
| DistiLLM | WikiText2↓ | 23.58 | 27.31 | 22.50 | 21.43 | 20.32 | 20.09 | **18.65** |
| | PTB↓ | 46.83 | 43.71 | 44.08 | 41.61 | 38.17 | 36.82 | **32.86** |
| | Reason↑ | 55.20 | 57.12 | 56.29 | 57.71 | 58.55 | 58.42 | **60.34** |
| MiniPLM | WikiText2↓ | 22.71 | 25.04 | 21.37 | 20.74 | 19.77 | 18.92 | **17.27** |
| | PTB↓ | 43.09 | 42.02 | 42.71 | 41.28 | 39.03 | 37.22 | **33.08** |
| | Reason↑ | 56.43 | 58.21 | 56.07 | 57.50 | 59.39 | 59.21 | **61.42** |
| SKD | WikiText2↓ | 24.45 | 25.82 | 21.14 | 20.48 | 19.20 | 18.56 | **17.07** |
| | PTB↓ | 44.73 | 41.49 | 42.02 | 39.67 | 36.52 | 34.31 | **30.75** |
| | Reason↑ | 56.67 | 59.61 | 57.38 | 58.79 | 59.82 | 59.50 | **61.94** |

## H  EXTENDED EXPERIMENTS ON POST-DISTILLATION RECOVERY TRAINING

In addition to the post-pruning and post-quantization experiments, we also supplement post-distillation experiments to achieve a full LLM post-compression recovery training picture. In detail, we adopt DistiLLM (Ko et al., 2024), MiniPLM (Gu et al.), SKD (Xu et al.), respectively, to distill a LLaMA2-7B model into a 1.5B scale small language model. The corresponding post-distillation recovery performance has been provided in the Table 11. From the table, we can observe that our PASER can effectively help recover LLM capabilities compressed with different recent distillation methods. Therefore, we can conclude that our proposed PASER can generalize well across different kinds of LLM compression paradigms, not limited to the pruning discussed in the main text.

## I  EXPERIMENTS ON RECOVERY TRAINING WITH KNOWLEDGE DISTILLATION

Inspired by (Muralidharan et al., 2024), we explore the knowledge distillation as the recovery post-training paradigm instead of the standard supervised learning with the groundtruth label. Here, we set the original model $M_o$ as the teacher and the pruned model $M_p$ as the student. The mean KL divergence (Kullback & Leibler, 1951) between the output probability distribution of $M_o$ and that of $M_p$ is taken as the objective function. Comparing the corresponding results under different pruning schemes in Table 12 with that in Table 1, we can first observe that knowledge distillation can effectively improve the recovery performance on both reasoning and language modeling tasks in most cases. In particular, the reasoning performance of PASER is improved by 0.348 points on average among such four pruning schemes. Interestingly, the knowledge distillation recovery performance of Full Data under LLM-Pruner is much better than that with standard label-supervised learning. This demonstrates that knowledge distillation is also a promising approach to avoid the misleading information from the irrelevant or conflicting samples existing in the original dataset. Because its learning process directly imitates the unpruned model behavior instead of the provided labels, thus better preserving the thinking and decision-making consistency with the original model. As a summary, distilling the knowledge of unpruned model into the pruned model can be regarded

Table 12: Knowledge distillation recovery performance of different instruction tuning data selection methods under various pruning schemes on LLaMA2-7B model. The 'bold' represents the best performance under the same pruning scheme. Here, the Alpaca is taken as the original dataset.

| Pruning | Recovery Post-training | WikiText2↓ | PTB↓ | BoolQ | PIQA | HellaSwag | WinoGrande | ARC-e | ARC-c | OBQA | Average |
|---|---|---|---|---|---|---|---|---|---|---|---|
| w/o pruning | w/o Training | 12.62 | 22.14 | 71.13 | 78.40 | 72.79 | 67.17 | 69.36 | 40.70 | 40.80 | 62.91 |
| LLM-Pruner ratio=25% | w/o Training | 20.34 | 38.81 | 61.87 | 76.61 | 65.86 | 60.22 | 63.13 | 37.37 | 39.40 | 57.78 |
| | Full Data | 24.72 | 43.91 | 63.30 | 76.01 | 67.18 | 62.27 | 64.23 | 36.86 | 39.20 | 58.44 |
| | Random | 23.82 | 41.20 | **68.03** | 74.89 | 66.27 | 64.51 | 64.65 | 32.58 | 38.30 | 58.46 |
| | Instruction Mining | 22.65 | 39.40 | 62.17 | 75.98 | 66.74 | 61.29 | 63.01 | 38.32 | 39.60 | 58.16 |
| | IFD | 19.17 | 32.30 | 64.13 | 77.55 | 67.89 | 61.56 | 64.09 | 38.19 | **40.40** | 59.12 |
| | Nuggets | 18.64 | 32.19 | 64.46 | 76.66 | 67.26 | 64.88 | 66.50 | 36.52 | 39.20 | 59.35 |
| | PASER | **15.91** | **25.39** | 67.89 | **77.81** | **69.62** | **67.63** | **68.46** | **39.87** | 40.20 | **61.64** |
| SliceGPT ratio=25% | w/o training | 44.53 | 80.07 | 65.54 | 66.87 | 54.16 | 63.38 | 58.46 | 34.56 | 36.90 | 54.27 |
| | Full Data | 35.48 | 66.25 | 69.35 | 70.34 | 58.50 | 66.76 | 62.95 | 37.14 | 38.70 | 57.68 |
| | Random | 38.63 | 65.67 | 67.19 | 68.59 | 56.21 | 64.94 | 60.63 | 35.61 | 37.80 | 55.85 |
| | Instruction Mining | 35.56 | 62.14 | 68.41 | 69.51 | 57.08 | 66.33 | 62.51 | 36.59 | 38.00 | 56.92 |
| | IFD | 33.50 | 61.33 | 69.51 | 70.82 | 58.70 | 67.49 | 64.09 | 37.22 | 38.50 | 58.05 |
| | Nuggets | 21.39 | 32.83 | 70.17 | 71.49 | 59.11 | 67.94 | **72.51** | 37.54 | 38.70 | 59.64 |
| | PASER | **11.87** | **20.91** | **73.43** | **80.32** | **74.46** | **69.76** | 71.95 | **42.26** | **41.70** | **64.84** |
| Wanda sparsity=2:4 | w/o training | 42.10 | 76.85 | 69.30 | 71.99 | 53.06 | 62.75 | 60.94 | 28.07 | 34.60 | 54.39 |
| | Full Data | 25.92 | 47.85 | 71.09 | 75.14 | 64.10 | 65.62 | 65.64 | 34.38 | 37.50 | 59.07 |
| | Random | 34.98 | 63.47 | 70.18 | 73.62 | 59.15 | 63.83 | 63.70 | 32.13 | 36.50 | 57.02 |
| | Instruction Mining | 30.56 | 55.56 | 71.03 | 73.97 | 61.69 | 64.56 | 64.86 | 33.93 | 37.00 | 58.15 |
| | IFD | 24.08 | 41.44 | 71.78 | 75.89 | 64.83 | 65.72 | **68.89** | 35.97 | 38.00 | 60.15 |
| | Nuggets | 23.14 | 40.10 | **72.26** | 76.50 | 65.33 | 66.03 | 66.52 | 37.27 | 38.60 | 60.36 |
| | PASER | **13.84** | **23.54** | 71.25 | **78.15** | **72.06** | 66.64 | 68.66 | 39.38 | 40.50 | 62.38 |
| SparseGPT sparsity=50% | w/o training | 19.26 | 36.41 | 71.22 | 75.60 | 62.85 | 66.06 | 69.11 | 36.86 | 37.80 | 59.93 |
| | Full Data | 28.17 | 52.82 | 68.52 | 75.77 | 57.84 | 69.26 | 60.43 | 37.72 | 37.00 | 58.08 |
| | Random | 25.31 | 43.22 | 69.74 | 74.91 | 60.28 | 68.10 | 64.06 | 39.95 | **39.80** | 59.55 |
| | Instruction Mining | 21.56 | 39.61 | 71.12 | 74.85 | 62.53 | 66.06 | 68.07 | 36.85 | 37.80 | 59.61 |
| | IFD | 17.76 | 31.25 | 71.70 | 75.76 | 63.43 | 66.06 | 69.14 | 36.59 | 37.60 | 60.04 |
| | Nuggets | 14.83 | 25.38 | 72.18 | 75.95 | 63.91 | 66.29 | 69.75 | 36.86 | 37.70 | 60.38 |
| | PASER | **13.00** | **22.24** | **75.07** | **78.66** | **66.90** | 69.31 | **72.85** | 38.89 | 39.60 | 63.04 |

Table 13: Recovery performance comparison between different combinations of knowledge distillation (KD) and supervised fine-tuning (SF) under various pruning schemes. The 'bold' represents the best performance under the same pruning scheme.

| Recovery Training | WikiText2↓ | PTB↓ | BoolQ | PIQA | HellaSwag | WinoGrande | ARC-e | ARC-c | OBQA | Average |
|---|---|---|---|---|---|---|---|---|---|---|
| LLM-Pruner (ratio=25%) | | | | | | | | | | |
| KD | 15.91 | 25.39 | 67.89 | 77.81 | 69.62 | 67.63 | 68.46 | 39.87 | 40.20 | **61.64** |
| SF | 16.40 | 26.35 | 67.25 | 77.29 | 68.98 | 66.97 | 67.84 | 39.54 | 39.80 | 61.10 |
| First KD, then SF | 16.15 | 25.87 | 67.57 | 77.55 | 69.31 | 67.30 | 68.15 | 39.71 | 40.00 | 61.37 |
| First SF, then KD | 16.28 | 26.02 | 67.41 | 77.43 | 69.15 | 67.11 | 67.96 | 39.63 | 39.90 | 61.23 |
| SliceGPT (ratio=25%) | | | | | | | | | | |
| KD | 11.87 | 20.91 | 73.43 | 80.32 | 74.46 | 69.76 | 71.95 | 42.26 | 41.70 | **64.84** |
| SF | 12.24 | 21.53 | 72.75 | 79.84 | 73.92 | 69.18 | 71.37 | 41.82 | 41.30 | 64.31 |
| First KD, then SF | 12.06 | 21.24 | 73.12 | 80.05 | 74.18 | 69.45 | 71.62 | 42.03 | 41.50 | 64.56 |
| First SF, then KD | 12.15 | 21.38 | 72.94 | 79.95 | 74.05 | 69.32 | 71.51 | 41.95 | 41.40 | 64.45 |
| Wanda (sparsity=2:4) | | | | | | | | | | |
| KD | 13.84 | 23.54 | 71.25 | 78.15 | 72.06 | 66.64 | 68.66 | 39.38 | 40.50 | **62.38** |
| SF | 14.13 | 27.22 | 70.77 | 77.87 | 71.78 | 66.26 | 68.30 | 39.04 | 40.10 | 62.02 |
| First KD, then SF | 13.97 | 25.31 | 71.02 | 78.03 | 71.94 | 66.47 | 68.49 | 39.23 | 40.30 | 62.21 |
| First SF, then KD | 14.05 | 26.28 | 70.89 | 77.95 | 71.85 | 66.35 | 68.41 | 39.15 | 40.20 | 62.11 |
| SparseGPT (sparsity=50%) | | | | | | | | | | |
| KD | 13.00 | 22.24 | 75.07 | 78.66 | 66.90 | 69.31 | 72.85 | 38.89 | 39.60 | **63.04** |
| SF | 13.33 | 23.77 | 74.79 | 78.38 | 66.62 | 69.03 | 72.57 | 38.70 | 39.40 | 62.78 |
| First KD, then SF | 13.15 | 22.96 | 74.94 | 78.53 | 66.78 | 69.18 | 72.72 | 38.81 | 39.50 | 62.92 |
| First SF, then KD | 13.24 | 23.35 | 74.85 | 78.45 | 66.70 | 69.11 | 72.64 | 38.75 | 39.45 | 62.85 |

as an effective way to enhance the recovery performance, though bring more memory overhead. Furthermore, stronger layer-wise distillation can also be taken into consideration (Jiao et al., 2020).

**Exploration on Combined Training Strategies** Given the complementary potential of knowledge distillation (KD) and supervised fine-tuning (SF), we further explore whether combining these two approaches could lead to enhanced recovery performance. Specifically, we investigate two cascading strategies: (1) first applying KD followed by SF, and (2) first conducting SF followed by KD. Table 13 presents the results under different pruning schemes.

Interestingly, the results show that neither cascading strategy consistently outperforms individual KD or SF approaches. This suggests that these two training paradigms might actually serve similar functions in recovering model capabilities, making their combination redundant rather than comple-

Table 14: Recovery performance of different instruction tuning data selection methods on several more recent and advanced LLMs under LLM-Pruner scheme. The 'bold' represents the best performance on the same target LLM. Here, the Alpaca is taken as the original dataset. The "Reason" indicates the averaged performance on 7 common sense reasoning datasets.

| Model | Benchmark | w/o pruning | w/o Training | Full Data | Random | Instruction Mining | IFD | Nuggets | PASER |
|---|---|---|---|---|---|---|---|---|---|
| LLaMA3.1-8B ratio=25% | WikiText2↓ | 6.63 | 14.85 | 9.74 | 12.16 | 13.84 | 11.32 | 10.81 | **7.56** |
| | PTB↓ | 16.72 | 29.21 | 18.38 | 23.47 | 26.52 | 21.75 | 20.46 | **17.83** |
| | Reason↑ | 70.81 | 63.92 | 67.95 | 65.73 | 65.34 | 66.82 | 67.58 | **70.05** |
| Qwen2.5-7B ratio=25% | WikiText2↓ | 6.94 | 15.82 | 11.68 | 13.94 | 15.37 | 10.95 | 10.23 | **8.16** |
| | PTB↓ | 20.58 | 32.95 | 24.43 | 29.16 | 30.54 | 22.89 | 21.37 | **19.85** |
| | Reason↑ | 70.49 | 62.71 | 67.08 | 64.29 | 63.75 | 66.93 | 67.42 | **69.14** |
| Qwen3-8B ratio=25% | WikiText2↓ | 6.68 | 14.29 | 9.18 | 11.83 | 13.26 | 10.17 | 9.45 | **7.24** |
| | PTB↓ | 18.45 | 30.56 | 20.74 | 25.41 | 28.13 | 22.68 | 21.05 | **18.97** |
| | Reason↑ | 71.73 | 64.25 | 68.62 | 66.37 | 65.81 | 67.95 | 68.24 | **70.58** |
| Mixtral-8x7B ratio=25% | WikiText2↓ | 7.52 | 15.87 | 12.83 | 14.32 | 14.27 | 13.65 | 12.98 | **9.62** |
| | PTB↓ | 17.46 | 31.45 | 24.87 | 27.13 | 26.58 | 25.21 | 24.07 | **19.75** |
| | Reason↑ | 72.04 | 61.56 | 64.83 | 63.75 | 65.21 | 66.48 | 67.24 | **70.87** |

mentary. Knowledge distillation shows slightly better performance across all pruning schemes, which could be attributed to its ability to capture the nuanced knowledge encoded in the teacher model's full output distribution. However, the marginal gains from combining approaches do not justify the additional computational overhead required for cascaded training.

## J EXPERIMENTS ON MORE RECENT LLMS

In addition to the previous experiments conducted on LLaMA2/3 and Baichuan2 families, we also supplement some results on more recent and advanced LLMs: LLaMA3.1-8B [2], Qwen2.5-7B (Yang et al., 2024) [3], Qwen3-8B [4]. Besides, the Mixture of Experts(MOE) model Mixtral-8×7B (Jiang et al., 2024) [5] is also employed to conduct the experiments. The evaluation results have been provided in Table 14. From Table 14, we can observe that PASER consistently outperforms all baseline methods across different recent and advanced LLMs, including the MOE architecture (Mixtral-8x7B). For LLaMA3.1-8B, PASER achieves perplexity scores (7.56 on WikiText2, 17.83 on PTB) that are remarkably close to those of the unpruned model (6.63 and 16.72, respectively), with only a 0.76 point gap in reasoning performance (70.05 vs. 70.81). Notably, PASER demonstrates robust performance across different model architectures. For the MoE-based Mixtral-8x7B, PASER significantly narrows the performance gap from 10.48 points (w/o Training: 61.56 vs. w/o pruning: 72.04) to just 1.17 points (PASER: 70.87 vs. w/o pruning: 72.04), representing a 88.8% recovery of lost capability. This demonstrates that our capability-aware selection approach effectively addresses the specialized recovery needs of sparse models with expert-based architectures. Among the most recent models, Qwen3-8B shows the strongest recovery with PASER, achieving 98.4% of the unpruned model's reasoning performance (70.58 vs. 71.73). Meanwhile, Qwen2.5-7B demonstrates slightly lower recovery effectiveness, potentially due to differences in its architecture and pretraining approach. These results further validate PASER's applicability across diverse and state-of-the-art LLM architectures, confirming that our capability degradation-aware instruction selection and negative tuning effects mitigation strategies generalize well beyond the models in the main paper.

## K DISCUSSION ON COMPARISON WITH GENERAL DATA SELECTION METHODS

To further demonstrate our PASER is an unique contribution and solution for pruned-LLM recovery, we discuss the comparison with previous general data selection methods (e.g., LESS (Xia et al., 2024b), DELIFT (Agarwal et al., 2025), SMART (Renduchintala et al., 2024), Datamodels (Ilyas et al., 2022)) from four perspectives: 1) Why general data selection methods do not suffice for pruned-LLM recovery? (Appendix K.1) 2) How PASER differs from previous general data selection methods?

---

[2] https://huggingface.co/meta-llama/Llama-3.1-8B

[3] https://huggingface.co/Qwen/Qwen2.5-7B

[4] Qwen/Qwen3-8B

[5] https://huggingface.co/mistralai/Mixtral-8x7B-v0.1

(Appendix K.2) 3) What is specific comparison between PASER and gradient-based sampling? (Appendix K.3) 4) Can PASER benefit from gradient-derived dense information? (Appendix K.4)

## K.1 Q1: WHY GENERAL DATA SELECTION METHODS DO NOT SUFFICE FOR PRUNED-LLM RECOVERY?

While existing data selection methods such as LESS (Xia et al., 2024b), DELIFT (Agarwal et al., 2025), SMART (Renduchintala et al., 2024), and Datamodels (Ilyas et al., 2022) have demonstrated effectiveness for various training objectives (e.g., improving generalization, robustness, or efficiency), they are not specifically tailored to the unique recovery challenges posed by pruning. Notably:

- **Uneven Capability Degradation**: Pruning introduces non-uniform degradation across different model capabilities (e.g., commonsense, coding, math), as shown in Figure 4 of Appendix A. General selection methods typically assume uniform importance across examples or rely on training dynamics in full-capacity models, failing to prioritize capability-specific weaknesses that are critical in post-pruning recovery (see Section 1 and Section 3.3). For example, as discussed in the Line 48-50, existing data selection methods generally favor samples with clear structure and natural, human-like expressions. However, these samples may not exactly target specific capabilities (e.g., math, code) severely compromised during pruning. In fact, in the post-pruning recovery scenario, even if the expression of samples regarding severely compromised capabilities is not so clear or structured, we should still prioritize them.

- **Performance Sensitivity to Instruction Content**: As shown empirically in Figure 3 of Appendix A, recovery performance varies drastically depending on which subset is selected, an issue rarely addressed by general-purpose selection algorithms. This is a unique challenge in pruned model recovery, as pruning disrupts the model's internal structure and weakens its representational capacity, making the choice of recovery data more critical and sensitive than in standard fine-tuning scenarios.

- **Incompatibility with Recovery Training Goals**: Many existing methods select samples that maximize representativeness or informativeness during pre-training or fine-tuning, which may inadvertently include instructions that are irrelevant or conflicting with each other, thus resulting in harmful impact for recovering pruned capabilities.

Thus, applying these methods directly without modification often leads to inefficient use of recovery budgets and fails to address the nuanced degradation patterns introduced by pruning.

## K.2 Q2: HOW PASER DIFFERS FROM PREVIOUS GENERAL DATA SELECTION METHODS?

PASER is designed from first principles for post-pruning recovery and introduces three key innovations absent in prior methods:

- **Capability-Aware Clustering and Budget Allocation**: Unlike data selection methods that focus on sample diversity or informativeness globally, PASER performs semantic-structural clustering (Section 3.2) to uncover latent instruction groups aligned with distinct capabilities. We then assess degradation in each cluster using Jensen-Shannon Divergence between pruned and original model outputs (Section 3.3), and allocate recovery budget proportionally. This enables targeted and balanced recovery, which existing methods do not consider. In contrast, existing methods typically treat all capabilities uniformly and are agnostic to which skills the model has lost post-pruning. As a result, they may waste recovery budget on less-affected areas while failing to restore critical capabilities.

- **Efficiency-Driven Sample Selection**: PASER integrates computational efficiency by computing a sample-wise efficiency score based on degradation severity and expected training cost. This is crucial in recovery settings where data budgets and compute are highly constrained. In contrast, datamodel-based selection methods (e.g., Datamodels (Ilyas et al., 2022)) require training large probes or repeated forward/backward passes, making them computationally expensive and impractical for efficient recovery. LESS is also time-consuming, because it involves gradient computation and LoRA training during the selection process.

- **Negative Tuning Mitigation via Concept Consistency Graph**: General-purpose selection approaches rarely guard against harmful instructions. PASER introduces a Concept Consistency

Graph (Section 3.4) to detect and filter irrelevant or conflicting samples that might lead to negative tuning effects. This is a particularly pressing issue when the model's representational capacity has been impaired by pruning, which results in models more sensitive to conflicting information. However, existing methods rarely address this risk, most of which assume all data in high-quality pools is beneficial. However, our results (e.g., Table 1) show that even using the full dataset can degrade performance due to negative tuning effects, especially for pruned models with weakened internal representations

Besides, we would like to highlight that across all pruning schemes and models (Tables 1, 2, 3), PASER consistently outperforms random, full data, and several instruction-tuning selection baselines (e.g., Nuggets, IFD), despite using only 4%–20% of the original data. Additionally, our ablation study (Table 4) confirms the importance of each PASER module in achieving superior recovery.

In summary, PASER is motivated by and explicitly designed for the unique characteristics and challenges of pruned LLM recovery. Existing general data selection methods neither target uneven capability loss nor mitigate the risk of irrelevant or conflicting data. PASER fills this gap by introducing capability-driven, efficiency-aware, and consistency-preserving data selection, thus leading to substantial gains in both recovery quality and training cost.

### K.3  Q3: WHAT IS SPECIFIC DIFFERENCE BETWEEN PASER AND GRADIENT-BASED SAMPLING?

We would like to clarify the fundamental differences between PASER and gradient-based data selection strategies such as LESS (Xia et al., 2024b), and supplement this with a controlled empirical comparison.

**Conceptual Differences** Gradient-based methods (e.g., LESS) typically rely on computing influence scores or gradient norms to identify impactful training samples. These scores are often computed with respect to specific layers or objectives (e.g., classification loss or instruction adherence) and require gradient backpropagation, which incurs high computational cost—especially when evaluating all samples across large instruction tuning sets. In contrast, PASER is specifically designed for capability recovery after structured/semi-structured/unstructured pruning, where the pruned model exhibits uneven degradation across semantic capabilities. Rather than using gradient signals, PASER leverages: 1) Semantic clustering to disentangle instruction groups aligned with distinct capabilities; 2) Capability Degradation Scores (CDS) based on Jensen-Shannon divergence (JSD) between the pruned and original models' output distributions; 3) Efficiency-aware scoring to optimize recovery benefit per unit of compute cost; 4) Negative tuning effect mitigation using concept consistency graphs. This design enables PASER to approximate dense information signals (through token-level divergence) while being more effective, efficient, and robust than backpropagation-dependent strategies.

**Empirical Comparison** To further clarify the performance and efficiency differences, we compare PASER with a LESS-style gradient-based sampling baseline, which selects the top 20% samples from Alpaca with the highest gradient norm of loss w.r.t. embedding parameters. As shown in Table 15,

Table 15: Recovery performance of different data selection methods on LLaMA2-7B under different LLM distillation schemes. Here, the Alpaca is taken as the original dataset. The "Reason" indicates the averaged performance on 7 commonsense reasoning datasets.

| Pruning | Recovery Data Selection | Reasoning ↑ | WikiText2 ↓ | PTB ↓ | Data Selection Time (mins) ↓ | Recovery Training Time (mins) ↓ |
|---|---|---|---|---|---|---|
| LLM-Pruner (25%) | PASER | 61.10±0.11 | 16.40±0.28 | 26.35±0.30 | 5.25 | 28.97 |
| | LESS-style | 458.48±0.37 | 20.21±2.16 | 34.20±6.37 | 163.10 | 41.35 |
| SliceGPT (25%) | PASER | 64.31±0.08 | 12.24±0.32 | 21.53±0.41 | 5.24 | 29.14 |
| | LESS-style | 59.09±0.35 | 25.83±4.03 | 38.42±5.28 | 148.51 | 42.04 |
| Wanda (2:4) | PASER | 62.02±0.14 | 14.13±0.21 | 27.22±0.29 | 5.35 | 30.37 |
| | LESS-style | 58.64±0.51 | 21.84±3.89 | 39.25±6.02 | 152.38 | 42.43 |
| SparseGPT (50%) | PASER | 62.78±0.10 | 13.33±0.25 | 23.77±0.33 | 5.22 | 28.63 |
| | LESS-style | 59.36±0.43 | 19.89±3.64 | 37.31±5.15 | 170.49 | 41.16 |

PASER not only achieves better recovery performance across both language modeling and reasoning tasks, but also incurs significantly less data selection time (up to 30× less) due to its gradient-free design. The recovery training time is also reduced (↓ 14.2%), as PASER's efficiency-driven sampling

strategy favors shorter and semantically cohesive instructions. Furthermore, PASER consistently exhibits lower performance variance across different pruning settings (e.g., reasoning accuracy std of 0.08–0.14 vs. 0.35–0.51 for LESS-style), indicating greater robustness under heterogeneous post-pruning degradation. In summary, PASER offers a more effective, efficient, robust, and pruning-aware alternative to gradient-based sampling strategies, especially in scenarios where gradient signals become noisy or unreliable due to structural capacity loss.

### K.4 Q4: Can PASER Benefit from Gradient-derived Dense Information?

While PASER does not explicitly rely on gradient information, we acknowledge that gradient-derived signals capture dense local sensitivity and can, in principle, provide useful guidance for data selection. In our current design, we approximate similar effects using token-level output distributional divergence (JSD) to reflect fine-grained capability deterioration without requiring backward computation. This makes PASER highly efficient and model-agnostic. That said, we agree with the reviewer that incorporating lightweight gradient signals could further enhance selection granularity. In future work, we plan to explore hybrid approaches by integrating gradient-based information into PASER in the following ways:

- **Gradient-Guided Cluster Weighting**: Use average gradient norms within each capability cluster to refine degradation scores alongside our JSD-based measure.

- **Gradient-Augmented Efficiency Scoring**: Combine gradient-based sample importance with our current JSD-based efficiency metric, possibly as a weighted fusion.

- **LoRA-style Gradient Probing**: Introduce low-rank adapters to cheaply approximate gradient information in a parameter-efficient manner during selection, especially useful for large LLMs.

We believe this line of integration can combine the dense expressiveness of gradients with PASER's efficient, capability-aware framework, further improving recovery quality without incurring excessive compute overhead.

Table 16: The detailed ablation study for our proposed three components under various pruning schemes on LLaMA2-7B model. The 'bold' represents the best performance under the same pruning scheme. Here, the Alpaca is taken as the original dataset.

| Pruning | Recovery Post-training | WikiText2↓ | PTB↓ | BoolQ | PIQA | HellaSwag | WinoGrande | ARC-e | ARC-c | OBQA | Average |
|---|---|---|---|---|---|---|---|---|---|---|---|
| LLM-Pruner ratio=25% | PASER w/o S$^2$RIC | 18.73 | 32.84 | 65.31 | 76.84 | 67.59 | 64.85 | 65.92 | 37.96 | 39.20 | 59.67 |
| | PASER w/o CDAIS | 17.56 | 30.15 | 66.27 | 77.03 | 68.15 | 65.73 | 66.58 | 38.54 | 39.50 | 60.26 |
| | PASER w/o NTEM | 19.82 | 35.60 | 64.83 | **77.52** | 67.34 | 64.48 | 63.59 | 36.78 | **40.20** | 59.25 |
| | PASER | **16.40** | **26.35** | **67.25** | 77.29 | **68.98** | **66.97** | **67.84** | **39.54** | 39.80 | **61.10** |
| SliceGPT ratio=25% | PASER w/o S$^2$RIC | 14.83 | 25.42 | 71.15 | 78.91 | 72.25 | 67.84 | 69.95 | 40.82 | 40.30 | 63.03 |
| | PASER w/o CDAIS | 14.16 | 24.92 | 70.89 | 78.56 | 71.84 | 67.45 | 69.58 | 40.47 | 40.00 | 62.68 |
| | PASER w/o NTEM | 15.37 | 27.81 | 69.97 | 77.33 | 70.68 | 65.92 | 68.03 | 39.39 | **42.10** | 61.92 |
| | PASER | **12.24** | **21.53** | **72.75** | **79.84** | **73.92** | **69.18** | **71.37** | **41.82** | 41.30 | **64.31** |
| Wanda sparsity=2:4 | PASER w/o S$^2$RIC | 15.84 | 30.25 | 69.26 | 77.42 | 70.31 | 65.82 | 67.84 | 38.67 | 39.00 | 61.19 |
| | PASER w/o CDAIS | 15.46 | 29.48 | 69.14 | 77.35 | 70.27 | 65.74 | 67.79 | 38.75 | 39.60 | 61.23 |
| | PASER w/o NTEM | 16.79 | 31.52 | 69.51 | 76.92 | 70.76 | 65.23 | 67.28 | 38.47 | **41.20** | 61.34 |
| | PASER | **14.13** | **27.22** | **70.77** | **77.87** | **71.78** | **66.26** | **68.30** | **39.04** | 40.10 | **62.02** |
| SparseGPT sparsity=50% | PASER w/o S$^2$RIC | 14.89 | 26.31 | 73.25 | 77.45 | **70.15** | 68.47 | 69.28 | 39.82 | 39.80 | 62.60 |
| | PASER w/o CDAIS | 14.62 | 25.84 | 72.91 | 77.50 | 69.93 | 68.12 | 69.05 | **39.94** | 40.00 | 62.49 |
| | PASER w/o NTEM | 15.91 | 28.19 | 71.53 | **78.62** | 65.48 | 67.21 | 69.79 | 39.18 | **40.50** | 61.76 |
| | PASER | **13.33** | **23.77** | **74.79** | 78.38 | 66.62 | **69.03** | **72.57** | 38.70 | 39.40 | **62.78** |

## L Detailed Ablation Study Results

In this section, we present comprehensive ablation results of the three key components in PASER: semantic-structural recovery instruction clustering (S$^2$RIC), capability degradation-aware instruction selection (CDAIS), and negative tuning effects mitigation (NTEM). Table 16 shows the detailed performance across different evaluation metrics.

The detailed results reveal the distinct contributions of each component under different pruning schemes. For structured pruning like LLM-Pruner, removing S$^2$RIC leads to significant degradation in both language modeling (perplexity increases from 16.40 to 18.73 on WikiText2) and reasoning tasks (average score drops by 1.43 points), highlighting its importance in addressing uneven capability degradation. The impact of CDAIS is particularly evident under SliceGPT, where its removal

causes a 1.63-point drop in average reasoning performance while maintaining relatively stable language modeling metrics, demonstrating its effectiveness in balancing recovery priorities. Under semi-structured pruning (Wanda), all three components show more balanced contributions, with performance drops ranging from 0.68 to 0.83 points when each is removed. This suggests that semi-structured pruning requires a more holistic recovery approach. For unstructured pruning (SparseGPT) where capability degradation tends to be more uniform, NTEM plays a particularly crucial role - its removal leads to the largest drop in language modeling performance (perplexity increases from 13.33 to 15.91 on WikiText2) and affects complex reasoning tasks like WinoGrande and ARC-e significantly. Notably, the full PASER framework consistently achieves the best performance across almost all metrics under various pruning schemes, with only occasional exceptions in individual tasks (e.g., OBQA in LLM-Pruner and PIQA in SparseGPT). This comprehensive superiority validates our design choice of combining these three components for effective pruned model recovery.

## M    HYPERPARAMETER ROBUSTNESS ANALYSIS

To demonstrate the robustness of PASER to various hyperparameter settings, we conducted a comprehensive sensitivity analysis. This analysis helps establish that our method's strong performance is not dependent on specific hyperparameter configurations but remains consistent across a reasonable range of settings.

### M.1    SENSITIVITY TO EMBEDDING DIMENSION

The dimensionality of the manifold representation after applying the diffusion kernel is controlled by parameter $d$. We evaluate PASER's performance with $d$ ranging from 4 to 64 on LLaMA2-7B under LLM-Pruner (ratio=25%). As shown in left subfigure of Figure 5, clustering results remains consistent (Rand Index >0.85) across this range, and average reasoning performance variations is controlled within $\pm 0.2$ points. This indicates that PASER is relatively stable to the specific choice of embedding dimension, though we select $d = 16$ based on the eigenvalue decay pattern, where eigenvalues beyond this dimension contribute negligibly to the representation.

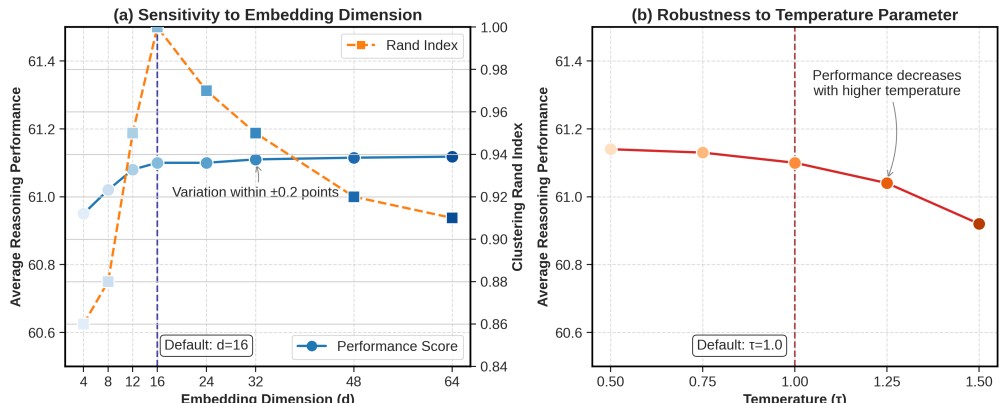

Figure 5: (a) Sensitivity to embedding dimension $d$ after manifold learning, here the clustering results under $d = 16$ is taken as the reference; (b) Robustness to temperature parameter $\tau$ in Equation 5.

### M.2    ROBUSTNESS TO TEMPERATURE PARAMETER

For the Jensen-Shannon divergence (JSD) calculation in the capability degradation score, we investigate the effect of temperature $\tau$ in the softmax function (Equation 5). In right subfigure of Figure 5, we test values in the range $\tau \in \{0.5, 0.75, 1.0, 1.25, 1.5\}$ and observe that while lower temperatures ($\tau < 0.75$) lead to slightly sharper capability distinctions, the overall performance differences are minimal (<0.3 points). Our default setting of $\tau = 1.0$ provides a good balance between sensitivity to capability differences and stability across pruning schemes.

Overall, our hyperparameter sensitivity analysis demonstrates that PASER's performance is robust across a wide range of hyperparameter settings. This stability enhances the practical applicability of our method, as it does not require extensive hyperparameter tuning to achieve strong performance in real-world scenarios.

## N    EMPIRICAL EXPLORATION FOR COMPONENT SELECTION

To demonstrate the selection optimality of each component in our PASER framework, we conduct the empirical exploration on alternative initial instruction embedding models, dimensionality reduction methods, clustering methods, divergence measurement methods, and budget allocation strategy, to investigate if they can achieve better final recovery performance.

Table 17: Recovery performance of multiple PASER versions integrated with initial instruction embedding models under various pruning schemes on LLaMA2-7B model. The PASER(SentenceBERT) is the version we employ in the main body. The 'bold' represents the best performance under the same pruning scheme. Here, the Alpaca is taken as the original dataset.

| Pruning | Recovery Post-training | WikiText2↓ | PTB↓ | BoolQ | PIQA | HellaSwag | WinoGrande | ARC-e | ARC-c | OBQA | Average |
|---|---|---|---|---|---|---|---|---|---|---|---|
| w/o pruning | w/o Training | 12.62 | 22.14 | 71.13 | 78.40 | 72.79 | 67.17 | 69.36 | 40.70 | 40.80 | 62.91 |
| LLM-Pruner ratio=25% | w/o Training | 20.34 | 38.81 | 61.87 | 76.61 | 65.86 | 60.22 | 63.13 | 37.37 | 39.40 | 57.78 |
| | PASER(Qwen2.5-7B) | 16.38 | 26.33 | 67.27 | **77.32** | **69.01** | **67.00** | 67.86 | 39.56 | 39.80 | 61.12 |
| | PASER(LLaMA3-8B) | **16.37** | **26.31** | **67.29** | **77.32** | **69.01** | **67.00** | 67.84 | **39.58** | **40.00** | **61.15** |
| | PASER(SentenceBERT) | 16.40 | 26.35 | 67.25 | 77.29 | 68.98 | 66.97 | 67.84 | 39.54 | 39.80 | 61.10 |
| SliceGPT ratio=25% | w/o Training | 44.53 | 80.07 | 65.54 | 66.87 | 54.16 | 63.38 | 58.46 | 34.56 | 36.90 | 54.27 |
| | PASER(Qwen2.5-7B) | 12.22 | 21.51 | 72.78 | **79.86** | **73.94** | **69.20** | 71.39 | **41.86** | **41.30** | 64.33 |
| | PASER(LLaMA3-8B) | **12.21** | **21.50** | **72.80** | **79.86** | **73.94** | **69.20** | 71.40 | 41.85 | **41.30** | **64.34** |
| | PASER(SentenceBERT) | 12.24 | 21.53 | 72.75 | 79.84 | 73.92 | 69.18 | 71.37 | 41.82 | **41.30** | 64.31 |
| Wanda sparsity=2:4 | w/o Training | 42.10 | 76.85 | 69.30 | 71.99 | 53.06 | 62.75 | 60.94 | 28.07 | 34.60 | 54.39 |
| | PASER(Qwen2.5-7B) | 14.11 | **27.20** | 70.79 | 77.87 | 71.78 | 66.26 | 68.32 | 39.03 | **40.10** | 62.02 |
| | PASER(LLaMA3-8B) | **14.10** | 27.22 | **70.80** | **77.90** | **71.80** | **66.29** | **68.33** | **39.07** | **40.10** | **62.04** |
| | PASER(SentenceBERT) | 14.13 | 27.22 | 70.77 | 77.87 | 71.78 | 66.26 | 68.30 | 39.04 | **40.10** | 62.02 |
| SparseGPT sparsity=50% | w/o Training | 19.26 | 36.41 | 71.22 | 75.60 | 62.85 | 66.06 | 69.11 | 36.86 | 37.80 | 59.93 |
| | PASER(Qwen2.5-7B) | 13.62 | 24.36 | 74.65 | 78.26 | 66.46 | 68.80 | 72.50 | **38.72** | 39.30 | 62.67 |
| | PASER(LLaMA3-8B) | 13.50 | 23.98 | 74.72 | 78.31 | 66.55 | 68.96 | 72.50 | 38.63 | 39.30 | 62.71 |
| | PASER(SentenceBERT) | **13.33** | **23.77** | **74.79** | **78.38** | **66.62** | **69.03** | **72.57** | 38.70 | **39.40** | **62.78** |

### N.1    EXPLORATION ON OTHER POSSIBLE INSTRUCTION EMBEDDING MODELS

In the Semantic-Structal Recovery Instruction Clustering module of our PASER framework, we adopt the SentenceBERT (Reimers & Gurevych, 2019) to obtain the initial instruction embedding, considering that SentenceBERT has demonstrated strong transfer capability across various text semantic tasks. To further validate that SentenceBERT is competitive enough here with the empirical evidence, we replace it with two other advanced LLMs: Qwen2.5-7B (Yang et al., 2024) and LLaMA3-8B (Dubey et al., 2024) here and conduct the experiments. As shown in Table 17, using much larger pretrained language models like Qwen2.5-7B and LLaMA3-8B as embedding generators provides marginal performance improvements (less than 0.05 points on average) compared to SentenceBERT. While LLaMA3-8B achieves the highest performance in three configurations, these gains are negligible given the substantially increased computational costs associated with using such large models for initial embeddings. Besides, under the SparseGPT, sparsity=50% setting, SentenceBERT performs obviously better than LLaMA3-8B and Qwen2.5-7B. This finding validates our design choice of SentenceBERT as the default embedding model for PASER, as it strikes an optimal balance between semantic representation quality and computational efficiency. SentenceBERT requires significantly fewer resources while still enabling effective capability-specific clustering that supports our targeted recovery approach. The results demonstrate that while more powerful embedding models may theoretically provide better semantic representations, in practice, the performance difference is minimal for our specific task of recovery data selection.

### N.2    EXPLORATION ON OTHER POSSIBLE DIMENSIONALITY REDUCTION METHODS

To validate the suitability of diffusion kernel as dimensionality reduction approach which focus on uncovering the instruction geometric structure in the semantic space to facilitate the following capability clustering process, we replace it with other alternative schemes and compare the performance. As shown in Table 18, among different dimensionality reduction methods, our diffusion

Table 18: Recovery performance of multiple PASER versions integrated with different dimensionality reduction approaches under various pruning schemes on LLaMA2-7B model. The PASER(Diffusion) is the version we employ in the main body. The 'bold' represents the best performance under the same pruning scheme. Here, the Alpaca is taken as the original dataset.

| Pruning | Recovery Post-training | WikiText2↓ | PTB↓ | BoolQ | PIQA | HellaSwag | WinoGrande | ARC-e | ARC-c | OBQA | Average |
|---|---|---|---|---|---|---|---|---|---|---|---|
| w/o pruning | w/o Training | 12.62 | 22.14 | 71.13 | 78.40 | 72.79 | 67.17 | 69.36 | 40.70 | 40.80 | 62.91 |
| LLM-Pruner ratio=25% | w/o Training | 20.34 | 38.81 | 61.87 | 76.61 | 65.86 | 60.22 | 63.13 | 37.37 | 39.40 | 57.78 |
| | PASER(UMAP) | 16.92 | 27.83 | 66.31 | 76.84 | 67.59 | 65.85 | 66.92 | 38.96 | 39.20 | 60.31 |
| | PASER(PCA) | 17.05 | 28.16 | 66.18 | 76.73 | 67.46 | 65.73 | 66.84 | 38.87 | 39.10 | 60.18 |
| | PASER(t-SNE) | 17.21 | 28.42 | 66.05 | 76.68 | 67.34 | 65.64 | 66.74 | 38.73 | 39.00 | 60.05 |
| | PASER(Diffusion) | **16.40** | **26.35** | **67.25** | **77.29** | **68.98** | **66.97** | **67.84** | **39.54** | **39.80** | **61.10** |
| SliceGPT ratio=25% | w/o Training | 44.53 | 80.07 | 65.54 | 66.87 | 54.16 | 63.38 | 58.46 | 34.56 | 36.90 | 54.27 |
| | PASER(UMAP) | 14.83 | 25.42 | 71.15 | 78.91 | 72.25 | 67.84 | 69.95 | 40.82 | 40.30 | 63.03 |
| | PASER(PCA) | 13.87 | 23.75 | 71.73 | 79.31 | 72.79 | 68.48 | 70.53 | 41.28 | 40.70 | 63.55 |
| | PASER(t-SNE) | 13.59 | 23.24 | 72.04 | 79.53 | 73.06 | 68.73 | 70.81 | 41.43 | 40.90 | 63.79 |
| | PASER(Diffusion) | **12.24** | **21.53** | **72.75** | **79.84** | **73.92** | **69.18** | **71.37** | **41.82** | **41.30** | **64.31** |
| Wanda sparsity=2:4 | w/o Training | 42.10 | 76.85 | 69.30 | 71.99 | 53.06 | 62.75 | 60.94 | 28.07 | 34.60 | 54.39 |
| | PASER(UMAP) | 15.84 | 30.25 | 69.26 | 77.42 | 70.31 | 65.82 | 67.84 | 38.67 | 39.00 | 61.19 |
| | PASER(PCA) | 15.46 | 29.48 | 69.14 | 77.35 | 70.27 | 65.74 | 67.79 | 38.75 | 39.60 | 61.23 |
| | PASER(t-SNE) | 14.72 | 28.21 | 70.14 | 77.62 | 71.08 | 66.01 | 68.02 | 38.85 | 39.80 | 61.65 |
| | PASER(Diffusion) | **14.13** | **27.22** | **70.77** | **77.87** | **71.78** | **66.26** | **68.30** | **39.04** | **40.10** | **62.02** |
| SparseGPT sparsity=50% | w/o Training | 19.26 | 36.41 | 71.22 | 75.60 | 62.85 | 66.06 | 69.11 | 36.86 | 37.80 | 59.93 |
| | PASER(UMAP) | 14.89 | 26.31 | 73.25 | 77.45 | 65.15 | 68.47 | 71.28 | 37.82 | 38.80 | 61.75 |
| | PASER(PCA) | 14.62 | 25.84 | 73.51 | 77.68 | 65.52 | 68.73 | 71.61 | 38.14 | 38.90 | 62.01 |
| | PASER(t-SNE) | 13.98 | 24.95 | 74.13 | 78.04 | 66.14 | 68.86 | 72.09 | 38.43 | 39.10 | 62.40 |
| | PASER(Diffusion) | **13.33** | **23.77** | **74.79** | **78.38** | **66.62** | **69.03** | **72.57** | **38.70** | **39.40** | **62.78** |

kernel approach consistently outperforms alternatives across all pruning schemes. The diffusion kernel's ability to preserve manifold structure while adapting to the intrinsic geometry of instruction data proves particularly advantageous for capability-aware clustering. UMAP, PCA, and t-SNE each demonstrate reasonable performance, but they fall short by 0.38-1.28 points compared to the diffusion kernel approach on average reasoning tasks. The performance gap is especially pronounced under structured pruning schemes like LLM-Pruner (0.79 points advantage) and SliceGPT (0.52 points advantage), where capability degradation tends to be more severe and uneven. This confirms that the diffusion kernel's non-linear dimensionality reduction better captures the complex relationships between instructions targeting similar capabilities, enabling more effective recovery prioritization. While t-SNE shows competitive performance in some scenarios, its sensitivity to hyperparameters and initialization makes it less stable for diverse instruction sets compared to our approach.

Table 19: Recovery performance of multiple PASER versions integrated with different data clustering approaches under various pruning schemes on LLaMA2-7B model. The PASER(S$^2$RIC) is the version we employ in the main body. The 'bold' represents the best performance under the same pruning scheme. Here, the Alpaca is taken as the original dataset.

| Pruning | Recovery Post-training | WikiText2↓ | PTB↓ | BoolQ | PIQA | HellaSwag | WinoGrande | ARC-e | ARC-c | OBQA | Average |
|---|---|---|---|---|---|---|---|---|---|---|---|
| w/o pruning | w/o Training | 12.62 | 22.14 | 71.13 | 78.40 | 72.79 | 67.17 | 69.36 | 40.70 | 40.80 | 62.91 |
| LLM-Pruner ratio=25% | w/o Training | 20.34 | 38.81 | 61.87 | 76.61 | 65.86 | 60.22 | 63.13 | 37.37 | 39.40 | 57.78 |
| | PASER(NMF_TFIDF) | 17.82 | 29.45 | 65.93 | 76.88 | 67.42 | 65.19 | 66.37 | 38.81 | 39.60 | 60.03 |
| | PASER(LDA_TFIDF) | 17.56 | 28.91 | 66.18 | 77.02 | 67.76 | 65.58 | 66.92 | 38.95 | 39.70 | 60.30 |
| | PASER(KMeans_TFIDF) | 17.21 | 28.13 | 66.47 | 77.15 | 68.04 | 65.92 | 67.23 | 39.12 | **39.80** | 60.53 |
| | PASER(Spectral_MTEB) | 16.82 | 27.24 | 66.89 | 77.23 | 68.46 | 66.38 | 67.56 | 39.31 | **39.80** | 60.80 |
| | PASER(Spectral_BERT) | 16.61 | 26.79 | 67.06 | 77.26 | 68.72 | 66.68 | 67.71 | 39.43 | **39.80** | 60.95 |
| | PASER(S$^2$RIC) | **16.40** | **26.35** | **67.25** | **77.29** | **68.98** | **66.97** | **67.84** | **39.54** | **39.80** | **61.10** |
| SliceGPT ratio=25% | w/o Training | 44.53 | 80.07 | 65.54 | 66.87 | 54.16 | 63.38 | 58.46 | 34.56 | 36.90 | 54.27 |
| | PASER(NMF_TFIDF) | 14.27 | 24.36 | 70.89 | 78.76 | 72.13 | 67.69 | 70.12 | **41.95** | 40.80 | 63.21 |
| | PASER(LDA_TFIDF) | 14.86 | 25.19 | 70.31 | 78.42 | 71.64 | 67.25 | 69.58 | 40.37 | 40.60 | 62.60 |
| | PASER(KMeans_TFIDF) | 13.58 | 23.42 | 71.46 | 79.07 | 72.61 | 68.14 | 70.48 | 41.08 | 41.00 | 63.41 |
| | PASER(Spectral_MTEB) | 12.91 | 22.47 | 72.08 | 79.41 | 73.18 | 68.62 | 70.87 | 41.43 | 41.10 | 63.81 |
| | PASER(Spectral_BERT) | 12.58 | 22.01 | 72.41 | 79.63 | 73.55 | 68.91 | 71.12 | 41.63 | 41.20 | 64.06 |
| | PASER(S$^2$RIC) | **12.24** | **21.53** | **72.75** | **79.84** | **73.92** | **69.18** | **71.37** | 41.82 | **41.30** | **64.31** |
| Wanda sparsity=2:4 | w/o Training | 42.10 | 76.85 | 69.30 | 71.99 | 53.06 | 62.75 | 60.94 | 28.07 | 34.60 | 54.39 |
| | PASER(NMF_TFIDF) | 16.18 | 30.94 | 70.09 | 76.68 | 69.98 | 64.82 | 66.92 | 38.14 | 39.60 | 60.89 |
| | PASER(LDA_TFIDF) | 18.74 | 34.98 | 69.85 | 76.31 | 69.42 | 64.37 | 66.48 | 37.82 | 39.40 | 60.52 |
| | PASER(KMeans_TFIDF) | 15.49 | 29.76 | **70.92** | 77.03 | 70.51 | 65.28 | 67.38 | 38.47 | **40.30** | 61.41 |
| | PASER(Spectral_MTEB) | 14.81 | 28.49 | 70.54 | 77.42 | 71.12 | 65.75 | 67.82 | 38.74 | 39.90 | 61.61 |
| | PASER(Spectral_BERT) | 14.47 | 27.86 | 70.66 | 77.65 | 71.45 | 66.01 | 68.06 | 38.89 | 40.00 | 61.82 |
| | PASER(S$^2$RIC) | **14.13** | **27.22** | 70.77 | **77.87** | **71.78** | **66.26** | **68.30** | **39.04** | 40.10 | **62.02** |
| SparseGPT sparsity=50% | w/o Training | 19.26 | 36.41 | 71.22 | 75.60 | 62.85 | 66.06 | 69.11 | 36.86 | 37.80 | 59.93 |
| | PASER(NMF_TFIDF) | 15.97 | 28.13 | 72.63 | 76.94 | 64.37 | 67.18 | 70.39 | 37.54 | 38.60 | 61.09 |
| | PASER(LDA_TFIDF) | 15.41 | 27.09 | 73.12 | 77.31 | 64.93 | 67.63 | 70.92 | 37.86 | 38.80 | 61.51 |
| | PASER(KMeans_TFIDF) | 14.72 | 25.91 | 73.61 | 77.66 | 65.46 | 68.09 | 71.48 | 38.19 | 39.00 | 61.93 |
| | PASER(Spectral_MTEB) | 14.03 | 24.84 | 74.16 | 78.01 | 66.02 | 68.54 | 71.98 | 38.44 | 39.20 | 62.34 |
| | PASER(Spectral_BERT) | 13.68 | 24.31 | 74.48 | 78.21 | 66.32 | 68.79 | 72.28 | **38.75** | 39.30 | 62.59 |
| | PASER(S$^2$RIC) | **13.33** | **23.77** | **74.79** | **78.38** | **66.62** | **69.03** | **72.57** | 38.70 | **39.40** | **62.78** |

## N.3    EXPLORATION ON OTHER POSSIBLE CLUSTERING METHODS

To discuss the impact of different instruction tuning data clustering approaches, we replace our Semantic-structural Recovery Instruction Clustering (S$^2$RIC) module with some other common text clustering method: NMF_TFIDF, LDA_TFIDF, KMeans_TFIDF, Spectral_MTEB, Spectral_BERT (Xu et al., 2024a). The reasoning performance comparison among different PASER versions with such clustering methods is provided in Table 19. From the table, we can find that integrating other instruction clustering methods with PASER can bring the performance decline to some extent among all four pruning schemes. Especially, the clustering method with traditional statistics-based text representation technique, TFIDF, generally behaves worse than semantic embedding-based text representation techniques like BERT. Therefore, we can conclude that our semantic-structural recovery instruction clustering is at least a competitive approach as the clustering component of PASER. Though, comparing these results with those in Table 1, we can observe the advantages of PASER over other general instruction tuning data selection methods can still be stably maintained. This further demonstrates that the potential of the clustering-based data selection for effective and balanced LLM capability recovery.

Table 20: Recovery performance of multiple PASER versions integrated with different capability degradation assessment approaches under various pruning schemes on LLaMA2-7B model. The PASER(JSD) is the version we employ in the main body. The 'bold' represents the best performance under the same pruning scheme. Here, the Alpaca is taken as the original dataset.

| Pruning | Recovery Post-training | WikiText2↓ | PTB↓ | BoolQ | PIQA | HellaSwag | WinoGrande | ARC-e | ARC-c | OBQA | Average |
|---|---|---|---|---|---|---|---|---|---|---|---|
| w/o pruning | w/o Training | 12.62 | 22.14 | 71.13 | 78.40 | 72.79 | 67.17 | 69.36 | 40.70 | 40.80 | 62.91 |
| LLM-Pruner ratio=25% | w/o Training | 20.34 | 38.81 | 61.87 | 76.61 | 65.86 | 60.22 | 63.13 | 37.37 | 39.40 | 57.78 |
| | PASER(KLD) | 16.91 | 27.54 | 66.52 | 76.95 | 68.24 | 65.84 | 66.58 | 38.74 | 39.70 | 60.37 |
| | PASER(WD) | 16.73 | 27.26 | 66.81 | 77.10 | 68.58 | 66.18 | 67.08 | 39.09 | 39.80 | 60.59 |
| | PASER(JSD) | **16.40** | **26.35** | **67.25** | **77.29** | **68.98** | **66.97** | **67.84** | **39.54** | **39.80** | **61.10** |
| SliceGPT ratio=25% | w/o Training | 44.53 | 80.07 | 65.54 | 66.87 | 54.16 | 63.38 | 58.46 | 34.56 | 36.90 | 54.27 |
| | PASER(KLD) | 13.81 | 23.64 | 71.59 | 78.87 | 72.28 | 67.85 | 69.94 | 40.84 | 40.50 | 63.12 |
| | PASER(WD) | 13.10 | 22.75 | 72.08 | 79.25 | 72.93 | 68.39 | 70.48 | 41.26 | 40.90 | 63.61 |
| | PASER(JSD) | **12.24** | **21.53** | **72.75** | **79.84** | **73.92** | **69.18** | **71.37** | **41.82** | **41.30** | **64.31** |
| Wanda sparsity=2:4 | w/o Training | 42.10 | 76.85 | 69.30 | 71.99 | 53.06 | 62.75 | 60.94 | 28.07 | 34.60 | 54.39 |
| | PASER(KLD) | 15.29 | 29.08 | 69.85 | 77.13 | 70.65 | 65.47 | 67.58 | 38.52 | 39.20 | 61.20 |
| | PASER(WD) | 14.64 | 28.21 | 70.42 | 77.54 | 71.23 | 65.86 | 67.95 | 38.78 | 39.70 | 61.64 |
| | PASER(JSD) | **14.13** | **27.22** | 70.77 | **77.87** | **71.78** | **66.26** | **68.30** | **39.04** | 40.10 | **62.02** |
| SparseGPT sparsity=50% | w/o Training | 19.26 | 36.41 | 71.22 | 75.60 | 62.85 | 66.06 | 69.11 | 36.86 | 37.80 | 59.93 |
| | PASER(KLD) | 14.76 | 25.97 | 73.51 | 77.71 | 65.87 | 68.35 | 71.82 | 38.36 | 38.90 | 62.07 |
| | PASER(WD) | 14.21 | 25.18 | 74.15 | 78.05 | 66.24 | 68.68 | 72.19 | 38.53 | 39.10 | 62.42 |
| | PASER(JSD) | **13.33** | **23.77** | **74.79** | **78.38** | **66.62** | **69.03** | **72.57** | **38.70** | **39.40** | **62.78** |

## N.4    EXPLORATION ON OTHER POSSIBLE CAPABILITY DEGRADATION ASSESSMENT METHODS

To further demonstrate the advantages of utilizing Jensen-Shannon divergence (JSD) as the capability degradation assessment approach (elaborated in Section 3.3), we compare it with alternative schemes like Kullback–Leibler divergence (KLD) (Kullback & Leibler, 1951) and Wasserstein distance (WD) (Vaserstein, 1969). As shown in Table 20, JSD consistently outperforms alternative divergence measures across all pruning schemes. While KLD is a widely used dissimilarity measure, its asymmetric nature $(\text{KLD}(P||Q) \neq \text{KLD}(Q||P))$ makes it less suitable for comparing output distributions between original and pruned models, resulting in a 0.73 points average performance drop compared to JSD. The WD performs better than KLD, but still falls 0.51 points short of JSD on average reasoning performance. Although WD can capture gradual distribution shifts that might be missed by other divergence measures, it appears to be more sensitive to outlier probability differences that don't necessarily correlate with capability degradation. This suggests that major shifts in probability mass, rather than fine-grained transportation costs, are more indicative of capability deterioration. JSD's symmetry and boundedness (always between 0 and 1) provide a more stable and interpretable measure of capability divergence, enabling more accurate prioritization of severely affected capabilities. Its ability to balance sensitivity to both major and minor distribution shifts makes it particularly effective for detecting subtle capability degradations that might be missed by other measures. The superior performance of JSD across all four pruning schemes validates our design choice in Section 3.3.

### N.5 EXPLORATION ON OTHER POSSIBLE BUDGET ALLOCATION STRATEGIES

Beyond our proportional allocation strategy in Equation 6, we explore alternative allocation strategies including equal allocation, square-root scaling ($n_k \propto \sqrt{\text{CDS}(c_k)}$), and logarithmic scaling ($n_k \propto \log(1 + \text{CDS}(c_k))$). As shown in Table 21, the linear proportional allocation consistently outperforms alternatives, showing ∼0.4-0.8 points higher average performance across pruning schemes. This validates our design choice while demonstrating that PASER maintains reasonable performance even with different allocation strategies.

Table 21: Recovery performance of different budget allocation strategies under various pruning schemes on LLaMA2-7B model. The Linear proportional is the version we employ in the main body. The 'bold' represents the best performance under the same pruning scheme. Here, the Alpaca is taken as the original dataset.

| Allocation Strategy | LLM-Pruner | SliceGPT | Wanda | SparseGPT |
|---|---|---|---|---|
| Equal | 60.34 | 63.52 | 61.28 | 62.05 |
| Square-root scaling | 60.68 | 63.81 | 61.56 | 62.31 |
| Logarithmic scaling | 60.51 | 63.65 | 61.43 | 62.18 |
| Linear proportional | **61.10** | **64.31** | **62.02** | **62.78** |

## O  FINE-GRAINED DATA SELECTION TIME CONSUMPTION ANALYSIS

In addition to the recovery post-training efficiency analysis provided in the Figure 2 and Section 4.2, we also analyze the fine-grained time consumption of each component to provide a deeper understanding for PASER's efficiency advantages during data selection phase. In the experiments, we follow

Table 22: Time consumption breakdown of PASER components on different-sized datasets.

| Component | Alpaca (52K) | | LaMini (2.58M) | |
|---|---|---|---|---|
| | Time (min) | Percentage | Time (hrs) | Percentage |
| *Semantic-Structural Recovery Instruction Clustering* | | | | |
| SentenceBERT Embedding | 0.63 | 11% | 0.12 | 10% |
| Diffusion Kernel | 1.64 | 28% | 0.36 | 31% |
| NMF Spectral Clustering | 0.82 | 14% | 0.15 | 13% |
| *Capability Degradation-Aware Instruction Selection* | | | | |
| JSD Computation | 0.35 | 6% | 0.07 | 6% |
| Budget Allocation | 0.12 | 2% | 0.02 | 2% |
| *Negative Tuning Effects Mitigation* | | | | |
| Concept Extraction | 1.22 | 21% | 0.26 | 22% |
| CCG Construction | 0.71 | 12% | 0.14 | 12% |
| Consistency Checking | 0.34 | 6% | 0.06 | 5% |
| **Total Selection Time** | 5.83 | 100% | 1.18 | 100% |
| Recovery Training (Full Data) | 83.63 | – | 57.97 | – |
| Recovery Training (PASER) | 26.71 | – | 4.29 | – |
| **Time Saving** | 56.92 | 68.1% | 53.68 | 92.6% |

the standard settings in the main body experiments: |B|=10.4K (20%) for Alpaca and |B|=103.2K (4%) for LaMini. Our component-wise time consumption analysis reveals the relative computational costs across PASER's pipeline, as detailed in Table 22. The semantic-structural clustering process ($S^2$RIC) consumes approximately 54% of the total selection time, with the Diffusion Kernel phase (30%) requiring the most computation in this component due to eigendecomposition operations.

The capability degradation assessment and budget allocation constitute about 8% of the selection time. Here, JSD computation is the most intensive operation (6%), while budget allocation using Equation 6 requires minimal overhead (2%). This aligns with our theoretical analysis, as JSD computation involves calculating probability distributions across the vocabulary space for each token.

The negative tuning effects mitigation component, particularly building and maintaining the Concept Consistency Graph, accounts for 39% of the selection process. Concept extraction (22%) and CCG construction during sample selection (12%) are the key operations. For Alpaca, the complete CCG-based filtering process required around 2.27 minutes but prevented negative tuning effects that would have otherwise caused performance degradation of 0.68-2.39 points across tasks.

For larger datasets like LaMini (2.58M samples), we implemented optimizations to maintain scalability: (1) approximate $k$-nearest neighbors with locality-sensitive hashing for manifold learning, reducing complexity from $O(N^2)$ to $O(N \log N)$; (2) parallel processing for JSD computation across multiple CPU cores; and (3) incremental CCG updates with optimized data structures. With these optimizations, the complete data selection process for LaMini took approximately 1.18 hours, significantly less than the days required for full dataset recovery training.

Despite the overhead of data selection (5.83 minutes for Alpaca and 1.18 hours for LaMini), PASER achieves substantial overall time savings: 68.1% for Alpaca and 92.6% for LaMini compared to full dataset training. These savings stem from the reduced training data volume (20% for Alpaca and 4% for LaMini) and the focus on efficiency-driven sample selection that prioritizes samples with high information-to-computation ratios.

The empirical measurements confirm our theoretical analysis in Section 3.5, with the number of concepts per sample $C$ being relatively small (average of 5-7 concepts per instruction), the dominant time complexity factor remains $O(N \log N)$. This enables PASER to scale efficiently to large instruction tuning datasets while maintaining its performance advantages over conventional recovery approaches.

**Scaling Experiments on Extremely Large Dataset** To further explore PASER's scalability to extremely large datasets, we investigate the online resources and find that the xP3x (Muennighoff et al., 2023) [6] which contains 532 million samples is the largest open-sourced instruction tuning dataset we can download. We conduct experiments on a 8-node GPU cluster, where each node is equipped with dual AMD EPYC 9K84 CPUs (192 cores / 384 threads per node) and 8 NVIDIA H20 GPUs. The fine-grained time consumption on xP3x for each component is as follows: SentenceBERT Embedding: 0.34 hours; Manifold Learing: 0.98 hours; NMF Spectral Clustering: 0.43 hours; JSD Computation: 0.19 hours; Budget Allocation: 0.06 hours; Concept Extraction: 0.65 hours; CCG Construction: 0.39 hours; Consistency Checking: 0.17 hours; Total Selection Time: 3.21 hours; Recovery Training (Full Data): 50+ days (estimated); Recovery Training (PASER): 13.9 hours; Time Saving: $\sim$99%. Such results on nearly billion-scale datasets further validates PASER's time efficiency. In the future, beyond above mentioned efficiency optimization methods, we can also consider applying quantization (FP16/INT8) to embeddings during the clustering phase, and calculating JSD over the top-K logits (or a low-rank logit projection) instead of full logits during the capability degradation assessment.

# P  CASE STUDY FOR RECOVERY INSTRUCTION CLUSTERING

To illustrate the effectiveness of our Semantic-Structural Recovery Instruction Clustering (S$^2$RIC) approach for grouping samples focusing on similar capabilities together, we conduct a case study of clustered instruction samples from the Alpaca dataset. Specifically, we provide representative samples from several obtained clusters as follows.

## P.1  CLUSTER 1: BASIC FACTUAL KNOWLEDGE AND INFORMATION RETRIEVAL

- **Instruction**: "Find the five largest cities in France."

- **Instruction**: "What is the capital of France?"

- **Instruction**: "Find the population density of United States."

These instructions primarily test the model's ability to recall basic facts and information, corresponding to general knowledge capabilities.

---

[6] https://huggingface.co/datasets/CohereLabs/xP3x

## P.2 Cluster 2: Language Understanding and Translation

- **Instruction**: "Translate the word 'giraffe' to French."

- **Instruction**: "Pick the correct Spanish translation of "Hello"."

- **Instruction**: "Difference in meaning between "done freely" and "freely done¨? For instance, is there any difference in meaning between these two sentences?"

This cluster focuses on language-related tasks, including translation, idiomatic expressions, and grammatical analysis.

## P.3 Cluster 3: Logical Reasoning and Problem Solving

- **Instruction**: "A friend shares the following text with you and asks for your opinion: 'Purple-eyed individuals have a stronger psychic connection to the cosmos and have more chances to predict the future.' Analyze the statements and point out logical fallacies or unsupported claims."

- **Instruction**: "Explain how to solve a Sudoku puzzle in three steps."

- **Instruction**: "Answer this math question: What is the value of 3 to the power of 5?"

These instructions test the model's ability to perform mathematical calculations, logical deductions, and pattern recognition.

## P.4 Cluster 4: Creative Writing and Text Generation

- **Instruction**: "Write a microblog post about a recent experience you had."

- **Instruction**: "Compose a haiku about the wonders of technology."

- **Instruction**: "Create an illustration of the inside of a castle."

This cluster groups tasks that require creative text generation, showcasing the model's ability to produce original content across various formats and topics.

## P.5 Cluster 5: Summarization and Information Extraction

- **Instruction**: "Summarize the techniques used for summarizing text."

- **Instruction**: "Extract the main argument from the passage."

- **Instruction**: "Provide a brief summary of the article "A Brief History of the Automobile Industry"."

These instructions focus on the model's capability to condense information and identify key points from longer texts.

This study demonstrates that $S^2$RIC effectively groups instructions targeting similar LLM capabilities, enabling PASER to allocate recovery efforts, i.e., data budget, more strategically. By focusing intensively on clusters where capabilities have degraded most after the pruning, while maintaining awareness of all capability clusters, PASER achieves both targeted and balanced recovery. This dual approach ensures efficient and comprehensive restoration of the model's diverse functionalities, optimizing the recovery process for pruned LLMs.

## Q Visualization for Recovery Instruction Clustering

To better illustrate the semantic relationship of clusters after our Semantic-Structural Recovery Instruction Clustering, we provide the visualization results in the Figure 6. Notably, the numbered samples in each cluster correspond to those in above Appendix P. For example, the instructions numbered as 1,2,3 in *Language Understanding* cluster are the three samples provided in Appendix P.2. From this figure, we can find that our approach successfully identifies meaningful semantic structures in the instruction space that correspond to different LLM capabilities.

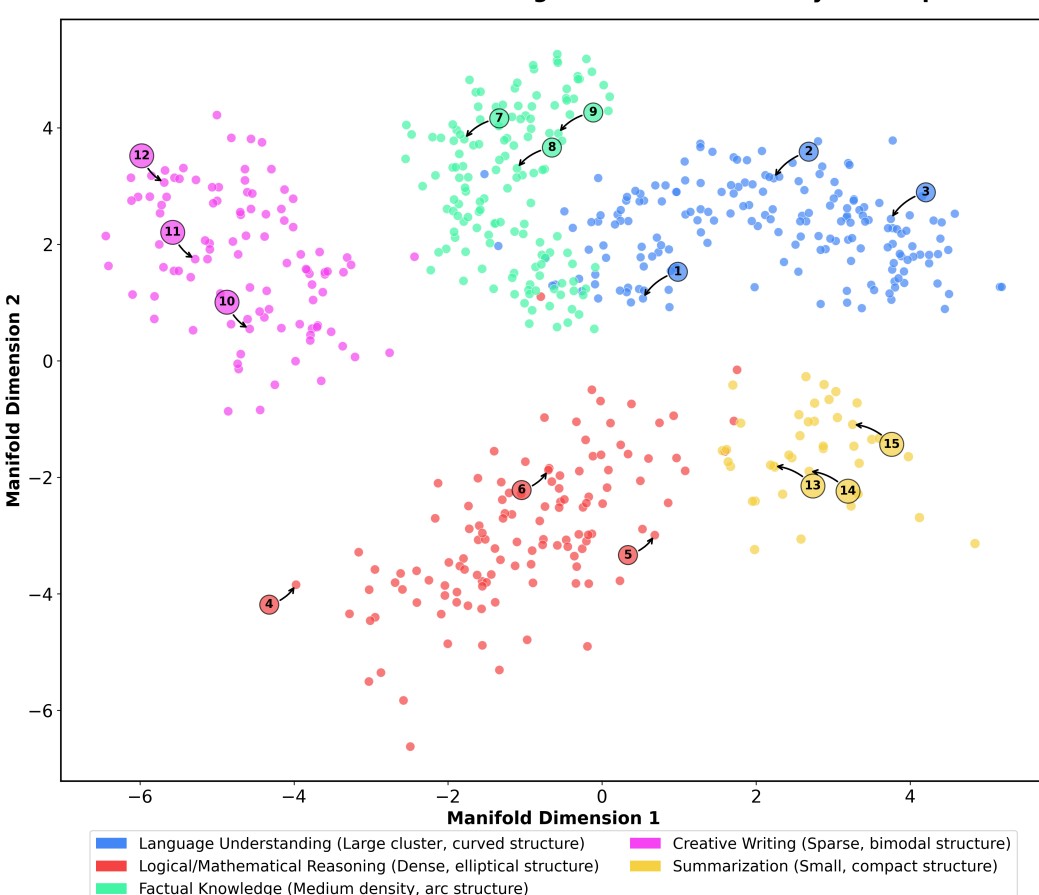

Figure 6: Visualization for the semantic-structural clustering of recovery instruction data by LLM capabilities. For ease of presentation, only the first two manifold dimensions are utilized.

## R    CASE STUDY FOR NEGATIVE TUNING EFFECTS MITIGATION

To illustrate the negative tuning effects mitigation process in PASER, we present the following case study, including both initial concept consistency graph construction and new sample evaluation.

### R.1    CONSTRUCTING INITIAL CONCEPT CONSISTENCY GRAPH

We begin with three accepted instruction-output pairs to construct our initial Concept Consistency Graph (CCG):

1. **Instruction:** "Explain the basic principles of quantum computing."

   **Output:** "Quantum computing is based on the principles of quantum mechanics. It uses quantum bits or qubits, which can exist in multiple states simultaneously due to superposition. Quantum entanglement allows qubits to be correlated with each other. These properties enable quantum computers to perform certain calculations much faster than classical computers for specific problems."

   **Concepts:** {quantum computing, qubit, superposition, entanglement}

2. **Instruction:** "Describe the main components of a classical computer."

   **Output:** "A classical computer typically consists of several key components: the Central Processing Unit (CPU) which performs calculations and controls other components; Random Access Memory (RAM) for temporary data storage; a hard drive for long-term data storage; and various

input/output devices. These components work together to process and store information using binary logic."

**Concepts:** {CPU, RAM, hard drive, binary logic}

3. **Instruction:** "What are the fundamental concepts in deep learning?"

   **Output:** "Deep learning is a subset of machine learning based on artificial neural networks. Key concepts include: neural networks with multiple layers (deep neural networks), which can learn hierarchical representations of data; backpropagation, an algorithm for training these networks by adjusting weights; and various optimization techniques to improve learning. Deep learning has shown remarkable success in tasks such as image and speech recognition."

   **Concepts:** {deep learning, neural network, backpropagation, optimization}

Based on these samples, we construct an initial Concept Consistency Graph (CCG):

$$V = \{\text{quantum computing, qubit, superposition, entanglement,}$$
$$\text{CPU, RAM, hard drive, binary logic,}$$
$$\text{deep learning, neural network, backpropagation, optimization}\}$$

The edges $E$ in the graph represent the co-occurrence of concepts within each instruction-output pair.

$$E = \{(\text{quantum computing, qubit}), (\text{quantum computing, superposition}),$$
$$(\text{quantum computing, entanglement}), (\text{qubit, superposition}),$$
$$(\text{qubit, entanglement}), (\text{superposition, entanglement}),$$
$$(\text{CPU, RAM}), (\text{CPU, hard drive}), (\text{CPU, binary logic}),$$
$$(\text{RAM, hard drive}), (\text{RAM, binary logic}), (\text{hard drive, binary logic}),$$
$$(\text{deep learning, neural network}), (\text{deep learning, backpropagation}),$$
$$(\text{deep learning, optimization}), (\text{neural network, backpropagation}),$$
$$(\text{neural network, optimization}), (\text{backpropagation, optimization})\}$$

This explicit definition of E clearly shows the relationships between concepts within each domain, while highlighting the absence of edges between concepts from different domains.

## R.2 EVALUATING NEW SAMPLES

We now evaluate two new samples using PASER's negative tuning effects mitigation process.

### R.2.1 SAMPLE 1: ACCEPTED

**Instruction:** "How does increasing the number of qubits affect a quantum computer's performance?"

**Output:** "Increasing the number of qubits in a quantum computer can significantly enhance its performance and capabilities:

1. Exponential growth in computational power: Each additional qubit doubles the number of quantum states that can be represented, leading to an exponential increase in computational power.

2. Handling more complex problems: More qubits allow quantum computers to tackle larger and more complex problems that are intractable for classical computers.

3. Improved error correction: A larger number of qubits can be used to implement more robust quantum error correction codes, improving the overall stability and reliability of quantum computations.

4. Enhanced quantum parallelism: More qubits enable greater parallelism in quantum algorithms, potentially leading to faster solution times for certain problems.

5. Challenges in scalability: However, increasing the number of qubits also introduces challenges in maintaining coherence and reducing error rates, which are active areas of research in quantum computing.

In summary, while increasing the number of qubits can dramatically improve a quantum computer's capabilities, it also comes with technical challenges that researchers are working to overcome."

**Concepts:** {qubit, quantum computer, computational power, quantum states, error correction, quantum parallelism, coherence}

Applying PASER's negative tuning effects mitigation process:

1. **Concept Extraction:** $C(x, y) = \{$qubit, quantum computer, computational power, quantum states, error correction, quantum parallelism, coherence$\}$

2. **Consistency Check:** For each pair $(v_i, v_j) \in C(x, y)$:
   - (qubit, quantum computer): both $\in V$ and $(v_i, v_j) \in E$
   - (qubit, computational power): qubit $\in V$, computational power $\notin V$
   - (qubit, quantum states): qubit $\in V$, quantum states $\notin V$
   - Other pairs: at least one concept $\notin V$

3. **Decision:** IsConsistent(x, y) = True, because for all pairs, either $(v_i, v_j) \in E$ or $\{v_i, v_j\} \not\subset V$

PASER accepts this sample as it is consistent with the existing CCG and introduces relevant new concepts without conflicts.

### R.2.2 SAMPLE 2: REJECTED

**Instruction:** "Explain how quantum computing can be used to significantly speed up deep learning algorithms."

**Output:** "Quantum computing has the potential to speed up certain aspects of deep learning algorithms. Quantum algorithms could potentially optimize the training process of neural networks, especially in handling large-dimensional data. However, it's important to note that the exact advantages and applications are still being researched, and classical computers remain more practical for most current deep learning tasks."

**Concepts:** {quantum computing, deep learning, neural network, speedup}

Applying PASER's negative tuning effects mitigation process:

1. **Concept Extraction:** $C(x, y) = \{$quantum computing, deep learning, neural network, speedup$\}$

2. **Consistency Check:** For each pair $(v_i, v_j) \in C(x, y)$:
   - (quantum computing, deep learning): both $\in V$, but $(v_i, v_j) \notin E$
   - (quantum computing, neural network): both $\in V$, but $(v_i, v_j) \notin E$
   - (deep learning, neural network): both $\in V$ and $(v_i, v_j) \in E$
   - (speedup, any other concept): speedup $\notin V$

3. **Decision:** IsConsistent(x, y) = False, because the pairs (quantum computing, deep learning) and (quantum computing, neural network) have both concepts in V, but these edges do not exist in E. This introduces new relationships between existing concepts that are not present in the current CCG.

PASER rejects this sample because it introduces direct relationships between quantum computing and deep learning/neural networks, which were not present in the initial CCG. While these concepts existed separately in the CCG, their combination in this context could lead to potential misunderstandings or oversimplifications about the current state and capabilities of quantum computing in machine learning.

### R.3 CONCLUSION

This case study demonstrates PASER's negative tuning effects mitigation process in action. By accepting Sample 1, PASER allows for the introduction of new, relevant concepts that expand the concept consistency graph without introducing conflicts. By rejecting Sample 2, PASER prevents the introduction of potentially misleading relationships between existing concepts from different domains, thus mitigating the risk of negative tuning effects during the recovery process.

## S    ANALYSIS ON PRUNING SEVERITY AND PASER RECOVERABILITY

This section provides analysis and clarification on how different pruning strategies affect model recoverability and how PASER interacts with such degradation patterns.

**Capability-Specific Impact of Different Pruning Strategies** Pruning does not degrade all capabilities uniformly. As discussed in the main text and Figure 4, unstructured pruning typically leads to diffuse and moderate degradation, leaving many capabilities at least partially recoverable. In contrast, semi-structured and structured pruning introduce more localized and capability-selective damage, often disproportionately affecting reasoning-heavy or composition-heavy capabilities. In some architectures, structured pruning (e.g., LLM-Pruner) or high-ratio pruning may disrupt functional pathways, such as attention-head specialization or layerwise routing, thereby causing structural degradation rather than a simple reduction in parameter count.

**Recoverable vs. Irrecoverable Capability Regions** PASER focuses recovery efforts on the parts of the pruned model that remain expressively intact. Capabilities whose representational support is degraded but still present tend to respond well to targeted recovery through our capability-aware data selection. By contrast, capabilities whose computational pathways have been effectively removed by pruning exhibit limited improvement regardless of the quality or quantity of recovery data. This distinction explains why PASER produces substantial gains in most settings but cannot fully restore capability clusters whose underlying structure has been fundamentally pruned away.

**Interaction Between PASER's Pruning-Aware Design and Model Degradation** PASER's pruning-aware design further shapes its interaction with different levels of model degradation. As described in Sections 3.3 and 3.4, PASER identifies capability clusters that suffer more severe pruning-induced degradation and allocates recovery data accordingly. It filters conflicting or harmful samples that could exacerbate uneven degradation during fine-tuning, and it avoids over-training intact clusters, which helps preserve the remaining expressivity of the pruned model. These mechanisms enable PASER to remain effective across a range of pruning strategies, even when the degradation is substantial or uneven. Nevertheless, PASER necessarily operates within the recoverable subspace: once key structural representations are removed by pruning, recovery naturally plateaus. This limitation is inherent to all post-training methods, not specific to our approach.

**Why Baichuan2-7B Shows Weaker Recovery in Some Cases** The weaker recovery observed in Baichuan2-7B (as shown in Table 2) can be understood in this context. Further inspection suggests that Baichuan2-7B contains more functionally specialized attention heads and feedforward pathways, and some of its reasoning-related components exhibit lower redundancy compared to models such as LLaMA or Qwen. As a result, structured pruning causes more irreversible representational loss in this architecture, and the affected capability clusters become inherently harder to restore. Structural pruning therefore removes more of the functional backbone needed for downstream recovery, limiting the improvements achievable during post-training.

In summary, pruning severity sets the upper bound on the recoverability of a pruned model. PASER leverages pruning-aware signals to maximize recovery within the expressive capacity that remains, but structural representational loss cannot be reconstructed through post-training alone. Architectural differences further influence this recoverability boundary, helping to explain why some model families are more sensitive to structured pruning than others.

## T    IMPLEMENTATION DETAILS AND HYPERPARAMETER SETTINGS

Most of experiments are conducted on the server with $8 \times$ NVIDIA RTX 6000 Ada GPUs. Part of experiments on 70B model and MOE models are conducted on NVIDIA H800 Tensor Core GPU cluster. For the Semantic-Structural Recovery Instruction Clustering, we use consistent settings across all experiments: diffusion time $t$ is automatically selected using the spectral gap method, and the embedding dimension $d$ is set to 16. The optimal number of clusters $K$ is determined adaptively through NMF approximation error minimization, typically resulting in 8-12 clusters for Alpaca and 15-20 clusters for LaMini. For the JSD calculation in capability degradation score, we use a temperature $\tau$=1.0 for the output probability distribution. The computational cost is approximated using the quadratic term of sequence length with a coefficient of 1.0 across all experiments. For concept extraction, we use a maximum of 10 concepts per instruction-response pair with a minimum

phrase length of 2 words and a maximum of 4 words. The concept similarity threshold for consistency checking is set to 0.75 across all experiments. We maintain these same hyperparameter settings across all models and pruning schemes to ensure fair comparison. During the recovery post-training phase, we take the the low-rank approximation, LoRA (Hu et al.), to improve the efficiency. The corresponding hyperparameters are set as following: rank=8, batch size=64, epochs=2, learning rate = 1e-4 (Alpaca series experiments), 5e-5 (LaMini series experiments). As for the structured pruning, we set the pruning ratio as 25% for LLaMA2-7B/LLaMA3-8B/Baichuan2-7B and 50% for LLaMA2-13B/LLaMA2-70B/Baichuan-13B models. For the other two kinds of pruning schemes, we follow the previous work (Frantar & Alistarh, 2023). Specifically, we adopt the 2:4 semi-structured sparsity patterns and implement 50% unstructured weight sparsity. Except the experiments for recovery post-training efficiency analysis, we set the ratio of recovery data budget $B$ to original dataset size $N$ as 20% for Alpaca and 4% for LaMini. As for the implementation of concept extraction in Section 3.4, we use the open-source library `rake-nltk` [7]. To ensure statistical robustness, all the results reported in this paper are the averages of five runs with different seeds. Statistical significance is also assessed using two-tailed independent t-tests, with results considered significant when $p < 0.01$. In our experiments, we find all our reported results for PASER can pass this statistical significance test. For facilitating the reproduction of our work, we provide the code in the supplementary materials, also seen in Github `https://github.com/BokwaiHo/PASER`.

## U  LIMITATION ANALYSIS

While PASER demonstrates significant improvements in recovery performance and efficiency for pruned large language models, there are several limitations to consider:

- **Computational overhead:** Although PASER reduces the recovery training time, the initial clustering and data selection process introduces some computational overhead. For very large instruction tuning datasets, this overhead may become non-trivial.

- **Dependence on initial pruning quality:** The effectiveness of PASER may vary depending on the quality and method of the initial model pruning. Poorly pruned models might not benefit as much from the targeted recovery approach.

- **Potential bias in capability recovery:** While PASER aims for balanced capability recovery, there might still be some bias towards certain capabilities based on the initial clustering results and the composition of the instruction tuning dataset.

- **Scalability to extremely large models:** The paper primarily demonstrates results on models up to 70B parameters. The scalability and effectiveness of PASER on even larger models (e.g., 100B+ parameters) need further investigation.

- **Long-term Stability:** The long-term stability of models recovered using PASER, especially under continued fine-tuning or adaptation, has not been thoroughly examined in this work.

**Limitations of Concept Consistency Graph.** In addition to the above analysis, we further discuss the concept consistency graph's potential limitations. Indeed, while CCG helps mitigate negative tuning effect, we acknowledge there are scenarios where semantic conflicts might not be fully captured:

- **Cross-domain Knowledge Integration:** When instructions involve integrating knowledge from multiple distinct domains, CCG might miss subtle conflicts in their interactions. For example, when concepts from physics and biology are combined in interdisciplinary problems, their complex relationships and potential incompatibilities may not be fully reflected in simple co-occurrence patterns.

- **Context-dependent Semantics:** The same concept pairs might have different relationships depending on context. For instance, terms like "positive" and "negative" could be contradictory in sentiment analysis but complementary in mathematics, making it challenging for CCG to maintain consistent concept relationships across different contexts.

- **Temporal or Version-specific Conflicts:** In rapidly evolving domains like technology or scientific research, concept relationships might change over time. An instruction about "state-of-the-art

---

[7]`https://pypi.org/project/rake-nltk/`

performance" or "current best practices" could contain outdated or conflicting information that is not immediately apparent from concept co-occurrence analysis.

- **Nuanced Conceptual Dependencies:** When instructions involve subtle logical dependencies or conditional relationships between concepts, the binary edge representation in CCG might not fully capture these complex interactions. This is particularly evident in reasoning tasks where conclusions depend on specific combinations of conditions.

Our results acknowledge these inherent limitations while demonstrating CCG's overall effectiveness in practical applications.

# V    DECLARATION OF LLM USAGE

We used a general-purpose large language model assistant solely for language editing (grammar polishing, minor rephrasing, and shortening) and for non-substantive code scaffolding (e.g., generating boilerplate for plotting or CLI argument parsing). All research ideas, algorithmic designs (PASER), theoretical analyses, experiment setups, and reported results were conceived, implemented, and verified by the authors. No figures, tables, proofs, or claims were generated by an LLM without human verification, and no conclusions rely on unverifiable LLM outputs. No new training or evaluation data were created with an LLM for this work. We rely on publicly available instruction-tuning datasets (e.g., Alpaca and LaMini), whose responses were originally synthesized by third-party LLMs as documented by their providers; we used them as-is without further LLM-based modification. The assistant was not used to generate novel technical content unique to this paper (e.g., PASER's methodology, equations, or ablation study design). All code that implements PASER's selection pipeline and all experimental decisions were written and reviewed by the authors. The LLM is not an author and bears no responsibility for the work; accountability lies with the human authors. When LLM suggestions influenced wording, the authors performed fact-checking and ensured compliance with dataset licenses and double-blind review requirements

