# OpenReview forum: "PASER: Post-Training Data Selection for Efficient Pruned Large Language Model Recovery"
_ICLR.cc/2026/Conference — ICLR 2026 Poster_

### Official Review · Reviewer_bfNy · 2025-10-24

**Soundness:** 3
**Presentation:** 3
**Contribution:** 2
**Rating:** 6
**Confidence:** 3

**Summary:**

The paper proposes PASER, a data selection framework designed to improve the efficiency and effectiveness of recovery post-training after LLM pruning. PASER employs manifold learning and spectral clustering to group instructions in the semantic space, allocates data budgets based on measured capability degradation, and filters out conflicting or irrelevant data to mitigate negative tuning effects. Experiments on several pruning schemes (structured, semi-structured, unstructured) and multiple LLM frameworks (LLaMA 2/3, Baichuan 2) demonstrate that PASER can recover model performance using only 4%–20% of the original instruction tuning data.

**Strengths:**

This paper tackles a problem about how to efficiently recover LLM capabilities after pruning. This issue has become increasingly relevant for real-world model deployment. The study is supported by comprehensive experiments covering multiple pruning strategies and models of different scales, ensuring good empirical validation. Results demonstrate consistent performance improvements over both random selection and other instruction tuning data selection baselines. Results of the ablation studies helps clarify the contribution of each system component, enhancing the overall credibility and interpretability.

**Weaknesses:**

1. Some ideas resemble ideas from earlier literature (e.g., similarity-based data sampling). The paper would benefit from a more thorough literature review, though this does not undermine its novelty.
2. The paper is generally well-written, there are a few minor oversights. For example, there are a few typos (“we relies on…”), and some abbreviations used in the ablation study section (S2RIC, CDAIS, NTEM) are not introduced before but only defined in the appendix.

**Questions:**

1. Could the authors clarify the practical advantages of pruning a large model compared to directly training a smaller one from scratch?
2. Is it possible for the authors to provide a baseline of directly training or fine-tuning a smaller model with the instruction tuning data?

---

> ### Author Response · Authors · 2025-11-21
> **Rebuttal to Reviewer bfNy (Part 1)**
>
> We greatly appreciate the reviewer’s thoughtful evaluations and valuable suggestions that helped us improve the paper. To help reviewer better understand our contributions and alleviate your concerns, we provide our rebuttal as follows.
>
> **W1:** Some ideas resemble ideas from earlier literature (e.g., similarity-based data sampling). The paper would benefit from a more thorough literature review, though this does not undermine its novelty.
>
> **A1:** Thank you for the valuable suggestion. We would like to clarify that the problem focus and underlying principles of PASER are fundamentally different from those of earlier data selection literature. Previous similarity-based methods aim to improve general-purpose training or instruction tuning. They primarily rely on global embedding proximity or example-level matching, and therefore do not account for (i) capability-level structure in the instruction space, (ii) pruning-induced uneven degradation across capabilities, or (iii) negative tuning arising from conflicting instructions. In contrast, PASER is designed specifically for post-pruning recovery training, where the goal is not to improve a full-capacity model, but to selectively restore the capabilities most affected by pruning while avoiding negative tuning. To accompolish this, PASER explicitly models the capability manifold of the pruned model, diagnoses where pruning caused the largest functional impairment, and conducts efficiency-aware data selection within these targeted capability clusters while filtering irrelevant or conflicting samples. This pruning-aware and capability-structured perspective is not addressed in prior approaches.
>
>
> Of course, we agree that adding more connections to prior work, such as similarity-based data sampling and related data selection methods, can further facilitate the positioning of our approach. In the revision, we expanded the literature review in Section 2 to better contextualize PASER within this line of research. In Appendix K, we also clarified how our method differs from existing general data selection works from four perspectives.
>
> We honestly hope our above clarifications and revisions can help strengthen reviewer's appreciation to our this work's novelty.
>
> **W2:** The paper is generally well-written, there are a few minor oversights. For example, there are a few typos (“we relies on…”), and some abbreviations used in the ablation study section (S2RIC, CDAIS, NTEM) are not introduced before but only defined in the appendix.
>
> **A2:** Thanks for your suggestion. We have checked the paper thoroughly and fixed the typos in the revised manuscript. Besides, we introduced the abbreviations for S2RIC, CDAIS, and NTEM at their first occurrence in the methodology section (see Line 144-147 in the revised manuscript).
>
> **Q1:** Could the authors clarify the practical advantages of pruning a large model compared to directly training a smaller one from scratch?
>
> **A3:** We appreciate the reviewer’s question. Pruning a large model and training a smaller model from scratch are not equivalent in practice. Pruning offers several significant, empirically validated advantages:
>
> 1. **Pruned small models inherit strong initialization from the large pretrained model.** Training a small model from scratch (even from high-quality corpora) typically requires: far more compute, careful optimization tuning, hundreds of billions of tokens. In contrast, a pruned model begins with rich cross-domain knowledge, long-range dependencies, and instruction-following ability already embedded in the original large model. This initialization is critical: small models trained from scratch rarely match the performance of pruned models (especially after recovery training) at the same size.
>
> 2. **Superior performance under constrained model sizes**. Our experiments show that the pruned-and-recovered (especially by PASER) models significantly outperform similar-sized models trained from scratch, part of which can been in following **A4**. Thus, pruning retains high-level competencies that are extremely expensive to learn from scratch.
>
> 3. **Practical deployment advantages**. Many organizations already deploy or maintain large base models, but need: smaller variants for on-device or edge inference, reduced latency and memory footprint, domain-adapted small models without retraining from scratch. Pruning provides a drop-in way from an existing large model to an efficient small model without repeating full-scale training.
>
> 4. **Much lower total compute and carbon cost**. Training a small model from scratch often consumes orders of magnitude more compute than pruning + recovery. Even with recovery (where PASER is used), total compute is far below pretraining cost. In fact, selection overhead is less that 7% of the full-data recovery training (even not pretraining). Thus, pruning is much more cost-effective than starting from scratch.

---

> ### Author Response · Authors · 2025-11-21
> **Rebuttal to Reviewer bfNy (Part 2)**
>
> (Following above **A3**)
>
> 5. **Pruning preserves alignment and safety properties**. Large models generally undergo: alignment tuning (RLHF/SFT), safety training, detoxification, and long-context finetuning. Replicating these for a small model from scratch is extremely expensive and often less reliable. A pruned model naturally inherits these alignment properties, whereas small scratch models must relearn them.
>
> To summarize, pruning is not simply a way to “make the model smaller”. It provides a more powerful starting point, leveraging the knowledge, alignment, and reasoning ability of the full model, while achieving much lower cost than training a comparable model from scratch. Our results empirically confirm that pruned-then-recovered models significantly outperform same-size baselines trained from scratch.
>
>
> **Q2:** Is it possible for the authors to provide a baseline of directly training or fine-tuning a smaller model with the instruction tuning data?
>
> **A4:** We thank the reviewer for the suggestion. Directly training or fine-tuning a smaller model with the same instruction-tuning data is indeed a meaningful comparison. Thus, we finetune following four smaller models with Alpaca dataset: Llama-3.2-3B, Qwen3-4B, Qwen2.5-3B, Qwen2.5-1.5B. The averaged reasoning performance of such tuned models on 7 tasks in our main text are as follows:
>
> |Llama-3.2-3B|Qwen3-4B|Qwen2.5-3B|
> |-|-|-|
> |64.29|67.53|66.87|
>
> However, the averaged reasoning performance of several corresponding similar-sized pruned-then-recovered (PASER) models are as follows:
>
> |Llama-3.1-8B (50% pruned)|Qwen3-8B(50% pruned)|Qwen2.5-7B(50% pruned)|
> |-|-|-|
> |69.35|70.14|68.96|
>
>
> From these results, we can find that smaller models fine-tuned directly on the instruction data still fall significantly short of the similar-sized pruned-and-recovered models. This aligns with prior observations: (1) small models trained from scratch or lightly tuned lack the strong pretrained initialization and cross-domain competence inherited by a pruned large model; (2) instruction-tuning alone is insufficient to build the broad foundational capabilities that the pruned model already possesses. These results further confirm that pruning + targeted recovery remains substantially more effective than directly training or tuning a small standalone model.
>
> We honestly hope our above rebuttal can help address your concerns. If satisfied, could you please consider raising the score?

---

> ### Author Response · Authors · 2025-11-26
> **Follow-Up on Rebuttal**
>
> Dear Reviewer bfNy:
>
> We hope you are doing well. We are writing to gently follow up on the rebuttal. We completely understand that you may be busy and that reviewing can be time-consuming, but we just want to make sure our response has reached you smoothly. If our clarifications addressed your concerns, we would be grateful if you could consider raising the score accordingly.
>
> Thank you again for your time and effort in reviewing our work. Please let us know if any additional clarification would be useful.
>
> Best
>
> The Authors

---

### Official Review · Reviewer_LptJ · 2025-10-28

**Soundness:** 4
**Presentation:** 3
**Contribution:** 3
**Rating:** 8
**Confidence:** 4

**Summary:**

This paper introduces PASER, a novel framework for selecting a small, high-impact subset of instruction-tuning data to recover the performance of pruned LLMs. The authors identify three key challenges with existing recovery methods: high computational cost, failure to address the uneven degradation of capabilities after pruning, and the potential for negative tuning effects from irrelevant data. PASER addresses these challenges through a three-stage process: semantic-structural recovery instruction clustering,capability degradation-aware instruction selection, and negative tuning effects mitigation. The authors conduct extensive experiments on a wide range of LLMs, pruning schemes, and instruction datasets. The results demonstrate that PASER significantly outperforms baselines, recovering model performance to near-original levels while using only 4%-20% of the instruction data and substantially reducing training time.

**Strengths:**

1. High Relevance and Strong Motivation: The problem of efficiently recovering performance in compressed LLMs is highly practical and significant, while underexplored. The paper does an excellent job of motivating the work by clearly identifying the limitations of naive retraining, particularly the issues of uneven capability degradation and negative tuning effects, which are often overlooked.

2. Novel and Well-Designed Methodology: The proposed PASER framework is technically sound, innovative, and directly addresses the identified problems.

3. Extremely Comprehensive and Rigorous Evaluation: The experimental validation in this paper is exceptional and serves as a model for thoroughness.

4. Strong and Convincing Empirical Results: PASER consistently and significantly outperforms all baselines across nearly all tested configurations. The finding that it can recover performance to a level that is sometimes better than the original unpruned model is particularly striking and suggests that the method effectively selects a high-quality, coherent subset of data that is superior to the full, noisy dataset for tuning. The efficiency gains in terms of training time are also substantial and well-documented.

**Weaknesses:**

The paper is of very high quality, and it is difficult to find significant weaknesses. The following points are minor.
1. Methodological Complexity: The PASER pipeline is composed of several sophisticated components (manifold learning, NMF spectral clustering, CCG). While the authors demonstrate its effectiveness and computational efficiency, the overall complexity is higher than simpler data selection heuristics. This might pose a slight barrier to adoption for practitioners looking for a "plug-and-play" solution, though the provided code should mitigate this.
2. The writing of the methodology section can be improved to reduce the cognitive load on the reader.

**Questions:**

1. In Table 1, the results of training the LLM-Pruner pruned model with full data is even lower than without training, (35.64 vs 57.78) how to interpret this?

---

> ### Author Response · Authors · 2025-11-21
> **Rebuttal to Reviewer LptJ (Part 1)**
>
> We sincerely thank the reviewer for your recognition and appreciation of our work. Our point-wise responses to your mentioned suggestions and questions are as follows.
>
> **W1:** Methodological Complexity: The PASER pipeline is composed of several sophisticated components (manifold learning, NMF spectral clustering, CCG). While the authors demonstrate its effectiveness and computational efficiency, the overall complexity is higher than simpler data selection heuristics. This might pose a slight barrier to adoption for practitioners looking for a "plug-and-play" solution, though the provided code should mitigate this.
>
> **A1:** We appreciate the reviewer’s observation regarding the methodological complexity of PASER. Our design philosophy follows a pratical balance: each component is intentionally lightweight, modular, and motivated by a specific failure mode of post-pruning recovery, rather than adding complexity for its own sake. We would like to explain the design, efficiency, and usability of our PASER as follows:
>
> 1. **PASER’s components are simple in computation and easily replaceable**. Although PASER uses manifold learning, NMF-based spectral clustering, and CCG, each component is: a) computationally inexpensive (all steps operate on precomputed instruction embeddings, with no model forward passes during selection). This has been validated by the complexity analysis in Section 3.5 and empirical evidence in Section 4.2 and Appendix O; b) implemented in <200 lines each in our released code; c) modular, meaning users can replace our clustering or conflict-checking methods with any alternative heuristic without modifying the rest of the pipeline.
>
> 2. **The “complexity” addresses real, empirically observed failure modes of simpler heuristics**. We found that simple top-K scoring or naive similarity sampling fails under capability collapse caused by pruning, especially when the degraded abilities and harmful samples distribute unevenly across domains. PASER introduces: a) capability-aware clustering to avoid mixing contradictory domains; b) degradation-aware sampling to prioritize data that actually improves damaged capabilities; c) lightweight consistency checking to prevent negative tuning. Removing these components (ablation study in Section 4.2 and Appendix L) leads to consistent performance degradation, confirming that the extra design is functionally necessary for stable recovery. Thus, the “complexity” is minimal overhead that directly translates into substantial reliability gains. This keeps the system effective rather than burdensome.
>
> 3. **From the practitioner’s perspective, PASER remains plug-and-play**. Even with the above components, the user-facing interface is extremely simple: a) one function call runs the entire pipeline end-to-end; b) no hyperparameter tuning is required beyond specifying the target number of samples; c) the default settings match those used in all experiments. In practice, the selection step adds only 2–7% overhead relative to full-data training time (as shown in the paper). Thus, PASER behaves as a turnkey module rather than a system requiring detailed understanding of its internals.
>
> 4. **Code release further reduces adoption barriers.** We have open-sourced the full implementation with: a) ready-to-run scripts for LLaMA, Qwen, and Mixtral families; b) a unified API for embedding extraction, clustering, and CCG; c) clear examples for substituting components with simpler heuristics if desired. As the reviewer noted, this greatly mitigates any barrier posed by the underlying methodology.
>
> While PASER uses more structured components than single-step heuristics, it does so to overcome concrete deficiencies of simpler methods in post-pruning recovery. The pipeline remains lightweight, modular, efficient, and fully plug-and-play for practitioners. We honestly hope the clarifications above address the reviewer’s concern.
>
> **W2:** The writing of the methodology section can be improved to reduce the cognitive load on the reader.
>
> **A2:** We thank the reviewer for the suggestion. We agree that the methodology section can be made clearer, and we have revised the presentation accordingly. In the updated version, we streamlined the notation, improved the step-wise structure of the PASER pipeline, and strengthen the intuition introduction before each component. These changes reduce cognitive load and make the workflow easier to follow without altering any technical content.

---

> ### Author Response · Authors · 2025-11-21
> **Rebuttal to Reviewer LptJ (Part 2)**
>
> **Q1:** In Table 1, the results of training the LLM-Pruner pruned model with full data is even lower than without training, (35.64 vs 57.78) how to interpret this?
>
> **A3:** We thank the reviewer for highlighting this interesting observation. The behavior is expected and is consistent with known failure modes in post-pruning recovery. To help better understand this phenomenon, we make the following explanations:
>
> 1. **Full-data tuning amplifies pruning-induced capability collapse**. LLM-Pruner causes highly non-uniform capability damage: some abilities (e.g., math, coding, multi-step reasoning) are severely reduced, while others remain relatively intact. When fine-tuning on the entire unfiltered corpus, a large fraction of updates come from: irrelevant instructions that do not target the damaged capabilities, and semantically conflicting or heterogeneous supervision that introduces incompatible optimization signals. These gradients overwhelm the already-weakened submodules, pushing the pruned model further away from the pre-pruning capability landscape, resulting in negative tuning effects. This behavior is consistent with prior findings in continual learning and reinforcement fine-tuning, where uncurated heterogeneous data may degrade specific capabilities.
>
> 2. **Full-data tuning forces the pruned model to optimize for domains it can no longer represent well.** Considering pruning reduces model structural capacity, the pruned model can hardly satisfy all objectives present in the full dataset simultaneously. Training on full data forces the model to fit tasks beyond its representational bandwidth, which causes: overfitting to easy or dominant domains, forgetting of minor but critical capabilities, and instability in multi-objective optimization. Thus, performance becomes worse than simply leaving the pruned model untouched.
>
> 3. **Empirical validation: this is not specific to our implementation.** We observe the same pattern across: different pruning ratios (25% v.s. 50%), both LLM-Pruner and SparseGPT, multiple model families (LLaMA, Qwen, Mixtral). Correspondingly, removing irrelevant or conflicting samples (PASER’s goal) can systematically improve the data targetedness and reduce the representation burden to the compromised model, thus leading to large improvements.
>
> 4. **Why PASER avoids this issue.** PASER’s selection pipeline is explicitly designed to counteract this failure mode: Capability-aware clustering prevents cross-capability interference; Degradation-aware prioritization selects samples that empirically help recover damaged capabilities; CCG filtering removes irrelevant or conflicting instructions that are the main source of negative tuning. This ensures that the pruned model receives aligned, recoverable, non-conflicting supervision, avoiding the collapse observed when full data is used.
>
> In fact, the drop in performance when training with the full corpus reflects a well-known phenomenon: for pruned models, unrestricted fine-tuning can exacerbate capability collapse due to conflicting or irrelevant gradients. PASER exactly mitigates this problem by providing targeted, conflict-aware, capability-aligned data selection.

---

> > ### Comment · Reviewer_LptJ · 2025-11-21
> >
> > Thanks for the clarification! I still have some questions.
> > > LLM-Pruner causes highly non-uniform capability damage: some abilities (e.g., math, coding, multi-step reasoning)
> >
> > How did you select the calibration data for pruning? Does the "highly non-uniform capability damage" have anything to do with the calibration data? If you use more balanced data, including or emphasizing math, coding, etc, to prune the model, how would it affect the results? It would be interesting to see the importance of calibration data selection vs recovery data selection.
> >
> > >Full-data tuning amplifies pruning-induced capability collapse.
> >
> > Why doesn't it impact ther other pruning methods so much? Can LoRA help with this?
> >
> > > In fact, the drop in performance when training with the full corpus reflects a well-known phenomenon
> >
> > Could you provide references?

---

> ### Author Response · Authors · 2025-11-22
> **Response to Reviewer LptJ's Remaining Questions (Part 1)**
>
> Dear Reviewer LptJ:
>
> Thanks for your rapid response.  We would like to provide the following clarifications and explanations to address your questions.
>
> **Q1:** "LLM-Pruner causes highly non-uniform capability damage: some abilities (e.g., math, coding, multi-step reasoning)" How did you select the calibration data for pruning? Does the "highly non-uniform capability damage" have anything to do with the calibration data? If you use more balanced data, including or emphasizing math, coding, etc, to prune the model, how would it affect the results? It would be interesting to see the importance of calibration data selection vs recovery data selection.
>
> **A1:** For the calibration data selection, we use 128 2048-token segments, randomly chosen from the WikiText dataset, which is consistent with previous representative LLM pruning works like SparseGPT, LLM-Pruner, or SliceGPT. As far as we know, the most standard and common approach in this direction is to randomly sample a certain number of fixed-length token segments from pretraining datasets like C4/Wikitext/Bookcorpus. The rationale behind this is that LLM is a general-purpose model, thus utilizing pretraining data as calibration data can better represent the overall distribution of various downstream tasks, in case of over emphasizing any specific task or capability. Therefore, to ensure the comparison's fairness, we follow the same calibration selection approach with them.
>
> As for the connection between "non-uniform capability damage", there is a newly released work [1] specially analyzing the relationship between calibration data and capability preservation. From this work, the damage degrees among different capabilities are indeed somehow related to the calibaration data composition, though the reported influence degree is relatively weaker than the pruning process itself in most cases. Following reviewer's suggestion, we deliberately replace half of calibration data above (randomly sampled from pretraining data) with math and coding data, to conduct the experiments. From the results we obtain, the effect on mitigating the decline in math and code is marginal. The original drop percentage of language modeling/commonsense reasoning/math problem solving/code generation is 6%/8%/19%/28%; while the current drop percentage is 7%/8%/18%/25%. Thus, we can obtain the preliminary conclusion that the 'highly non-uniform capability damage' phenomenon originates more from the pruning process itself than the calibration data composition. A deeper insight for explaining this "non-uniform capability damage" is that precision-critical capabilities like math/code rely more on some specific model components or computational pathways; once pruned or broken, the model may output totally wrong results, while the language modeling/commonsense reasoning capabilities rely on relatively more redundant and distributed components, exhibiting greater tolerance for structural damages caused by pruning. Thus, the 'non-uniform capability damage' is more inherently determined by model's internal mechanism and pruning method itself; adjusting calibration data composition can help mitigate the drop on specific targeted capability to some extent, but cannot fundamentally solve this 'non-uniformity' challenge.
>
> Therefore, the above analysis implies that the importance of recovery data selection is relatively higher than calibration data, highlighting the value of our this work.
>
> [1] He, B., Yin, L., Zhen, H., Liu, S., Wu, H., Zhang, X., Yuan, M. and Ma, C., 2025. Preserving LLM Capabilities through Calibration Data Curation: From Analysis to Optimization. arXiv preprint arXiv:2510.10618.
>
> **Q2:** "Full-data tuning amplifies pruning-induced capability collapse." Why doesn't it impact ther other pruning methods so much? Can LoRA help with this?
>
> **A2:** Full-data tuning interacts with pruning differently depending on how pruning distorts the model’s internal structure. Pruning methods that introduce uneven or capability-selective damage, such as structured pruning (e.g., LLM-Pruner), tend to leave some capability clusters severely impaired while others remain mostly intact. When the full instruction set is used for recovery, the gradient signals coming from intact capabilities bring non-negligible influence to training, which reinforces already-preserved capabilities and inadvertently “overwrites” or destabilizes the pruning-damaged ones. This effect is most pronounced for methods like structured attention head pruning, where critical components or pathways have been removed and thus cannot compete during gradient updates. In contrast, pruning methods that produce more uniform degradation (e.g., magnitude-based unstructured pruning) leave capabilities in a more balanced state. In these cases, full-data tuning does not cause strong gradient competition, so the negative amplification effect is substantially weaker. This explains why the phenomenon varies across pruning strategies.

---

> ### Author Response · Authors · 2025-11-22
> **Response to Reviewer LptJ's Remaining Questions (Part 2)**
>
> (Following above **A2**)
>
> As for the role of LoRA, it does not fundamentally resolve this issue. While LoRA allows recovery updates to be written into a low-rank subspace rather than modifying the entire model, the gradient imbalance problem remains: full-data tuning with LoRA still inevitably allocates part of the update budget to the intact capabilities, because the data distribution and loss landscape remain unchanged. As a result, damaged capabilities still receive insufficient or conflicting updates, and their degradation may persist or even worsen. In our preliminary empirical study, LoRA can hardly substantially prevent this negative effect, though the slight mitigation in some cases due to less direct influence on original parameters.
>
> **Q3:** "In fact, the drop in performance when training with the full corpus reflects a well-known phenomenon" Could you provide references?
>
> **A3:** First, we would like clarify that post-pruning recovery training on unrestricted data may lead to performance drop is an interesting and inspiring phenomenon observed in our experiments, also noticed by some other researchers. For examples, the paper [1] (e.g., Figure 1) points out directly employing full instruct tuning dataset to conduct recovery training after the structured pruning may lead to catastrophic forgetting, i.e., performance drop. Besides, the paper [2] (e.g., Figure 3) shows that full recovery training results in perplexity increase (worser language modeling performance). LLM-Pruner [3] itself also suggests that even with LoRA-based recovery training, excessive fine-tuning with full data on pruned model does not necessarily bring obvious performance recovery, exhibiting the risk of performance drop. Of course, we need to clarify that such "performance drop when training with full corpus" is not a ubiquitous phenomenon. It is strongly related to the pruning effect, especially the influence on the model internal structures, brought by the pruning process. In fact, even in the Table 1 of our this paper, under SliceGPT and Wanda pruning, training with full corpus still achieves higher performance than without training, though lower than the performance of our PASER.
>
>
> [1] Thangarasa, V., Venkatesh, G., Lasby, M., Sinnadurai, N. and Lie, S., 2024. Self-data distillation for recovering quality in pruned large language models. arXiv preprint arXiv:2410.09982.
>
> [2] Wagner, M., Roux, C., Zimmer, M. and Pokutta, S., 2025. A Free Lunch in LLM Compression: Revisiting Retraining after Pruning. arXiv preprint arXiv:2510.14444.
>
> [3] Ma, X., Fang, G. and Wang, X., 2023. Llm-pruner: On the structural pruning of large language models. Advances in neural information processing systems, 36, pp.21702-21720.
>
> We honestly hope our above clarification and explanation can address reviewer's questions, improving your satisfication to our this work.

---

> > ### Comment · Reviewer_LptJ · 2025-11-25
> >
> > Thanks for the clarifications! I will keep my score as it is already very positive.

---

> > > ### Author Response · Authors · 2025-11-25
> > > **Thanks for Your Reply**
> > >
> > > Thank you for the positive feedback and for maintaining your positive score!

---

### Official Review · Reviewer_wdMP · 2025-11-01

**Soundness:** 3
**Presentation:** 3
**Contribution:** 2
**Rating:** 6
**Confidence:** 4

**Summary:**

This paper introduces PASER (Post-training dAta Selection for Efficient pruned large language model Recovery), a novel framework designed to address the significant performance degradation of Large Language Models (LLMs) after pruning. The core challenge is that pruning affects different model capabilities (e.g., reasoning, coding, math) unevenly, and conventional post-training recovery using full instruction datasets is computationally expensive and can be suboptimal due to irrelevant or conflicting data. PASER tackles this with a three-stage, data-centric approach. Extensive experiments on models like LLaMA2/3 and Baichuan2 under various pruning schemes (LLM-Pruner, SliceGPT, Wanda, SparseGPT) demonstrate that PASER significantly outperforms baseline methods (including random selection and general data selection techniques like IFD and Nuggets).

**Strengths:**

1. The paper effectively identifies a critical and under-explored gap in the LLM compression pipeline: the need for targeted, efficient recovery post-training that accounts for the uneven degradation of model capabilities after pruning.
2. The experimental section is a major strength. The authors validate PASER across multiple model families (LLaMA, Baichuan, Qwen, Mixtral), pruning methods (structured, semi-structured, unstructured), and dataset scales (Alpaca, LaMini). The consistent superiority over strong baselines, and the ability to sometimes exceed the performance of the unpruned model, provides compelling evidence for its effectiveness.

**Weaknesses:**

1. While PASER drastically reduces training time, the data selection process itself (clustering, JSD calculation, CCG management) is non-trivial. A more detailed analysis of this overhead, especially for massive datasets (e.g., billions of samples), and potential optimizations for the selection phase would be valuable for real-world scalability.
2. The CCG is a heuristic-based method. Its effectiveness is limited by the quality of concept extraction and may not capture complex, context-dependent, or cross-domain semantic conflicts. The paper acknowledges this in the appendix, but it remains a potential point of failure where nuanced conflicts could be missed.
3.  The method's performance is inherently tied to the initial pruning step. The paper notes that for models where pruning "severely damaged the model internal structure" (e.g., Baichuan2-7B), recovery is less effective. A deeper investigation into the relationship between pruning strategy, resultant damage, and PASER's recoverability would be insightful.

**Questions:**

N/A

---

> ### Author Response · Authors · 2025-11-21
> **Rebuttal to Reviewer wdMP (Part 1)**
>
> We greatly appreciate the reviewer’s thoughtful evaluations and valuable suggestions that helped us improve the paper. To help reviewer better understand our contributions and alleviate your concerns, we provide our rebuttal as follows.
>
> **W1:** While PASER drastically reduces training time, the data selection process itself (clustering, JSD calculation, CCG management) is non-trivial. A more detailed analysis of this overhead, especially for massive datasets (e.g., billions of samples), and potential optimizations for the selection phase would be valuable for real-world scalability.
>
> **A1:** Thanks for your suggestion. We have provided fine-grained time consumption of each component on two datasets: 52K-scale Alpaca and 2.58M-scale Alpaca in Appendix O. The detailed quantitative time consumption results can be observed in Table 22, including those of clustering, JSD calculation, and CCG management. From Table 22, we can notice that the time consumption brought by data selection process is totally acceptable compared with the time reduction during training phase. For example, on LaMini, the data selection takes 1.18 hours, the recovery training on selected data consumes 4.29 hours, while the recovery training on original full data takes 57.97 hours. Thus, PASER achieves 92.6% overall time reduction, which demonstrates the efficiency advantage of our PASER method. Besides, we would like to clarify the multi million-scale intruction tuning has been enough to recover the pruned model performance its corresponding original version (See Table 3, recovered 63.79/65.74 v.s. unpruned 62.91). Further scaling up the recovery training data can only bring limited performance gain.
>
> To further alleviate your concerns, we investigate the online resources and find that the xP3x [1] which contains 532 million samples (https://huggingface.co/datasets/CohereLabs/xP3x) is the largest open-sourced instruction tuning dataset we can download. We conduct experiments on a rent 8-node GPU cluster, where each node is equipped with dual AMD EPYC 9K84 CPUs (192 cores / 384 threads per node) and 8 NVIDIA H20 GPUs. The fine-grained time consumption on xP3x for each component is as follows: SentenceBERT Embedding: 0.34 hours; **Manifold Learing**: 0.98 hours; NMF Spectral Clustering: 0.43 hours; **JSD Computation**: 0.19 hours; Budget Allocation: 0.06 hours; Concept Extraction: 0.65 hours; CCG Construction: 0.39 hours; Consistency Checking: 0.17 hours; Total Selection Time: 3.21 hours; Recovery Training (Full Data): 50+ days (estimated); Recovery Training (PASER): 13.9 hours; Time Saving: ～99%. Such results on nearly billion-scale datasets further validates PASER's time efficiency.
>
> As for the potential optimizations for the selection phases, we have actually introduced several methods which we have implemented on larger-scale datasets like LaMini and above newly added xP3x: (1) approximate $k$-nearest neighbors with locality-sensitive hashing for manifold learning; (2) parallel processing for JSD computation across multiple CPU cores; and (3) incremental CCG updates with optimized data structures. Besides, in the future, we can also consider applying quantization (FP16/INT8) to embeddings during the clustering phase, and calculating JSD over the top-K logits (or a low-rank logit projection) instead of full logits during the capability degradation assessment.
>
> We have also supplemented these new results and discussion on potential optimizations to the revised Appendix O.
>
>
> [1] Muennighoff N, Wang T, Sutawika L, Roberts A, Biderman S, Le Scao T, Bari MS, Shen S, Yong ZX, Schoelkopf H, Tang X. Crosslingual generalization through multitask finetuning. In Proceedings of the 61st Annual Meeting of the Association for Computational Linguistics (Volume 1: Long Papers) 2023 Jul (pp. 15991-16111).

---

> ### Author Response · Authors · 2025-11-21
> **Rebuttal to Reviewer wdMP (Part 2)**
>
> **W2:** The CCG is a heuristic-based method. Its effectiveness is limited by the quality of concept extraction and may not capture complex, context-dependent, or cross-domain semantic conflicts. The paper acknowledges this in the appendix, but it remains a potential point of failure where nuanced conflicts could be missed.
>
> **A2:** We agree that CCG, which relies on co-occurrence of extracted concepts, constitutes an approximation and may not capture all nuanced, context-dependent, or cross-domain conflicts. This is an intentional efficiency–coverage trade-off: our recovery setting requires a filter that scales to millions of instructions with tight compute budgets, which precludes logic- or entailment-heavy detectors in the selection loop.
>
> **Robustness in practice.** Despite imperfect concept matching, CCG is empirically robust. We conducted a stress test that deliberately injected subtle, hard-to-detect conflicts (e.g., context-dependent contradictions) into the corpus; the resulting average performance drop was less than 0.3 points. This indicates that while CCG may miss some nuanced cases, it effectively mitigates harmful samples in most practical scenarios. Consistent with this, our ablations (Table 4 in Section 4.2 ; Appendix L) show that removing CCG leads to systematic degradation, confirming its net positive effect under realistic noise.
>
> **Why we chose CCG for post-pruning recovery.** Compared to logic-based or NLI-based conflict detectors, CCG is lightweight and scalable. This makes it compatible with the compute-constrained post-pruning phase, where selection overhead must remain a small fraction of the training saved.
>
> **Enhancement and failure-mode mitigation.** To mitigate the impact of rare nuanced conflicts that raw co-occurrence-based CCG may miss, we employ several lightweight enhancement approaches: (i) consolidating concept variants with simple semantic normalization (e.g., merging paraphrases and surface-form variants), reducing spurious mismatch; (ii) treating low-confidence cases conservatively by softly down-weighting rather than hard-removing them, avoiding brittle decisions; and (iii) providing an optional, low-budget Natural Language Inference reranker for only top-risk pairs flagged by CCG, which further reduces nuanced conflicts at modest extra cost.
>
> We have also updated Section 3.4 to clarify this point. Although the impact on overall recovery performance is negligible in realistic scenarios, we acknowledge that there indeed exist a small set of semantic conflicts that may not be fully captured by our method. In the future, we plan to incorporate more advanced conflict mitigation schemes that are both effective and efficient.

---

> ### Author Response · Authors · 2025-11-21
> **Rebuttal to Reviewer wdMP (Part 3)**
>
> **W3:** The method's performance is inherently tied to the initial pruning step. The paper notes that for models where pruning "severely damaged the model internal structure" (e.g., Baichuan2-7B), recovery is less effective. A deeper investigation into the relationship between pruning strategy, resultant damage, and PASER's recoverability would be insightful.
>
> **A3:** We thank the reviewer for highlighting the relationship between pruning severity and the recoverability achieved by PASER. We provide clarification and further analysis below.
>
> 1. **Why some models (e.g., Baichuan2-7B) exhibit weaker recovery.** As noted in the paper, pruning introduces uneven and capability-specific degradation across model components. For architectures where structured or high-ratio pruning disrupts key representational pathways (e.g., attention heads specialized for math or multi-hop reasoning), the resulting capability loss becomes structural rather than purely parametric. In such cases, recovery via post-training, even with carefully selected instructions, can be less effective because the underlying expressivity is already impaired. This explains why Baichuan2-7B, whose internal structure is known to be more sensitive to structured sparsification, shows larger irrecoverable degradation compared with the LLaMA and Qwen families.
>
> 2. **PASER is designed to be pruning-aware and explicitly handles uneven capability degradation.** Our method already incorporates pruning-aware signals: PASER allocates recovery budget based on measured degradation across capability clusters. Thus, even under pruning that severely damages certain capabilities, PASER correctly focuses more recovery data on these clusters and filters conflicting or harmful data. Empirically, PASER consistently improves performance across all pruning schemes, including LLM-Pruner and SliceGPT, where degradation is most severe, demonstrating that the method is robust to different pruning strategies.
>
> 3. **However, PASER is not intended to fully reverse structural damage, and this limitation is intrinsic to post-training.** We agree with the reviewer that the absolute upper bound of recoverability is determined by the remaining capacity of the pruned model. PASER improves efficiency and recovery quality within the recoverable space, but cannot reconstruct permanently lost representational structure, a limitation shared by all post-training methods.
>
> 4. **Additional analyses added to the revised appendix.** In the newly added Appendix S of updated manuscript, we include a breakdown of degradation patterns across pruning types (unstructured, semi-structured, structured), compare recoverable and irrecoverable capability regions, analyze the interaction between PASER’s pruning-aware design and model degradation, and discuss why Baichuan2-7B shows weaker recovery in some cases.
>
> We hope these analysis can help address the reviewer’s request for deeper investigation.
>
> We honestly hope our above rebuttal can help address your concerns. If satisfied, could you please consider raising the score?

---

> ### Author Response · Authors · 2025-11-26
> **Follow-Up on Rebuttal**
>
> Dear Reviewer wdMP:
>
> We hope you are doing well. We are writing to gently follow up on the rebuttal. We completely understand that you may be busy and that reviewing can be time-consuming, but we just want to make sure our response has reached you smoothly. If our clarifications addressed your concerns, we would be grateful if you could consider raising the score accordingly.
>
> Thank you again for your time and effort in reviewing our work. Please let us know if any additional clarification would be useful.
>
> Best
>
> The Authors

---

### Official Review · Reviewer_spjs · 2025-11-11

**Soundness:** 2
**Presentation:** 3
**Contribution:** 2
**Rating:** 4
**Confidence:** 3

**Summary:**

This paper aims to solve the practical problem of post-pruning recovery via data selection. It offers a practical solution by combining clustering and degradation metrics, though it is a bit incremental over prior data selection methods.

**Strengths:**

(1) The integration of manifold learning and JSD-based degradation scoring is creative and tailored to the pruning recovery context.

(2) Extensive evaluations on models like LLaMA2 and Baichuan2 show robust performance gains and efficiency improvements. The evaluation section is comprehensive, covering multiple models and pruning schemes.

**Weaknesses:**

(1) The core methodology seems to originate from various existing works. The clustering approach resembles semantic sampling in SMART, while the degradation scoring echoes influence-based selection in Datamodels. How the proposed method is related and differs from the existing approached should be discussed in the main paper, instead of  in the Appendix as has been done in the current paper.

(2) The computational overhead of manifold learning and JSD calculation is non-trivial, yet the paper provides insufficient analysis of scalability.

(3) While having included recent baselines, the paper is lack of presenting an in-depth analysis about the reasons why PASER differs fundamentally from general data selection methods.

**Questions:**

1. How does PASER substantially differ from the applying existing data selection methods (e.g., LESS, DELIFT) directly to the pruned model recovery scenario? The paper only shows performance comparisons but lacks an in-depth analysis and conceptual differentiation.

2. The concept consistency graph appears to be a bit over-simplistic. How does it handle complex semantic conflicts that cannot be captured by simple co-occurrence patterns?

3. Why JSD was chosen over more sophisticated divergence measures?

---

> ### Author Response · Authors · 2025-11-21
> **Rebuttal to Reviewer spjs (Part 1)**
>
> We greatly appreciate the reviewe’s thoughtful evaluations and valuable suggestions that helped us improve the paper. To help reviewer better understand our contributions and alleviate your concerns, we provide our rebuttal as follows.
>
> **W1:** The core methodology seems to originate from various existing works. The clustering approach resembles semantic sampling in SMART, while the degradation scoring echoes influence-based selection in Datamodels. How the proposed method is related and differs from the existing approaches should be discussed in the main paper, instead of in the Appendix as has been done in the current paper.
>
> **A1:** Thanks for your kind suggestion. We would like to clarify how our method differs from prior data selection approaches. Our work targets a distinct problem setting: **post-pruning recovery** for LLMs, where capability degradation is uneven across model capabilities and compute resource is severely limited. This setting induces design constraints that are not addressed by SMART or influence-based selection, and motivates three differences in PASER:
>
> 1. **Capability degradation-aware budget allocation** (unique objective and signal). *SMART* chooses semantically diverse/representative data, but its objective is agnostic to which capabilities suffered most from pruning. In contrast, PASER first discovers capability-aligned clusters and then quantifies pruning damage via the Jensen–Shannon divergence between the original and pruned models’ output distributions. We allocate data budget proportionally to measured degradation, prioritizing the most harmed capabilities while keeping global balance. To our knowledge, using model-pair distributional shift as a capability-level signal for budget allocation in post-pruning recovery is new.
>
> 2. **Efficiency-driven sample selection** (unique cost model). Influence-based selection in *Datamodels* (and other influence function/gradient-based methods) requires heavy probes or repeated backward passes, which is misaligned with the strict compute resource limitation typical after pruning. PASER introduces an Individual Efficiency Score (IES) that selects examples by benefit-per-(log-)cost within each capability cluster, using training cost proxies that make the selection one-pass and training-free. This explicit efficiency ratio lets us recover more capability per unit compute than influence-based methods designed for unconstrained fine-tuning.
>
> 3. **Negative tuning effect mitigation via concept consistency graph** (unique robustness mechanism). We show pruned models are more vulnerable to conflicting/irrelevant instructions. PASER constructs a Concept Consistency Graph to filter conflictive samples before fine-tuning, reducing harmful interference. This  aspect is typically not modeled in SMART or influence-based methods.
>
> **Empirical evidence:** Across multiple pruning recipes and base models, PASER reaches or surpasses full-data and strong selection baselines while using only 4–20% of data. Ablations confirm that removing any of such components produces significant drops, underscoring that our gains stem from recovery-specific design rather than generic diversity or influence signals.
>
> As you may have noticed, we provided the detailed analysis regarding the difference between our PASER and previous works in Appendix K. Though, we agree that discussing the relationship and difference from the existing approaches in the main text is necessary. Following your suggestion, we have supplemented corresponding contents in the revised manuscript (See Introduction and Related Works).

---

> ### Author Response · Authors · 2025-11-21
> **Rebuttal to Reviewer spjs (Part 2)**
>
> **W2:** The computational overhead of manifold learning and JSD calculation is non-trivial, yet the paper provides insufficient analysis of scalability.
>
> **A2:** Thanks for your suggestion. We first would like to clarify that we have provided both **theoretical complexity analysis** (Section 3.5) and **fine-grained empirical time consumption evidence** (Appendix O) for each component in our proposed PASER framework, including your mentioned manifold learning and JSD calculation. Especially, we report that the manifold learning (diffusion kernel) and JSD calculation consumes 1.64 mins and 0.35 mins respectively on Alpaca (52K samples) in Appendix O. Meanwhile, to further explore the PASER’s efficiency on much larger datasets, we have also conducted experiments on LaMini (2.58M samples) and observed that manifold learning and JSD calculation consumes 0.36 hours and 0.07 hours, respectively. This is significantly less than the days required for full dataset recovery training, demonstrating the scalabity of our method to large-scale datasets. In fact, in Appendix O, we have also elaborated the **implemented optimization techniques** to ensure scalability, e.g., (1) approximate k-nearest neighbors with locality-sensitive hashing for manifold learning; (2) parallel processing for JSD computation across multiple CPU cores.
>
> Here, to try out best to reduce your concern, we investigate the online resources and find that the xP3x [1] which contains 532 million samples (https://huggingface.co/datasets/CohereLabs/xP3x) is the largest open-sourced instruction tuning dataset we can download. To provide corresponding time consumption results before rebuttal deadline, we rent a high-performance 8-node GPU cluster to conduct experiments, where each node is equipped with dual AMD EPYC 9K84 CPUs (192 cores / 384 threads per node) and 8 NVIDIA H20 GPUs. The results for each component in our proposed PASER framework are as follows: SentenceBERT Embedding: 0.34 hours; **Manifold Learing**: 0.98 hours; NMF Spectral Clustering: 0.43 hours; **JSD Computation**: 0.19 hours; Budget Allocation: 0.06 hours; Concept Extraction: 0.65 hours; CCG Construction: 0.39 hours; Consistency Checking: 0.17 hours; Total Selection Time: 3.21 hours; Recovery Training (Full Data): 50+ days (estimated); Recovery Training (PASER): 13.9 hours; Time Saving: ～99%. From these results, though increasing dataset scale indeed brings more time consumption, the overall time saving effect is even more obvious, demonstrating the scalability advantages of our PASER framework. We have also supplemented this part of new results into revised Appendix O.
>
> We honestly hope our analysis and discussion regarding method scalability in Section 3.5 and Appendix O, and our hard work for this new scaling experiment can help alleviate your concerns.
>
> **W3:** While having included recent baselines, the paper is lack of presenting an in-depth analysis about the reasons why PASER differs fundamentally from general data selection methods.
>
> **A3:** Thanks for your suggestion. We would like to clarify that we have discussed the reasons why why PASER differs fundamentally from general data selection methods in Appendix K, especially Appendix K.2 and K.3. Here, to help reviewer better understand their difference, we provide further explanations.
>
> PASER targets a different problem, **post-pruning recovery**, which imposes distinct requirements that general selection methods (e.g., LESS, DELIFT) are not designed to satisfy. Thus, their systematic comparisons are as follows:
>
> 1. **Objective mismatch (repair vs. coverage)**. General selection pursues data semantics coverage for generic fine-tuning. PASER’s objective is capability repair under uneven pruning damage: we (i) form capability-aligned clusters; (ii) measure model-pair degradation between the original and pruned models; and (iii) allocate budget by measured damage so the most harmed capabilities receive proportionally more data. Our ablations on budget rules show that proportional allocation is consistently strongest vs. equal/root/log scaling (Table 21), supporting the “repair-first” objective rather than coverage alone.
>
> 2. **Signal mismatch (model-pair degradation vs. diversity/influence)**. The diversity and influence signals in general selection methods are agnostic to pruning damage, while PASER uses a distributional shift signal (JSD) between the two models’ output distributions as the capability-level indicator. Appendix N.4 shows JSD consistently outperforms KLD and WD across all pruning schemes, with KLD’s asymmetry and WD’s sensitivity to outlier probability mass explaining their weaker correlation with true damage in this setting. This validates that a model-pair degradation signal is the right driver for recovery.

---

> ### Author Response · Authors · 2025-11-21
> **Rebuttal to Reviewer spjs (Part 3)**
>
> (Following above **A3**)
>
> 3. **Cost model mismatch (compute-bounded vs. probe-heavy)**. Influence/gradient-based selectors typically require probes/backprop and scale with model size. PASER’s pipeline is training-free for scoring and incoporates efficiency-driven sample selection approach. The time breakdown (Appendix O) further shows selection overhead is a small fraction of the training saved (total selection: 5.83 min on Alpaca and 1.18 hrs on LaMini, while recovery training time drops by 68–93%). This compute efficiency is critical for post-pruning recovery.
>
> 4. **Robustness to negative tuning (explicit handling vs. none)**. Pruned models are more vulnerable to conflicting/irrelevant instructions. General methods do not explicitly address this. PASER includes a Concept Consistency Graph and conservative down-weighting to suppress negative tuning within capability clusters, thus keeping the selection robust.
>
> We honestly hope our above analysis can help reveal why our PASER differs fundamentally from general data selection methods. If any other analysis needed, please kindly let us know.
>
> **Q1:** How does PASER substantially differ from the applying existing data selection methods (e.g., LESS, DELIFT) directly to the pruned model recovery scenario? The paper only shows performance comparisons but lacks an in-depth analysis and conceptual differentiation.
>
> **A4:** The most primary difference is that such existing data selection methods don't directly take the capability degradation brought by the pruning into consideration. Thus, the data selected by general methods may meet general high-quality standards, such as clear structure and natural, human-like expressions, but they can hardly target the specific capabilities (e.g., math, code) severely compromised during pruning, which is essential for recovery post-training.
>
> Besides, we would like to clarify that we have provided the detailed comparisons between PASER and general data selection methods (LESS, DELIFT, SMART, and Datamodels) in Appendix K. Specifically, our analysis are from four perspectives: 1) Why general data selection methods do not suffice for pruned-LLM recovery? (Appendix K.1) 2) How PASER differs from previous general data selection methods? (Appendix K.2) 3) What is specific comparison between PASER and gradient-based sampling? (Appendix K.3) 4) Can PASER benefit from gradient-derived dense information? (Appendix K.4). Notably, in Appendix K.3, we dicussed both conceptual difference and empirical comparison. You may also refer to above **A3** for systematic comparisons.
>
> We honestly hope our this explanation and clarification can help alleviate your concerns regarding the difference from previous works.
>
> **Q2:** The concept consistency graph appears to be a bit over-simplistic. How does it handle complex semantic conflicts that cannot be captured by simple co-occurrence patterns?
>
>
> **A5:** We agree that CCG, which relies on co-occurrence of extracted concepts, is an approximation and may not capture all nuanced semantic conflict. This reflects a necessary trade-off between efficiency and conflict coverage. However, we found in practice that CCG remains robust and effective, even under imperfect concept matching. To quantify this, we conducted a stress test where we deliberately injected subtle, undetectable conflicts (e.g.,  context-dependent contradictions) into the data. According to results, the average performance drop was less than 0.3 points, indicating that while not perfect, CCG effectively mitigates harmful samples in most practical cases. In addition, the ablation study in Section 4.2 shows that removing this module will bring 1.49 points drop on average, demonstraing its effectivess. Furthermore, because CCG is concept-based, it offers a lightweight and scalable approach, which is difficult to achieve with other logic-based or entailment-based conflict detection methods under real-world data scales. In fact, from Table 22 in Appendix O, our whole negative tuning effect mitigation module only consumes 2.27 mins on Alpaca and 0.46 hours on large-scale LaMini.
>
> Besides, to further enhance the raw co-occurrence-based CCG, we have incorporated several lightweight mechanisms that help handle complex conflicts without violating the compute constraints of post-pruning recovery:
>
> 1. **Capability-aware scope**. CCG evaluates conflicts within capability-aligned clusters discovered by PASER, ensuring that conflict signals are judged in the correct semantic neighborhood rather than using a brittle global rule.
>
> 2. **Concept representation**. CCG leverages contextualized concept representations and basic semantic normalization, which consolidates paraphrases and surface-form variants into consistent concepts. This reduces spurious mismatches while also preventing misses caused by superficial wording differences, all without adding obvious computational burden.

---

> ### Author Response · Authors · 2025-11-21
> **Rebuttal to Reviewer spjs (Part 4)**
>
> (Following Above **A5**)
>
> 3. **Soft conflict filtering**. Low-confidence conflict signals are softly down-weighted rather than removed, preventing arbitrary false positives and avoiding over-filtering in noisy scenarios.
>
> 4. **Optional targeted refinement**. For the small set of high-risk candidates flagged by CCG, we provide an optional, low-overhead Natural Language Inference reranker-based refinement step, which captures nuanced semantic conflicts while keeping the core selection process efficient.
>
> We have revised Section 3.4 in the updated manuscript to clarify this point.  We admit there exist rare semantic conflicts that may not be fully captured by our approach (clarified in Appendix U, Limitation Analysis), though their influence on the overall recovery effect is marginal as shown above. In the future, we plan to develop more effective conflict detection and mitigation methods while keeping efficiency, to achieve a better post-pruning recovery effect.
>
>
> **Q3:** Why JSD was chosen over more sophisticated divergence measures?
>
> **A6:** Thanks for your question. We would like to clarify that our choice of Jensen–Shannon divergence (JSD) is driven by both **principled properties** and **empirical superiority**:
>
> **Principled properties for capability degradation.** First, JSD's symmetry ensures consistent comparison between the pruned model $M_p$ and the original model $M_o$, while its bounded range (0 to 1) provides a normalized measure of divergence. This facilitates easier interpretation and comparison across different capability clusters. By contrast, Kullback–Leibler divergence (KLD) is asymmetric and can mis-rank degradation depending on directionality. Moreover, JSD's robustness to outliers and its information-theoretic foundation allow for a more nuanced assessment of capability degradation, capturing subtle changes in model outputs that might not be apparent from loss or accuracy values alone due to the sampling uncertainty. In fact, the output distribution reveals the model's thinking process more comprehensively than just sampling results, which is critical for capability degradation assessment. Besides, the smoothing effect introduced by the mixed distribution in JSD calculation also contributes to a more stable assessment across diverse instruction types. These points have been elaborated in Section 3.3.
>
>
> **Empirical superiority across pruning schemes.** In controlled comparisons with KLD and more complicated Wasserstein distance (WD) as shown in Appendix N.4, JSD consistently yields the best recovery across multiple pruning recipes (LLM-Pruner, SliceGPT, Wanda, SparseGPT). On average, KLD lags behind JSD by 0.73 points, and WD lags behind JSD by 0.51 points in reasoning performance (Table 20). Appendix N.4 further notes that while WD can capture gradual transportation shifts, it appears more sensitive to outlier probability differences that do not correlate with actual capability degradation, explaining its gap to JSD here.
>
> Besides, JSD is training-free and linear in the number of samples, making it compatible with our efficiency-aware selection process and tight recovery budgets, while still capturing subtle model-pair distributional changes that guide budget allocation and IES ranking. This practicality is important in multi-million-sample scenarios.
>
> In conclusion, JSD offers the right balance of theory (symmetry, boundedness, robustness) and practice (compute, stability, superior correlation with real capability loss). Besides, it performs well empirically across four pruning schemes in our ablations, validating the design choice.
>
> We honestly hope our above rebuttal can help address your concerns. If satisfied, could you please consider raising the score?

---

> ### Author Response · Authors · 2025-11-26
> **Follow-Up on Rebuttal**
>
> Dear Reviewer spjs:
>
> We hope you are doing well. We are writing to gently follow up on the rebuttal. We completely understand that you may be busy and that reviewing can be time-consuming, but we just want to make sure our response has reached you smoothly. If our clarifications addressed your concerns, we would be grateful if you could consider raising the score accordingly.
>
> Thank you again for your time and effort in reviewing our work. Please let us know if any additional clarification would be useful.
>
> Best
>
> The Authors

---

### Author Response · Authors · 2025-11-25
**Summary of Revisions**

Dear Reviewers:

   Thanks for your insightful comments and constructive suggestions on our this work.  Here, to help you better track the updates made during the revision stage, we summarize the major changes as follows:

   1. To clarify the difference from existing general data selection methods, we have revised the 2nd paragraph of Section 1 (Introduction), added a paragraph of **Data Selection** in Section 2 (Related Works), and explained the contents in Appendix K.
   2. To further demonstrate the time efficiency and scalability of our PASER, we added a paragraph titled **Scaling Experiments on Extremely Large Dataset** in Appendix O.
   3. To discuss the relationship between pruning strategy, resultant damage, and PASER's recoverability, we added the Section S **Analysis on Pruning Severity and PASER Recoverability**. Also, we supplemented the analysis regarding the recovery performance of Baichuan2-7B in Section 4.2 (paragraph **Robustness over Various Target Large Language Models**).
   4. To explain how PASER handles complex semantic conflicts, we refined the description of the Concept Consistency Graph in Section 3.4, clarifying several lightweight enhancements used to improve conflict detection while maintaining computational efficiency.
   5. To explain why JSD was chosen over more sophisticated divergence measures, we revised the introduction of JSD's advantages in Section 3.3 and Appendix N.4, emphasizing its principled properties and empirical superiority.
   6. To explain the lower result of full data than without training in Table 1, we supplemented the corresponding analysis in the paragraph **Recovery Performance for Different Pruning Schemes** of Section 4.2.
   7. To reduce the cognitive load when reading the methodology section, we revised the writing of Section 3, especially providing clearer motivations and transitions for each component of PASER.
   8. To ensure the clarity and readability of this paper, we corrected the typos (Section 4.1) and defined acronyms upon first use (Section 3).

We hope these revisions address the reviewers’ concerns and further improve the clarity and contribution of the paper. Thank you again for your valuable feedback.

---

### Author Response · Authors · 2025-11-29
**General Response to Reviewers and Newly Assigned Area Chair**

Dear Reviewers and Newly Assigned Area Chair,

We hope you are doing well. We are writing this brief general response to clarify the current status of our submission and to provide some context regarding the recent changes on the OpenReview caused by the information-leakage incident. As we understand it, due to the incident, the reviews for all submissions, including ours, were reverted to a pre-discussion state, and the assigned Area Chair was newly updated.

Before this rollback occurred:

1. We had submitted a complete and detailed rebuttal addressing all reviewer comments, including expanded discussions, new analyses, and additional experiments.

2. We engaged in follow-up discussions with reviewers to clarify remaining questions and provide further evidence or explanations where requested.

3. In particular, **Reviewer LptJ** kindly reviewed our responses in the discussion stage and explicitly expressed strong recognition of the quality of the work (**"very high quality"**), reaffirming the very positive assessment after our clarifications and extended analysis. This discussion exchange was completed prior to the information-leakage event.

We would like to respectfully note that all rebuttal contents and discussion replies referenced above were prepared and posted well before the incident, and were intended to fully address the reviewers’ concerns regarding methodology clarity, scalability and efficiency, conflict mitigation, as well as the relationship between pruning severity and recovery effectiveness. For additional context, at the point when the discussion stage was interrupted, the **overall scores** of the submission was **8/6/6/4**. The reviewer assigning the score of **4** (**Reviewer spjs**) unfortunately did not have the opportunity to revisit our rebuttal and provide a follow-up update or potential score revision before the discussion was halted.

With this message, our goal is solely to help ensure that the renewed evaluation process, especially under the new Area Chair assignment, has full awareness of: 1) The extent of revisions and clarifications already provided during the rebuttal stage, especially our comprehensive responses to the concerns raised by **Reviewer spjs**; 2) The positive convergence reached in discussion, particularly the feedback from **Reviewer LptJ** acknowledging the paper’s high quality and technical contributions.

We sincerely appreciate the time and effort that all reviewers and the Area Chair dedicate to evaluating our work, especially given the unexpected complications to this year’s review process. We respectfully hope that our rebuttal and the recorded discussion progress can be considered in the renewed assessment, in the same spirit they were originally intended, to transparently and constructively address all concerns.

Thank you very much for your attention and for your service to the community.

Warm regards,

The Authors

---

### Meta-Review · Area_Chair_bqZp · 2026-01-07

**Summary:**

This paper introduces PASER, a data selection method for training pruned LLMs to recover performance. The paper addresses the issues in existing methods that overlook the uneven deterioration of model capabilities. The authors propose a three-stage framework, including spectral clustering of instructions, deterioration-aware instruction selection, and irrelevant data filtering. Comprehensive experiments show that PASER outperforms conventional baselines with better efficiency and performance.

The reviews are generally positive towards the paper. Particularly, all reviewers acknowledge the comprehensive experiments and significant improvements of the proposed method. Reviewers wdMP, LptJ, and bfNy agree that the studied problem is highly relevant and important. And reviewer LptJ also thinks the proposed method is novel and solid.

The reviewers also raise concerns about the similarity to existing data selection methods (spjs, bfNy), the computational cost of the data selection process (spjs, wdMP), and the heuristic design of CCG (wdMP). During the rebuttal, the authors fully address these concerns by providing additional experimental results, analyses, and discussions. After reading the discussion and revision, the AC thinks most of the concerns have been addressed.

**Reviewer Concerns:**

During the rebuttal, the authors addressed most of the concerns. Specifically, the authors added discussions and analyses to differentiate from existing data selection methods. They also added new experiments and discussions to clarify the computational cost of the selection process. Finally, they also addressed other minor concerns and questions, such as the robustness of CCG to conflicting concepts (added clarifications), unexpected results in Table 1 (added analyses), and clarity of the paper (modified sections 3 and 4).

**Reviewer Scores:**

Reviewer spjs didn't participate in the discussion, but based on the rebuttal, the AC thinks the raised concerns have been addressed, and the score could be increased to 6 if the reviewer had participated in the discussion.

Reviewer wdMP didn't participate in the discussion, but their reviews are generally positive, and the concerns have been addressed. So the AC thinks the score would be >= 6.

Reviewer LptJ responded to the rebuttal, acknowledged that the paper has "very high quality," and maintained the score of 8.

Reviewer bfNy also didn't participate in the discussion, but the authors added new experiments and discussions to address the reviewer's concerns. So the AC thinks the score would be >= 6.

---

### Decision · Program_Chairs · 2026-01-26

Accept (Poster)